# Which Examples Preserve Decisions? Verified Inverse Design for Model Selection

## Abstract

Model selection recurs across deployment slices and robustness tests, while training and evaluation data can be expensive to acquire, retain, or repeatedly process. We study decision-preserving acquisition (DPA): given a fixed model-selection protocol and an observed multi-scenario winner map, identify compact training data support that reproduces the same winners with prescribed margins after retraining. Unlike coreset or data-evaluation objectives, DPA targets the stability of the model-selection decision itself. In an idealized convex ERM regime, DPA admits a Karush–Kuhn–Tucker feasibility view, while computing an exact minimum-cost design is NP-hard even in a simplified setting. These results characterize the computational expense of the idealized inverse problem. We propose VERDICT (**VER**ified **D**ecision **I**nverse **C**onsistency **T**raining), a propose-and-verify method for DPA. VERDICT derives a local decision-response operator, uses trust-region surrogates over working sets to propose lower-cost designs, and accepts a candidate only after rerunning the fixed protocol verifies the target margins under the stated decision convention. The resulting certificate is relative to the declared protocol and decision convention. We evaluate VERDICT in settings where verification is feasible, including decision-preserving subset selection under a train-and-verify setting and evaluation subset selection for a declared fixed candidate set.

## 1 Introduction

Model selection is a standard step in machine-learning pipelines: practitioners compare candidate model classes or hyperparameter configurations and choose a winner under a validation protocol (Feurer et al., 2015). In many applications, this protocol goes beyond a single average score: candidates are compared across *scenarios*, by which we mean protocol-specified evaluation contexts such as domains (Sagawa et al., 2019; Koh et al., 2021), data slices (Zhang et al., 2022; Tae & Whang, 2021), or noisy/robustness conditions (Birgé et al., 2013; Raschka, 2018; Ma et al., 2021). The observed selection outcome can be thought of as a scenario-wise winner map, which records the candidate selected in each context.

In such settings, practitioners may want to know which parts of the available data are actually needed to support the observed selection outcome, and whether the declared protocol omits scenarios on which that outcome would change. We therefore study an inverse, protocol-relative question: with the candidate set, training routine, validation scenarios, and decision convention fixed, what compact training or evaluation evidence is sufficient to reproduce the observed scenario-wise winners within some prescribed margins?

Existing subset selection, coreset, and data evaluation methods typically target predictive performance, full-data approximation, or the effect of individual examples on a scalar quantity (Mirzasoleiman et al., 2020; Ghorbani & Zou, 2019; Koh & Liang, 2017). These are useful goals, but they do not directly address our target: choosing low-cost training evidence so that a fixed model-selection protocol reproduces the same scenario-wise winners after retraining. We call this problem *decision-preserving acquisition* (DPA). DPA searches for compact or lower-cost acquisition designs that preserve the observed winner map with prescribed margins, and is useful in several protocol-relative settings. First, it can identify the compact evidence support of an observed model-selection decision, providing an audit of which training examples are needed to reproduce that outcome. Second, it can diagnose an underspecified evaluation protocol: failures on

held-out scenario families reveal comparisons that should be added to the declared scenario set. Third, it can construct compact evaluation subsets for repeated comparisons within a specified candidate family, so that the one-time subset construction cost is amortized over future evaluations. These uses do not require the selected evidence to approximate the full data distribution or preserve every possible future model; they require preservation of the comparisons encoded by the declared protocol.

This yields a decision-preserving view of data acquisition: the forward protocol maps an acquisition design to trained candidates and scenario-wise winners, while the inverse problem seeks a low-cost design that reproduces an observed winner map with prescribed margins. We call the map **realizable** under a budget if some admissible design reproduces it after retraining. This separates two questions: whether such a design exists and how to find one. In convex ERM, realizability admits a KKT feasibility characterization, while computing an exact minimum-cost decision-preserving design is NP-hard even in a simplified setting. These results characterize the structure and computational difficulty of the idealized inverse problem. We address the practical search problem with VERDICT (VERified Decision Inverse Consistency Training), a propose-and-verify method that uses implicit differentiation and trust-region surrogates over working sets to generate lower-cost candidates, which are then checked by rerunning the original protocol.

**Scope of verification.** Throughout the paper, "verified" refers to the acceptance procedure used by VERDICT: a proposed design is accepted only after the fixed training and evaluation protocol is rerun and the prescribed decision-margin constraints are checked under the stated decision convention. Verification is therefore relative to the declared protocol and randomness convention. Under a fixed-seed convention, the certificate applies to the specified seed; under an aggregated convention, it applies to the specified seed set and aggregation rule. Preservation under additional seeds or modified protocols requires separate verification.

**Contributions.** We make the following contributions:

- **Decision-preserving acquisition.** We formulate data acquisition around preserving the outcome of a fixed model-selection protocol. Given an observed multi-scenario winner map, DPA seeks low-cost training evidence that reproduces the same winners with prescribed margins after retraining.
- **Verified local search.** We derive a local decision-response operator that predicts how acquisition changes affect decision margins, and use it to propose lower-cost candidate designs. Candidates are accepted only after rerunning the fixed training/evaluation protocol verifies the required margins.
  **Structural context.** In an idealized convex ERM regime, we show that decision preservation can be expressed through a KKT feasibility system, and computing an exact minimum-cost decision-preserving design is NP-hard. The KKT characterization provides the basis for the local response analysis, while the hardness result provides computational context for the ideal global objective. Neither result implies that VERDICT is globally or approximately optimal.
- **Empirical results and downstream applications.** Our empirical focus is on settings where the fixed protocol can be rerun and decision preservation can be verified directly: fixed-feature convex ERM for training-subset acquisition and fixed-model evaluation-subset selection. Across all datasets, VERDICT finds compact verified subsets while random, stratified, influence-function-based, and diversity-based baselines often fail to preserve the same decision map at matched sizes. We also demonstrate a scenario-discovery application: held-out failures identify omitted decision-critical scenarios, and after adding them back, VERDICT preserves the expanded scenario set where all same-size baselines fail.

## 2 Related work

### 2.1 Subset selection, coresets, and budgeted training-data acquisition

A large literature studies subset selection for training efficiency or performance preservation. Coreset-style methods select a subset (or reweight samples) so that training on the reduced set closely matches training on the full dataset, in empirical risk minimization and deep learning pipelines (Mirzasoleiman et al., 2020). Related ideas appear in core-set formulations for active learning under labeling costs (Sener & Savarese, 2018), and in data evaluation methods that score data by estimated contribution to validation objectives (Ghorbani & Zou, 2019). Recent work studies budgeted data pruning and selection in large-scale regimes, including settings

where data curation changes scaling behavior (Sorscher et al., 2022). In large language model fine-tuning, several methods propose task-specific data selection procedures under limited budgets (Wang et al., 2025). These lines of work usually target predictive performance or training efficiency. They do not encode a winner-take-all rule defined by comparing multiple models across scenarios, nor do they impose margin constraints intended to preserve an observed selection map under a stated decision convention. Similarly, AUTOMATA accelerates hyperparameter tuning using gradient-based, configuration-specific weighted subsets that are adaptively updated during training (Killamsetty et al., 2022).

## 2.2 Training-data attribution and influence estimation

A complementary literature asks which training points influence model behavior or a validation outcome. Influence functions estimate the effect of upweighting or removing a training point on a downstream metric (Koh & Liang, 2017). Gradient-trajectory methods such as TracIn approximate influence using gradient dot products along the training path (Pruthi et al., 2020), and TRAK develops a scalable attribution technique for large models (Park et al., 2023). Recent work examines influence estimation in large language models and other high-dimensional settings, including rescaled and robust variants intended to improve stability (Rubinstein & Hopkins, 2025; Tu et al., 2025), and evaluations that report failure modes of standard influence estimates on large models (Li et al., 2025). Other attribution approaches use only the final trained model to estimate data importance (Wei et al., 2025), and some introduce alternative approximations such as zeroth-order influence estimators (Kokhlikyan et al., 2025). Outlier-focused analyses propose signals for identifying detrimental or mislabeled samples (Chhabra et al., 2025). Extensions to other generative model families yield scalable influence-based data attribution methods (Mlodozeniec et al., 2025). Our decision-response operator is related to influence functions in that it maps data perturbations to changes in a downstream quantity. The differences are the outcome and the goal: we map acquisition perturbations to shifts in decision margins across a quasi-active model set, and use this local model to propose constrained acquisition updates for an inverse feasibility problem.

## 2.3 Inverse optimization, bilevel structure, and decision-focused learning

Inverse optimization studies how to choose inputs so that observed decisions are optimal for a forward optimization problem (Heuberger, 2004). When the forward problem is convex, KKT conditions yield reformulations as MPECs (Luo et al., 1996) or other bilevel structures. Surveys provide overviews of bilevel optimization techniques and challenges (Colson et al., 2007). Recent work develops scalable inverse optimization methods in kernelized or large-scale settings (Long et al., 2024). Decision-focused learning incorporates the downstream decision problem into the training objective, often by differentiating through a proxy of the combinatorial optimizer or using a relaxation (Donti et al., 2017; Wilder et al., 2019). A survey summarizes foundations and benchmarks in decision-focused learning (Mandi et al., 2024). Our setting is closer to an inverse feasibility problem for a discrete decision rule induced by scenario-wise model comparisons. We address this with a verified bilevel template: a continuous relaxation proposes candidate acquisitions, and acceptance requires retraining-based checks when verification is feasible.

# 3 Problem Setup

We formalize decision-preserving acquisition under a model-selection protocol that fixes a set of candidate models, a training routine, validation scenarios, validation functionals, and a decision convention. The design variable is the acquired training evidence. Starting from a baseline design that induces an observed winner in each scenario, our goal is to find a lower-cost design that, after retraining the candidates under the same protocol, preserves those winners with specified margins. All notational uses are introduced here and Appendix C.1.

We use the term **scenario** to denote a protocol-defined evaluation context indexed by $t \in \mathcal{T}$, under which candidate models are compared via validation functionals $F_{m,t}$. Scenarios may correspond to domains, fixed data slices, robustness conditions, or other evaluation contexts specified by the protocol. For model $m \in \mathcal{M}$ and scenario $t$, $F_{m,t}$ denotes the validation loss used to compare candidates in that context, with lower values

preferred. A **design** $\alpha$ specifies the acquired training evidence used by the pipeline to train each candidate model. We write $F_{m,t}(\alpha)$ for the validation loss obtained by training model $m$ under design $\alpha$ and then evaluating it in scenario $t$. Given a baseline design and a fixed decision convention, the **winner** in scenario $t$, denoted $D_{\mathrm{val}}(t) \in \mathcal{M}$, is the candidate selected by that convention.

To preserve this decision with **stability margin** $\gamma_t \geq 0$, the original winner must remain ahead of every competing model under the same protocol:

$$F_{m,t}(\alpha) - F_{D_{\mathrm{val}}(t),t}(\alpha) \geq \gamma_t \quad \forall m \in \mathcal{M} \setminus \{D_{\mathrm{val}}(t)\}.$$

The decision-preserving acquisition problem is to search for a low-cost design $\alpha$ that satisfies these margin constraints after the pipeline is rerun. Thus feasibility is defined by the forward protocol itself: candidate models are retrained under $\alpha$, evaluated on the fixed scenarios, and checked against the original winner map.

**Candidate Pool, Models, and Scenarios.** We fix a pool of training examples $\mathcal{P} = \{z_i\}_{i=1}^N \subset \mathcal{Z}$, a set of models or hyperparameter configurations $\mathcal{M} = \{1, \ldots, M\}$, and a set of validation scenarios $\mathcal{T} = \{1, \ldots, T\}$.

**Design variables and retained subset size.** A design specifies which evidence units from the candidate pool are retained. We encode a design by

$$\alpha = (\alpha_i)_{i \in [N]} \in \{0,1\}^N,$$

where $\alpha_i = 1$ means that evidence unit $z_i$ is included in the protocol run and $\alpha_i = 0$ means that it is omitted.

The retained cardinality is

$$|\alpha| := \sum_{i=1}^N \alpha_i.$$

We require a minimum valid subset size $n_{\min} \geq 1$. Given a cardinality budget $B$, define

$$\mathcal{S}_B(n_{\min}) := \left\{ \alpha \in \{0,1\}^N : n_{\min} \leq |\alpha| \leq B \right\}.$$

For surrogate optimization, we use the relaxed set

$$\mathcal{S}_B^{\mathrm{rel}}(n_{\min}) := \left\{ \alpha \in [0,1]^N : n_{\min} \leq \sum_{i=1}^N \alpha_i \leq B \right\}.$$

Relaxed designs are used only to construct proposals; candidates accepted by VERDICT are integral and are checked by rerunning the fixed protocol.

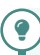 *Remark* 3.1 (Fixed-quality formulation and multi-quality extension). The fixed-quality inclusion model is the setting studied throughout this paper. A natural extension would allow an evidence unit to be acquired at one of several annotation, measurement, preprocessing, or fidelity levels with different costs; jointly selecting evidence units and their quality levels is left for future work.

**The Forward Pipeline.** Given a design $\alpha$, the pipeline acquires training data according to $\alpha$, trains each candidate model, and selects a winner by comparing validation functionals across scenarios. Let $w_m(\alpha)$ be the parameters returned by the training protocol. When training is stochastic, $w_m(\alpha)$ is defined under the randomness convention used for selection and verification, which is specified in Appendix C.3. Each scenario $t$ specifies a validation functional $F_{m,t}(w)$ evaluated at trained parameters, and we write

$$F_{m,t}(\alpha) := F_{m,t}(w_m(\alpha)).$$

**Inverse Design Interpretation.** The forward pipeline maps an acquisition design $\alpha$ to trained models, validation losses, and ultimately a scenario-wise winner map. This gives a decision-preserving selection problem: given observed winners under a baseline design, search for a lower-cost design whose forward execution reproduces those winners with the prescribed margins.

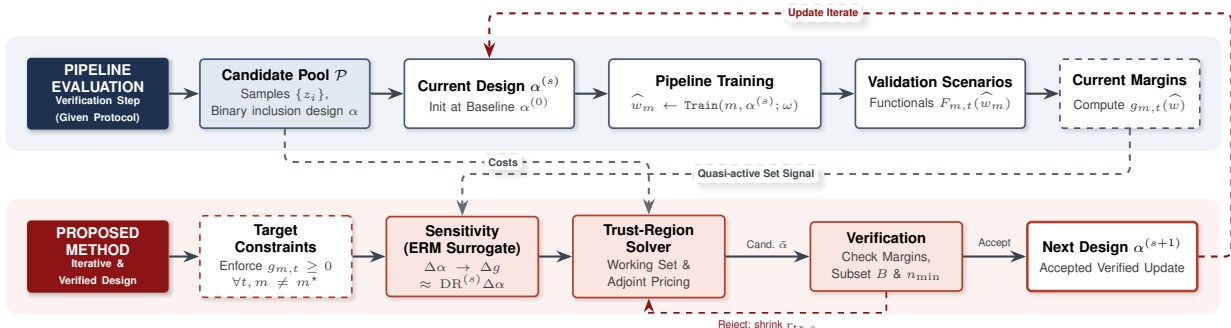

Figure 1: **Overview of decision-preserving acquisition. Left:** given a design $\alpha$, a protocol trains models and evaluates validation functionals across scenarios to form margin constraints. **Right:** we propose a smaller-subset candidate $\tilde{\alpha}$ using a local decision-response model, then accept it only after retraining verifies all margins under the decision convention that defines the observed winners.

**Relaxation and Structural Constraints.** The underlying subset-selection problem is combinatorial because a valid design is an inclusion vector $\alpha \in \{0, 1\}^N$. To construct proposals, we solve surrogate problems over the continuous relaxation $\tilde{\alpha} \in [0, 1]^N$. In the convex ERM setting, the coordinates of $\tilde{\alpha}$ act as per-example training weights. In general pipeline settings, the relaxed proposal is rounded to an integral design $\hat{\alpha} \in \{0, 1\}^N$ before retraining and verification. All decision and operational constraints are ultimately checked on $\hat{\alpha}$.

We impose the minimum-size constraint $\sum_{i=1}^N \alpha_i \geq n_{\min}$ during surrogate optimization and require $|\hat{\alpha}| \geq n_{\min}$ after rounding. This rules out degenerate subsets that contain too little evidence for the model-selection protocol to produce a meaningful comparison. When a cardinality budget $B$ is imposed, we analogously enforce $\sum_i \alpha_i \leq B$ in the relaxation and $|\hat{\alpha}| \leq B$ on the verified integral design.

## 3.1 Decision map, margins, and verification

Fix a baseline design $\alpha^{(0)}$ and a decision convention, either a fixed-seed convention or an aggregated convention over a finite seed set, with deterministic tie-breaking. For each scenario $t \in \mathcal{T}$, let $D_{\text{val}}(t) \in \mathcal{M}$ denote the observed winner under this convention. To preserve decisions with stability margins $\gamma_t \geq 0$, we enforce for every competitor $m \neq D_{\text{val}}(t)$ the constraint

$$g_{m,t}(\alpha) := F_{m,t}(\alpha) - F_{D_{\text{val}}(t),t}(\alpha) - \gamma_t \geq 0.$$

Under the aggregated convention, we define

$$\bar{g}_{m,t}(\alpha) := \bar{F}_{m,t}(\alpha) - \bar{F}_{D_{\text{val}}(t),t}(\alpha) - \gamma_t,$$

where $\bar{F}_{m,t}(\alpha)$ aggregates $F_{m,t}$ over the seed set as specified in Appendix C.3. Appendix C provides the operational semantics of the relaxation, the full randomness convention used for verification, and the margin selection protocol.

## 3.2 Inverse Objective and Verification

The ideal fixed-quality DPA problem minimizes retained subset cardinality subject to preserving the baseline selection map with margins and operational constraints, under the same convention used to define $D_{\text{val}}$:

$$\min_{\alpha \in \{0,1\}^N} |\alpha| \quad \text{s.t.} \quad g_{m,t}(\alpha) \geq 0 \quad \forall t \in \mathcal{T}, \ \forall m \neq D_{\text{val}}(t),$$

together with the minimum-size and protocol-validity constraints. When relaxed surrogates are used, we round to $\alpha^b$, and acceptance and verification are performed on $\alpha^b$ under the decision convention in Definition C.2.

# 4 Decision Response and Verified Preservation

This section formalizes the decision-preservation problem and derives the local response model used by VERDICT. We first show that, in convex ERM, decision preservation can be written as a feasibility problem through the lower-level KKT conditions of the trained models. This provides a formal reference point for the inverse view, but does not make the associated cost-minimization problem easy: even simple convex instances remain combinatorial. We then derive local response operators that approximate how acquisition changes affect decision margins. In general training pipelines, these operators are used only to propose candidate updates; accepted designs are validated by rerunning the fixed protocol and checking the prescribed margin constraints directly.

For $\boldsymbol{w} = (w_m)_{m \in \mathcal{M}} \in \prod_{m \in \mathcal{M}} \mathcal{W}$, define

$$g_{m,t}(\boldsymbol{w}) := F_{m,t}(w_m) - F_{D_{\text{val}}(t),t}(w_{D_{\text{val}}(t)}) - \gamma_t.$$

For a design $\boldsymbol{\alpha}$, write $g_{m,t}(\boldsymbol{\alpha}) := g_{m,t}(\boldsymbol{w}^\star(\boldsymbol{\alpha}))$ in the convex empirical risk minimization world and $g_{m,t}(\boldsymbol{\alpha}; \omega) := g_{m,t}(w_{\text{pipe}}(\boldsymbol{\alpha}; \omega))$ in the pipeline world. Under the aggregated convention, $\bar{g}_{m,t}(\boldsymbol{\alpha})$ is defined by the same decision convention in Definition C.2. This notation lets us express decision preservation either as constraints on trained parameters or, via the forward map, as constraints on designs.

## 4.1 Inverse Feasibility via KKT Conditions in Convex ERM

In the convex ERM regime, the relaxed forward map $\alpha \mapsto w^\star(\alpha)$ is defined by solving a strongly convex objective for each model. This allows us to replace retraining by an optimality system and to phrase inverse realizability as feasibility of a single set of constraints. For each candidate model $m \in \mathcal{M}$, let

$$L_m(w; \alpha) := R_m(w) + \frac{1}{N} \sum_{i=1}^{N} \alpha_i \, \ell_m(w; z_i)$$

denote the subset-weighted ERM objective under a relaxed design $\alpha \in \mathcal{S}_B^{\text{rel}}(n_{\min})$.

**Definition 4.1** (Normal Cones). For a closed convex set $C \subset \mathbb{R}^d$, $N_C(x)$ denotes the convex-analytic normal cone at $x$.

**Assumption 4.2** (Convex Forward Regime). For every $m \in \mathcal{M}$ and every $\boldsymbol{\alpha} \in \mathcal{S}_B^{\text{rel}}(n_{\min})$, the map $w \mapsto L_m(w; \boldsymbol{\alpha})$ is differentiable and $\mu$-strongly convex on $\mathcal{W}$, for some $\mu > 0$ independent of $(m, \boldsymbol{\alpha})$.

Under Assumption 4.2, each forward solution satisfies the variational inclusion

$$0 \in \nabla_w L_m(w; \boldsymbol{\alpha}) + N_{\mathcal{W}}(w). \tag{1}$$

**Definition 4.3** (Budgeted inverse KKT system). Fix a cardinality budget $B > 0$. The inverse system consists of variables $(\alpha, w)$ satisfying

$$\alpha \in \mathcal{S}_B^{\text{rel}}(n_{\min}), \tag{2}$$
$$0 \in \nabla_w L_m(w_m; \alpha) + N_W(w_m), \qquad \forall m \in \mathcal{M}, \tag{3}$$
$$g_{m,t}(w) \geq 0, \qquad \forall t \in \mathcal{T}, \ \forall m \neq D_{\text{val}}(t). \tag{4}$$

Integral subset designs correspond to

$$\alpha \in \mathcal{S}_B(n_{\min}) \subseteq \mathcal{S}_B^{\text{rel}}(n_{\min}).$$

**Proposition 4.4** (Equivalence of Realizability and Inverse Feasibility). *Suppose the convex-forward assumption holds. The observed decision map is realizable under cardinality budget $B$ in convex ERM if and only if the inverse system above is feasible. The proof for Proposition 4.4 is in Appendix D.*

## 4.2 Local Response and Decision-Response Operators

Theorem D.1 concerns the exact global minimum-cardinality DPA problem. VERDICT does not solve or approximate this global objective. Instead, we build a local proposal model around a verified reference design that preserves the decision map. The goal is to identify locally lower-cost candidates by approximating how trained parameters, consequently decision margins, change under small perturbations of the retained subset.

**Assumption 4.5** (Reference-design regularity). Assume that there exists a possibly relaxed reference design $\alpha^\sharp$ such that
$$g_{m,t}(\alpha^\sharp) \geq 0 \qquad \forall t \in \mathcal{T}, \ \forall m \neq D_{\mathrm{val}}(t).$$
Let $w_m^\sharp := w_m^\star(\alpha^\sharp)$. Assume that $w_m^\sharp \in \mathrm{int}(W)$ and that $H_m^\sharp := \nabla_w^2 L_m(w_m^\sharp; \alpha^\sharp)$ is positive definite for every $m \in \mathcal{M}$.

**Assumption 4.6** (Local smoothness). Assume that $w \mapsto \ell_m(w; z_i)$ and $w \mapsto R_m(w)$ are twice differentiable near $w_m^\sharp$, and that the validation functionals used in the quasi-active set are differentiable near $w_m^\sharp$.

**Lemma 4.7** (Implicit differentiation of trained parameters). *Under the preceding assumptions, for any perturbation $\Delta\alpha \in \mathbb{R}^N$,*
$$Dw_m^\star(\alpha^\sharp)[\Delta\alpha] = -(H_m^\sharp)^{-1}\left(\frac{1}{N}\sum_{i=1}^N \Delta\alpha_i \, \nabla_w \ell_m(w_m^\sharp; z_i)\right).$$

**Definition 4.8** (Response operator). For each example $i$, let $r_i^\sharp \in \mathbb{R}^{Md}$ stack the model-wise directions
$$(H_m^\sharp)^{-1}\nabla_w \ell_m(w_m^\sharp; z_i), \qquad m \in \mathcal{M}.$$
Define $\mathrm{Resp}^\sharp : \mathbb{R}^N \longrightarrow \mathbb{R}^{Md}$ by
$$\mathrm{Resp}^\sharp(\Delta\alpha) := -\frac{1}{N}\sum_{i=1}^N \Delta\alpha_i r_i^\sharp.$$
Then, for sufficiently small $\Delta\alpha$,
$$w^\star(\alpha^\sharp + \Delta\alpha) \approx w^\star(\alpha^\sharp) + \mathrm{Resp}^\sharp(\Delta\alpha).$$

**Definition 4.9** (Quasi-active sets and decision response). Fix a threshold $\tau \geq 0$ and define
$$\mathcal{J}_\tau := \left\{(m,t) : m \neq D_{\mathrm{val}}(t), \ t \in \mathcal{T}, \ g_{m,t}(\alpha^\sharp) \leq \tau\right\}.$$
Let $g_{\mathcal{J}_\tau}$ stack the margins indexed by $\mathcal{J}_\tau$. The decision-response operator is
$$\mathrm{DR}^{\sharp,\tau}(\Delta\alpha) := Dg_{\mathcal{J}_\tau}\left(w^\star(\alpha^\sharp)\right) \mathrm{Resp}^\sharp(\Delta\alpha).$$

## 4.3 Local Convex Surrogate and Candidate Expansion

The operators above provide a local linear model of the quasi-active margins around $\alpha^\sharp$. We use this model in a cardinality-reducing surrogate with a trust region, and then use dual information to expand the working set. Fix $\alpha^\sharp$ and $\mathcal{J}_\tau$, and let $s^{\sharp,\tau} := g_{\mathcal{J}_\tau}\left(w^\star(\alpha^\sharp)\right)$. Let $\mathcal{S}$ denote either $\mathcal{S}^{\mathrm{rel}}(n_{\min})$ or, when a hard cardinality budget is imposed, $\mathcal{S}_B^{\mathrm{rel}}(n_{\min})$. We solve

$$
\begin{aligned}
\min_{\alpha,\xi} \quad & \sum_{i=1}^N \alpha_i + \lambda_{\mathrm{lin}}\|\xi\|_1 \\
\text{s.t.} \quad & \alpha \in \mathcal{S}, \\
& \mathrm{DR}^{\sharp,\tau}(\alpha - \alpha^\sharp) \geq -s^{\sharp,\tau} - \xi, \\
& \xi \geq 0, \\
& \|\alpha - \alpha^\sharp\|_2 \leq r_{\mathrm{tr}}.
\end{aligned}
\tag{5}
$$

The slack variable $\xi$ accounts for linearization error and incomplete working sets. Candidate subsets are accepted or rejected by direct protocol verification.

**Proposition 4.10** (Adjoint scoring for working-set expansion). *Let $u \geq 0$ be dual multipliers associated with the linearized margin constraints in Equation 5. Define the example score $s_i := -\left(\left(\mathrm{DR}^{\sharp,\tau}\right)^* u\right)_i$. Examples with large-magnitude scores are predicted to have a large first-order effect on the quasi-active decision margins and can be added to the working set for the next surrogate solve.*

### 4.4 Verified Trust-Region SCA

In the general training setting, the convex surrogate acts as a proposal mechanism. We certify candidates by rerunning the pipeline and verifying margin constraints under the chosen convention, ensuring accepted designs satisfy the requirements.

**Proposition 4.11** (Verification of Accepted Designs). *Fix a decision convention and a training pipeline. Any algorithm accepting a candidate design only after retraining and verifying margin constraints yields decision-feasible designs. If budget and minimum-acquisition constraints are enforced at acceptance (including on rounded designs), the output satisfies these constraints.*

---

**Algorithm 1** Trust-Region Propose-and-Verify

---

**Require:** Baseline subset $\alpha^{(0)}$ feasible under the declared convention; initial radius $r_{\mathrm{tr},0} > 0$; minimum threshold $\tau_{\min} \geq 0$; threshold factor $\beta > 0$; trust-region factors $\beta_+ > 1$ and $\beta_- \in (0,1)$.

1: **for** $s = 0, 1, 2, \ldots$ **do**
2:      Set $\tau_s \leftarrow \max\{\tau_{\min}, \beta r_{\mathrm{tr},s}\}$. Construct the quasi-active set augmented with the pipeline active set.
3:      Compute the local decision-response operator $\mathrm{DR}^{(s)}$ and its associated adjoint scores.
4:      Solve Equation equation 5 with $\alpha^\sharp = \alpha^{(s)}$ and radius $r_{\mathrm{tr},s}$ to obtain a relaxed proposal $\widetilde{\alpha}$.
5:      Round $\widetilde{\alpha}$ to an integral candidate $\alpha^b \in \{0,1\}^N$. If $|\alpha^b| < n_{\min}$, apply completion. Reject the proposal if rounding or completion cannot satisfy the subset-size constraints.
6:      Rerun the forward protocol using $\alpha^b$ and check decision feasibility.
      **Subset-size feasibility:** $|\alpha^b| \geq n_{\min}$,    $|\alpha^b| \leq B$    when a hard cardinality budget is imposed.
7:      **if** all verification checks pass **then**
8:         $\alpha^{(s+1)} \leftarrow \alpha^b$ and $r_{\mathrm{tr},s+1} \leftarrow \beta_+ r_{\mathrm{tr},s}$                        ▷ Accept
9:      **else**
10:        $\alpha^{(s+1)} \leftarrow \alpha^{(s)}$ and $r_{\mathrm{tr},s+1} \leftarrow \beta_- r_{\mathrm{tr},s}$                      ▷ Reject
11:      **end if**
12:      Stop if $r_{\mathrm{tr},s+1} < \mathrm{tol}$ or no valid improvement is found.
13: **end for**

---

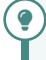

*Remark* 4.12 (The exact minimum-cardinality objective is combinatorial). Proposition 4.4 characterizes feasibility in convex ERM, while Theorem D.1 establishes that exact minimum-cardinality DPA is NP-hard. VERDICT does not solve or approximate this global objective and provides no optimality guarantee.

## 5 Computational Experiments

This section evaluates decision-preserving acquisition across three modalities: tabular, image, and text classification. Our main experiment studies training-subset acquisition under a fixed convex ERM protocol: given a full-data model-selection decision, we ask how much training evidence is needed so that retraining candidate models on a subset preserves the same scenario-wise winners with prescribed margins. We then evaluate two downstream uses of the same decision-preservation framework: decision-critical scenario discovery and low-cost evaluation sets for model selection. We evaluate all methods on nine fixed-feature classification datasets, namely Credit, Bank, Adult from the UCI Machine Learning Repository (Dua & Graff, 2017), OxfordPets (Parkhi et al., 2012), STL-10 (Coates et al., 2011), CIFAR10 and CIFAR100 (Krizhevsky et al., 2009), 20Newsgroups (Lang, 1995), and AG News (Zhang et al., 2015), all of which are treated as supervised classification problems under a fixed-feature convex ERM protocol. All experiments are run on Python

3.11.13 with five seeds on an Intel Xeon Gold 6230R (104 cores), 188 GiB RAM, and an NVIDIA RTX A6000 (48 GiB VRAM). We report mean and standard deviation over randomized methods.

All experiments study fixed-quality subset selection with unit cost per retained example; thus, the objective is retained subset size. For each scenario $t$, the full protocol selects a winner $D_{\text{val}}(t)$ according to validation loss. A candidate acquisition design is feasible if, after executing the same forward protocol under that design, the original winner remains ahead of every competitor by the required margin $F_{m,t}(\alpha) - F_{D_{\text{val}}(t),t}(\alpha) \geq \gamma_t, \quad \forall t, \; m \neq D_{\text{val}}(t)$, where lower validation loss is better and $\gamma_t$ is set as a fraction of the full-data winner gap. Unless otherwise stated, we use $\gamma_t = \eta \, \text{gap}_t$ with $\eta = 0.5$.

## 5.1 Direct Training Acquisition

This experiment directly instantiates the inverse-design formulation in the convex ERM setting. The raw input modality determines the feature map $\phi$, but after feature construction all methods operate on the same fixed design matrix. The full training pool defines the reference protocol: each candidate model is trained on the full pool, evaluated on validation scenarios, and compared by scenario-wise validation loss. This produces the observed winner map $D_{\text{val}}(t)$ and full-data winner gaps.

**Fixed-feature convex ERM protocol and scenarios.** All direct-acquisition experiments use fixed features. Each example $z_i$ is mapped once to $x_i = \phi(z_i)$ before acquisition: frozen image representations for vision, one-hot categorical plus standardized numerical features for tabular data, and TF-IDF/SVD features fit once on the training pool for text. Acquisition therefore changes only the weighted convex ERM objective, not the representation. For every acquired subset, we retrain all candidate models under the same objective and verify the original protocol-level margin constraints on fixed validation scenarios.

A scenario is a validation slice on which candidate models are compared. Vision and text scenarios are class-wise or class-group slices, while tabular scenarios are protocol-defined feature slices such as demographic groups, service categories, risk groups, and quantile buckets. Scenario construction details are given in Appendix F.

We compare VERDICT with six subset-selection baselines: uniform random sampling, class-stratified sampling, HighLoss, GradNorm, K-center, and DecisionIF. HighLoss and GradNorm select examples with the largest reference-model training loss and gradient norm, respectively. K-center performs greedy farthest-first selection in the frozen feature space, using a class-balanced heuristic for large datasets. DecisionIF is a one-shot decision-influence baseline that estimates, at the full-data solution, the first-order effect of removing each training example on quasi-active decision margins and retains examples whose removal is predicted to be most harmful. The loss-space family selects training subsets using the geometry or mean of per-example loss vectors computed under the full-data reference models, before retraining and verifying the resulting decision map. [1] Meanwhile, VERDICT starts from the full feasible design and iteratively proposes deletions using a local decision-response model. At each iteration, it forms a quasi-active constraint set, solves a greedy trust-region deletion surrogate, retrains all candidate models on the proposed subset, and accepts the proposal only if the original winner map, margins, and operational constraints are verified. We run VERDICT for 30 proposal iterations and report ablations of $\eta$ and VERDICT components in Appendices G.2 and G.3.

**Accepted and rejected proposals.** Table 1 reports the smallest retained training fraction at which each method passes the same retrain-and-check decision-preservation test. VERDICT, HighLoss, GradNorm and DecisionIF are deterministic and have no seed variance. VERDICT achieves the best verified fraction on 8 of 9 datasets, with especially large gains in vision: STL10 and CIFAR100 require only 7.3% and 2.0% of the training set respectively, while HighLoss, GradNorm and coverage-based baselines require substantially larger fractions or fail within the evaluated budget grid. This indicates that preserving a model-selection decision can require a much smaller and more targeted support than preserving generic training-set coverage. The only exception is Adult, where stratified sampling outperforms VERDICT. This suggests a regime in which the winner map is largely preserved by coarse scenario balance rather than by the decision-response

---

[1] We evaluate three loss-space baselines: loss-$k$-center, approximate loss-$k$-medoids, and FW-Mean. Tables 1 and 2 report the best-performing loss-space baseline for each dataset, while the complete per-method results are provided in Appendix G.4.

signal alone. VERDICT is most useful when feasible subsets are rare under generic sampling and depend on decision-relevant examples, while simpler sampling can be competitive when the protocol is highly redundant.

Table 1: Smallest retraining-verified retained fraction found on the evaluated budget grid for exact decision feasibility; lower is better. Randomized methods show mean $\pm$ standard deviation over 5 seeds. Entries $> 0.950$ indicate that the method did not become feasible within the evaluated budget grid.

| Task | Dataset | $n$ | Methods | | | | | | | |
|------|---------|-----|--------|-----------|----------|----------|---------|-----------|-----------|---------|
| | | | Random | Stratified | HighLoss | GradNorm | KCenter | DecisionIF | Loss-space | VERDICT |
| Tabular | Credit | 30000 | $0.867 \pm 0.029$ | $0.933 \pm 0.029$ | $> 0.950$ | $> 0.950$ | $0.960 \pm 0.042$ | $> 0.950$ | $\geq 0.930$ | **0.427** |
| | Bank | 45211 | $0.520 \pm 0.045$ | $0.540 \pm 0.042$ | $> 0.950$ | $> 0.950$ | $0.740 \pm 0.192$ | $> 0.950$ | $0.540 \pm 0.055$ | **0.413** |
| | Adult | 48842 | $0.520 \pm 0.329$ | $\mathbf{0.360 \pm 0.055}$ | $0.900$ | $0.900$ | $0.970 \pm 0.045$ | $> 0.950$ | $\geq 0.850$ | 0.619 |
| Vision | OxfordPets | 7349 | $> 0.950$ | $> 0.950$ | $> 0.950$ | $0.950$ | $> 0.950$ | $0.350$ | $> 0.950$ | **0.221** |
| | STL10 | 13000 | $0.850 \pm 0.100$ | $0.883 \pm 0.104$ | $> 0.950$ | $0.950$ | $0.990 \pm 0.022$ | $0.350$ | $0.910 \pm 0.022$ | **0.073** |
| | CIFAR10 | 50000 | $0.917 \pm 0.104$ | $0.833 \pm 0.029$ | $0.950$ | $0.950$ | $0.860 \pm 0.108$ | $0.950$ | $\geq 0.830$ | **0.144** |
| | CIFAR100 | 50000 | $0.550 \pm 0.132$ | $0.417 \pm 0.076$ | $> 0.950$ | $> 0.950$ | $0.500 \pm 0.122$ | $0.300$ | $0.460 \pm 0.089$ | **0.020** |
| Text | 20Newsgroups | 18846 | $0.817 \pm 0.076$ | $0.800 \pm 0.000$ | $> 0.950$ | $0.800$ | $0.800 \pm 0.035$ | $> 0.950$ | $0.620 \pm 0.027$ | **0.458** |
| | AG News | 127600 | $0.950 \pm 0.000$ | $0.950 \pm 0.000$ | $> 0.950$ | $> 0.950$ | $0.950 \pm 0.000$ | $> 0.950$ | $0.950$ | **0.544** |

**Verified search behavior.** Table 1 does not reveal whether feasibility is rare, whether the local surrogate is well calibrated, or whether generic sampling already succeeds at comparable budgets. Table 2 separates these regimes by examining random feasibility at VERDICT's final budget and the behavior of the verified local search. The accepted-path columns show that reductions in data are not obtained through only infinitesimal edits: several datasets admit large verified deletions in a single accepted proposal, followed by additional verified reductions. This supports the intended use of the response model as a large-step proposal mechanism for the inverse design problem.

The Random@final diagnostic asks whether VERDICT's final retained fraction is also feasible for unstructured random subsets. For most datasets, this rate is zero, indicating that VERDICT's final subsets are not typical random feasible subsets at the same size. Bank is the main exception: random subsets sometimes satisfy the constraints at VERDICT's final retained fraction, suggesting that the corresponding winner map has more redundancy under the chosen scenarios. Even there, VERDICT still attains the smallest retained fraction in Table 1, so random feasibility at the VERDICT budget suggests that the decision-preservation constraints are relatively redundant or insensitive to which exact examples are selected at that retained fraction. The rejected-proposal columns show why retrain-and-check verification is necessary. In every dataset, the surrogate eventually proposes a candidate with positive predicted margin that becomes infeasible after retraining, sometimes with winner flips. The effect is especially pronounced on Adult, Bank, STL10, and CIFAR10, where surrogate-feasible proposals can become clearly infeasible after retraining. The violations/winner-flips column further separates two boundary effects: some rejected proposals only break the prescribed margin, while others change the selected winner in at least one scenario.

Table 2: Verified search diagnostics for direct train-and-verify DPA.

| Dataset | Final frac. | Random@ VERDICT final frac. | Accept | Reject | Largest accepted deletion | Final verified margin | First rejected surrogate→ verified | Violations/ winner flips |
|---------|-------------|------------------------------|--------|--------|---------------------------|-----------------------|-------------------------------------|--------------------------|
| Credit | 0.427 | 0/50 | 23 | 7 | 1077 | $7.27 \times 10^{-5}$ | $3.80 \times 10^{-5} \to -1.59 \times 10^{-4}$ | 3/3 |
| Bank | 0.413 | 19/50 | 21 | 9 | 2582 | $2.50 \times 10^{-8}$ | $7.36 \times 10^{-7} \to -2.40 \times 10^{-3}$ | 5/4 |
| Adult | 0.619 | 0/50 | 22 | 8 | 1193 | $2.54 \times 10^{-6}$ | $4.30 \times 10^{-6} \to -3.02 \times 10^{-4}$ | 2/2 |
| OxfordPets | 0.221 | 0/50 | 21 | 9 | 313 | $4.96 \times 10^{-4}$ | $6.72 \times 10^{-3} \to -1.04 \times 10^{-2}$ | 1/1 |
| STL10 | 0.073 | 0/50 | 22 | 8 | 426 | $6.70 \times 10^{-4}$ | $7.60 \times 10^{-3} \to -5.57 \times 10^{-2}$ | 5/1 |
| CIFAR10 | 0.144 | 0/50 | 23 | 7 | 2783 | $3.43 \times 10^{-3}$ | $9.60 \times 10^{-4} \to -2.31 \times 10^{-4}$ | 1/0 |
| CIFAR100 | 0.020 | 0/50 | 22 | 8 | 3302 | $1.44 \times 10^{-4}$ | $4.33 \times 10^{-3} \to -5.42 \times 10^{-4}$ | 1/0 |
| 20Newsgroups | 0.458 | 0/50 | 22 | 8 | 501 | $6.58 \times 10^{-7}$ | $1.71 \times 10^{-7} \to -5.79 \times 10^{-4}$ | 1/1 |
| AG News | 0.544 | 0/50 | 19 | 11 | 7421 | $2.09 \times 10^{-6}$ | $2.36 \times 10^{-7} \to -1.99 \times 10^{-4}$ | 1/1 |

**Verified local search trajectories.** Figure 2 and Figure 3 in Appendix G.1 show the verified search process under the 30-proposal protocol. The key diagnostic is the discrepancy between the surrogate-predicted margin and the margin obtained after retraining across proposals. Across datasets, accepted proposals usually

have positive surrogate and retraining-verified margins, indicating that the local response model gives useful directions for reducing the retained fraction. At the same time, rejected proposals expose a consistent failure mode: a proposal may appear margin-preserving under the surrogate while violating the decision margin after retraining. In three representative datasets, the regimes differ: OxfordPets compresses rapidly and then saturates near a verified boundary; Bank benefits from the longer horizon, recovering from early and mid-search surrogate failures; and 20Newsgroups progresses more slowly near the feasibility boundary.

Figure 2: Verified local search trajectories for VERDICT. **Top row**: retained subset fraction across proposals. **Bottom row**: surrogate-predicted minimum decision margin for the corresponding proposals. The margin axis uses a symmetric logarithmic scale to show both small positive margins and negative verification failures.

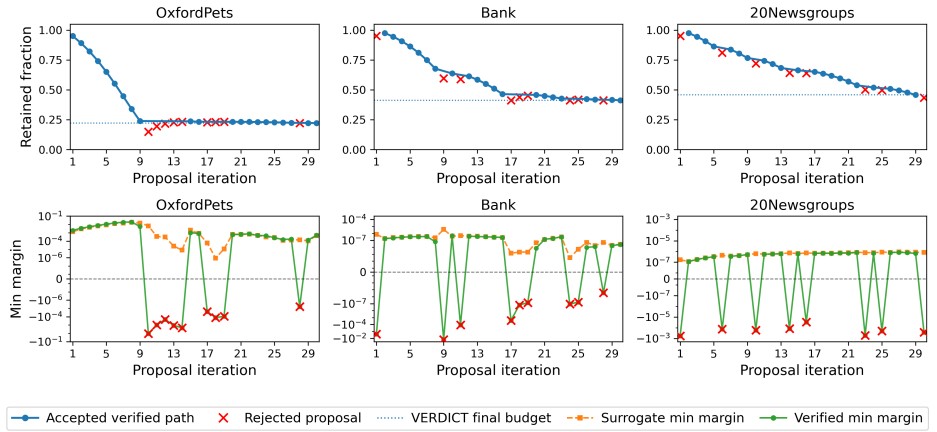

## 5.2 Downstream Applications

**Decision-critical scenario discovery.** DPA can be used to audit whether a declared protocol omits decision-critical scenarios. Since the guarantee is protocol-relative, omitted scenarios are not guaranteed to be preserved. We therefore run VERDICT on a set of scenarios that excludes a scenario family, then evaluate the resulting subset on the held-out family. A held-out failure indicates that the omitted scenario contains constraints that should be added to the declared protocol. After adding the omitted scenarios back, we report the additional retained evidence that VERDICT requires to preserve the expanded protocol. Table 3 summarizes representative examples across tabular, text, and vision data; full details and experimental results on leave-one-scenario audits and per-seed baseline results are presented in Appendix H.1.

Table 3: Decision-critical scenario discovery across three modalities. $\rho_{\text{search}}$ is the retained fraction selected using only the search scenario set; $\rho_{\text{expanded}}$ is the retained fraction after adding the omitted family back. $\Delta_{\min}^{\text{heldout}}$ is the minimum held-out margin before expansion, while $\Delta_{\min}^{\text{expanded}}$ is the final minimum margin after expansion. Negative held-out margins indicate violations.

| Dataset | Omitted family | Held-out audit | | | Expanded protocol | | | |
|---|---|---|---|---|---|---|---|---|
| | | Flips | Viol. | $\Delta_{\min}^{\text{heldout}}$ | $\rho_{\text{search}}$ | $\rho_{\text{expanded}}$ | $\Delta_{\min}^{\text{expanded}}$ | Best baseline feasible |
| Adult | age slices | 2 | 5 | $-2.15\times10^{-3}$ | 0.378 | 0.619 | $2.54\times10^{-6}$ | 0/20 |
| AG News | world news | 1 | 5 | $-8.18\times10^{-2}$ | 0.070 | 0.544 | $2.09\times10^{-6}$ | 0/20 |
| STL10 | failure splits | 3 | 9 | $-5.73\times10^{-2}$ | 0.061 | 0.073 | $6.70\times10^{-4}$ | 0/20 |

**Evaluation subsets for a fixed candidate set.** A second use of DPA is to construct compact evaluation subsets for repeated execution of a fixed model-selection protocol. In this setting, the candidate models are trained as usual, and VERDICT selects a small validation subset whose losses preserve the scenario-wise winners and margins induced by the full validation set. The declared candidate set may include multiple models, hyperparameter configurations, or checkpoints, and the resulting subset provides lower-cost evaluation support for repeated comparison, reproduction, or auditing of that fixed set. Its certificate applies to the specified candidate set and decision convention. We detail this application in Appendix H.2.

## 6 Discussion and Limitations

**Interpreting decision-preserving designs** In many model-selection workflows, the downstream action is a comparison result: which hyperparameters win on a slice, which model remains best under a robustness condition, or which candidate is selected before final training or deployment. DPA therefore targets the pairwise margins that determine this outcome rather than generic data coverage. This helps explain why VERDICT often finds smaller feasible subsets than coverage- or difficulty-based rules. The exceptions are also informative: when the winner map is governed mainly by coarse balance or redundant scenarios, as on Adult and Bank, generic sampling can be competitive. VERDICT is most useful when same-size generic subsets are rarely feasible and preservation depends on examples supporting near-active comparisons.

**Protocol dependence and preservation boundaries** The retained fraction must be interpreted relative to the candidate pool and scenario set. Well-separated candidates or redundant scenarios may admit many feasible subsets, whereas near ties and fine-grained scenarios require broader or more targeted evidence. Accordingly, small retained fractions may indicate compact margin-critical support, while larger fractions may reflect close comparisons or wider coverage needs. The Random@final diagnostic in Tables 1 and 2 helps distinguish these regimes: frequent random feasibility indicates an insensitive or redundant winner map, while low random feasibility indicates stronger dependence on active-comparison structure.

VERDICT's trajectory further reveals the boundary of the decision-preserving region. Early proposals can remove substantial evidence while maintaining positive verified margins; near the boundary, locally plausible deletions are more often rejected after retraining because of margin violations or winner flips. Rapid saturation on datasets such as OxfordPets, CIFAR100, STL10, and AG News contrasts with more gradual or failure-prone paths on Bank, Credit, CIFAR10, 20Newsgroups, and Adult. Thus, final compression and rejected proposals jointly characterize how tightly the protocol depends on its retained evidence.

**Role of the convex ERM analysis.** The convex ERM setting provides a clean reference case for the inverse view of DPA. In that regime, preservation can be expressed through the optimality conditions of the trained candidates, and the associated cost-minimization problem remains combinatorial. We use this analysis to motivate a local proposal model combined with direct verification, rather than to suggest that the surrogate alone characterizes general training pipelines. In the pipeline setting, the response model guides candidate updates, while the original train-and-evaluate protocol determines which designs are retained.

**Scope and limitations.** DPA is protocol-relative: its guarantees apply only to the specified candidate pool, scenarios, margins, costs, and decision convention. Our fixed-feature convex ERM setting isolates decision preservation from representation learning and stochastic training effects, while allowing accepted designs to be certified by explicit retraining or reevaluation. VERDICT trades acquisition or evaluation savings for search cost, and as a local search method can fail when the response model mispredicts margins, rounding breaks feasibility, or the winner map depends on broad coverage rather than near-active comparisons. Its certificates should not be interpreted as robustness, fairness, or deployment guarantees outside the declared scenario set. Extending DPA to end-to-end nonconvex models, highly stochastic pipelines, multi-quality acquisition, and expensive-verification settings remains future work.

## 7 Conclusion

We introduced decision-preserving acquisition, an inverse formulation that asks which acquired evidence is sufficient for a fixed model-selection protocol to reproduce an observed winner map with prescribed margins. VERDICT implements a verified local-search procedure that proposes lower-cost designs and accepts them only after rerunning the fixed protocol. Empirically, the direct train-and-verify experiments show that decision-preserving designs can be much smaller than subsets found by generic coverage-, difficulty-, or influence-based heuristics. The results suggest that model-selection outcomes can have compact but protocol-dependent evidence support, which can be interpreted when verification is feasible.

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

## A    Reproducibility Statement

We provide the information needed to reproduce the experiments in the paper, including the hardware and software environment, dataset splits, preprocessing steps, hyperparameters, implementation choices for VERDICT and the baselines, and the evaluation protocol used across settings. We report results over multiple random seeds where applicable using a consistent aggregation procedure, and we describe the metrics and verification checks used to generate each table and figure. We will release the full code package upon paper acceptance, which includes scripts for preprocessing, training and evaluation, configuration files for our method and all baselines, and instructions for data preparation when external resources are required.

## B    Impact Statement

This work develops an auditing-oriented tool: it can identify low-cost training evidence sufficient to reproduce an observed model-selection map under explicit margins and a specified decision convention. This can support transparency in data curation and quantify how much evidence is needed to support a deployed selection decision. However, the same capability can be misused to rationalize predetermined outcomes by cherry-picking scenarios, margins, candidate pools, or cost models to favor a desired selection. We mitigate this risk by requiring an explicit, documented decision convention (including aggregation and tie-breaking) and verifying feasibility under that convention when retraining is feasible; recommending that scenario sets, pools, and margin protocols be pre-specified and reported; and encouraging sensitivity analyses and the use of independent holdout scenarios not used in the inverse-design loop. Nevertheless, a DPA certificate should not be interpreted as evidence of universal or population-level model superiority, or as a guarantee for candidates, populations, or conditions outside the declared protocol.

## C    Problem Setup Details: Relaxations, Randomness, and Margins

### C.1    Notation and Protocol Conventions

Tables 4–8 summarize the main notation used throughout the paper. We use calligraphic letters for finite sets and roman or italic letters for their cardinalities when clarification is helpful. All decision-preservation statements are relative to the fixed candidate set, scenarios, margins, costs, and decision convention specified by the protocol.

Table 4: Notation for protocol objects.

| Symbol | Meaning |
|---|---|
| $\mathcal{P} = \{z_i\}_{i=1}^{N}$ | Candidate pool of training examples |
| $z_i$ | Training example indexed by $i$ |
| $N$ | Number of examples in the candidate pool |
| $\mathcal{M}$ | Finite set of candidate models or hyperparameter configurations |
| $M = |\mathcal{M}|$ | Number of candidate models |
| $\mathcal{T}$ | Finite set of validation scenarios |
| $T = |\mathcal{T}|$ | Number of validation scenarios |
| $t \in \mathcal{T}$ | Scenario index, e.g., domain, slice, class group, or robustness condition |
| $m \in \mathcal{M}$ | Candidate model or configuration index |
| $F_{m,t}$ | Validation functional for model $m$ on scenario $t$; lower is better |
| $F_{m,t}(\alpha)$ | Validation loss after training model $m$ under acquisition design $\alpha$ |
| $D_{\mathrm{val}}(t)$ | Winner selected by the baseline protocol on scenario $t$ |
| $\omega$ | Randomness used by the training and/or evaluation pipeline |
| $\omega_0$ | Fixed seed used under the fixed-seed decision convention |
| $\overline{F}_{m,t}(\alpha)$ | Aggregated validation loss over a specified finite seed set |

Table 5: Notation for fixed-quality subset designs.

| Symbol | Meaning |
| --- | --- |
| $\alpha \in \{0,1\}^N$ | Integral subset design |
| $\alpha_i$ | Indicator that example $z_i$ is retained |
| $\alpha_i \in [0,1]$ | Relaxed example weight used in surrogate optimization |
| $\widehat{\alpha}$ | Rounded integral subset evaluated by the verification protocol |
| $|\alpha|$ | Retained subset size, $\sum_{i=1}^N \alpha_i$ |
| $B$ | Maximum retained subset size |
| $n_{\min}$ | Minimum subset size required for a valid protocol run |
| $\mathcal{S}_B(n_{\min})$ | Feasible set of integral subsets satisfying the size constraints |
| $\mathcal{S}_B^{\mathrm{rel}}(n_{\min})$ | Relaxed feasible set used by surrogate problems |

Table 6: Notation for training and decision preservation.

| Symbol | Meaning |
| --- | --- |
| $w_m(\alpha)$ | Parameters returned by the training pipeline for model $m$ under design $\alpha$ |
| $w_m^{\star}(\alpha)$ | Exact optimizer for model $m$ in the convex ERM regime |
| $w_{\mathrm{pipe}}(\alpha;\omega)$ | Pipeline output under design $\alpha$ and randomness $\omega$ |
| $\gamma_t$ | Required stability margin for scenario $t$ |
| $g_{m,t}(\alpha)$ | Decision margin against competitor $m$ on scenario $t$ |
| $\overline{g}_{m,t}(\alpha)$ | Aggregated decision margin under the aggregated seed convention |
| $\mathrm{gap}_t$ | Full-data winner gap on scenario $t$ |
| $\eta$ | Fraction used to set margins, e.g., $\gamma_t = \eta\,\mathrm{gap}_t$ |

## C.2 Operational Semantics of Relaxed Subset Designs

The inverse problem uses an integral subset $\alpha \in \{0,1\}^N$, while the local surrogate is solved over $\alpha \in [0,1]^N$. In convex ERM, relaxed values are interpreted as per-example training weights. Before verification, a relaxed proposal is rounded to an integral subset $\widehat{\alpha}$. All decision-margin, minimum-size, and cardinality-budget checks are performed on $\widehat{\alpha}$, not on the relaxed proposal.

## C.3 Randomness Conventions and Verification

When training is stochastic, preserving a selection requires fixing how randomness is handled. We formalize training with an explicit randomness input and verify candidates under the same convention used to define the winners.

**Definition C.1** (Training pipeline protocol). For a seed/protocol specification $\omega$, define

$$\widehat{w}_m(\alpha;\omega) := \mathtt{Train}(m,\alpha;\omega),$$

Table 7: Notation for convex ERM analysis.

| Symbol | Meaning |
| --- | --- |
| $\ell_m(w;z_i)$ | Per-example training loss for model $m$ |
| $R_m(w)$ | Regularizer for model $m$ |
| $L_m(w;\alpha)$ | Acquisition-weighted ERM objective for model $m$ |
| $\mathcal{W}$ | Parameter domain |
| $N_{\mathcal{W}}(w)$ | Normal cone of $\mathcal{W}$ at $w$ |
| $\mu$ | Strong convexity parameter |
| $H_m^{\sharp}$ | Hessian $\nabla_w^2 L_m(w_m^{\sharp};\alpha^{\sharp})$ at a reference design |
| $\alpha^{\sharp}$ | Reference design used to form the local response approximation |
| $w_m^{\sharp}$ | Trained parameters at the reference design $\alpha^{\sharp}$ |

Table 8: Notation for VERDICT local search.

| Symbol | Meaning |
| --- | --- |
| $\Delta\alpha$ | Perturbation to the acquisition design |
| $\mathrm{Resp}^\sharp$ | Local parameter-response operator mapping $\Delta\alpha$ to predicted parameter change |
| $\mathrm{DR}^{\sharp,\tau}$ | Decision-response operator mapping $\Delta\alpha$ to predicted margin change |
| $\tau$ | Quasi-active margin threshold |
| $\mathcal{J}^\tau$ | Quasi-active set of near-binding winner-versus-competitor constraints |
| $s^{\sharp,\tau}$ | Vector of current quasi-active margins at $\alpha^\sharp$ |
| $\xi$ | Slack variable in the local surrogate problem |
| $\lambda_{\mathrm{lin}}$ | Penalty weight on surrogate slack |
| $r_{\mathrm{tr}}$ | Trust-region radius |
| $r_{\mathrm{tr},s}$ | Trust-region radius at VERDICT iteration $s$ |
| $\beta^+, \beta^-$ | Trust-region expansion and shrinkage factors |
| $u$ | Dual multipliers for linearized margin constraints |
| $s_{i,k}$ | Adjoint score used to rank acquisition variables for working-set expansion |
| $\alpha^{(s)}$ | Verified design at VERDICT iteration $s$ |
| $\widetilde{\alpha}$ | Relaxed candidate proposed by the surrogate solver |
| $\widehat{\alpha}$ | Rounded candidate passed to retrain-and-check verification |

$$w_{\mathrm{pipe}}(\alpha;\omega) := (\widehat{w}_m(\alpha;\omega))_{m\in\mathcal{M}}.$$

We observe a winner $D_{\mathrm{val}}(t) \in \mathcal{M}$ for each scenario $t$ under a fixed decision convention, including deterministic tie-breaking.

**Definition C.2** (Observed decision map and randomness convention). Fix a baseline design $\alpha^{(0)}$. The observed selection map $D_{\mathrm{val}} : \mathcal{T} \to \mathcal{M}$ is defined under one of two conventions:

1. **Fixed-seed convention.** There exists a seed $\omega_0$ such that

$$D_{\mathrm{val}}(t) \in \arg\min_{m\in\mathcal{M}} F_{m,t}\big(\widehat{w}_m(\alpha^{(0)};\omega_0)\big),$$

with a deterministic tie-break rule if needed.

2. **Aggregated convention.** Fix a finite seed set $\Omega$ and an aggregator Agg (such as the mean or median). Define

$$\bar{F}_{m,t}(\alpha) := \mathrm{Agg}_{\omega\in\Omega} F_{m,t}\big(\widehat{w}_m(\alpha;\omega)\big),$$

and set

$$D_{\mathrm{val}}(t) \in \arg\min_{m\in\mathcal{M}} \bar{F}_{m,t}(\alpha^{(0)}),$$

with a deterministic tie-break rule if needed.

All verification uses the same convention (including aggregation and tie-breaking) that defines $D_{\mathrm{val}}$.

**Margin Constraints.** Margins convert winner preservation into a stability requirement by enforcing that the winner stays ahead of competitors by at least $\gamma_t$. Fix margin requirements $\gamma_t \geq 0$. For any design $\alpha$ and competitor $m \neq D_{\mathrm{val}}(t)$, define the margin constraint

$$g_{m,t}(\alpha) := F_{m,t}(\alpha) - F_{D_{\mathrm{val}}(t),t}(\alpha) - \gamma_t.$$

Under the aggregated convention, define

$$\bar{g}_{m,t}(\alpha) := \bar{F}_{m,t}(\alpha) - \bar{F}_{D_{\mathrm{val}}(t),t}(\alpha) - \gamma_t.$$

We call $\alpha$ *decision-feasible* if it satisfies the corresponding margin system under the convention defining $D_{\mathrm{val}}$.

### C.4 Margin Selection Protocol

Margins should be chosen so the inverse problem is feasible. Unless otherwise stated, we set margins using the following protocol. The parameter $\eta$ scales the baseline gaps into target margins.

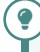 *Remark* C.3 (Margin protocol from a baseline). Fix $\eta \in (0, 1)$ and a baseline design $\alpha^{(0)}$ that produced $D_{\mathrm{val}}$ under the chosen convention.

- **Fixed-seed convention:** define

$$\mathrm{gap}_t(\alpha^{(0)}) := \min_{m \neq D_{\mathrm{val}}(t)} \left( F_{m,t}\big(\widehat{w}_m(\alpha^{(0)}; \omega_0)\big) \right.$$
$$\left. - F_{D_{\mathrm{val}}(t),t}\big(\widehat{w}_{D_{\mathrm{val}}(t)}(\alpha^{(0)}; \omega_0)\big) \right),$$

and set $\gamma_t := \eta \max\{\mathrm{gap}_t(\alpha^{(0)}), 0\}$.

- **Aggregated convention:** define

$$\mathrm{gap}_t(\alpha^{(0)}) := \min_{m \neq D_{\mathrm{val}}(t)} \left( \bar{F}_{m,t}(\alpha^{(0)}) \right.$$
$$\left. - \bar{F}_{D_{\mathrm{val}}(t),t}(\alpha^{(0)}) \right),$$

and set $\gamma_t := \eta \max\{\mathrm{gap}_t(\alpha^{(0)}), 0\}$.
In experiments, we fix $\eta$ (equivalently, $\gamma_t$) as part of the decision-convention specification.

## D Proofs

### D.1 Proof of Proposition 4.4 (Realizability $\Leftrightarrow$ inverse KKT feasibility)

In the convex ERM regime, each model solves

$$w_m^\star(\boldsymbol{\alpha}) \in \arg\min_{w \in \mathcal{W}} L_m(w; \boldsymbol{\alpha}),$$

and margins are evaluated at $\boldsymbol{w}^\star(\boldsymbol{\alpha}) = (w_m^\star(\boldsymbol{\alpha}))_{m \in \mathcal{M}}$. We show that the observed selection map is realizable under budget $B$ if and only if the inverse system in Definition 4.3 is feasible.

($\Rightarrow$) **Realizability implies feasibility.** Assume realizability under budget $B$: there exists $\boldsymbol{\alpha} \in \mathcal{S}_B^{\mathrm{rel}}(n_{\min})$ such that

$$g_{m,t}(\boldsymbol{w}^\star(\boldsymbol{\alpha})) \geq 0 \qquad \forall t \in \mathcal{T}, \ \forall m \neq D_{\mathrm{val}}(t).$$

Let $\boldsymbol{w} := \boldsymbol{w}^\star(\boldsymbol{\alpha})$. By optimality of each $w_m^\star(\boldsymbol{\alpha})$ and the KKT/variational characterization for convex constraints,

$$0 \in \nabla_w L_m(w_m^\star(\boldsymbol{\alpha}); \boldsymbol{\alpha}) + N_{\mathcal{W}}(w_m^\star(\boldsymbol{\alpha})), \qquad \forall m \in \mathcal{M}.$$

Thus $(\boldsymbol{\alpha}, \boldsymbol{w})$ satisfies equation 2–equation 4, hence the inverse system is feasible.

($\Leftarrow$) **Feasibility implies realizability.** Conversely, assume the inverse system is feasible: there exist $(\boldsymbol{\alpha}, \boldsymbol{w})$ such that

$$\boldsymbol{\alpha} \in \mathcal{S}_B^{\mathrm{rel}}(n_{\min}), \qquad 0 \in \nabla_w L_m(w_m; \boldsymbol{\alpha}) + N_{\mathcal{W}}(w_m) \ \forall m, \qquad g_{m,t}(\boldsymbol{w}) \geq 0 \ \forall(m, t).$$

For closed convex $\mathcal{W}$ and differentiable $L_m(\cdot; \boldsymbol{\alpha})$, the inclusion $0 \in \nabla_w L_m(w_m; \boldsymbol{\alpha}) + N_{\mathcal{W}}(w_m)$ is necessary and sufficient for $w_m$ to be a minimizer of $L_m(\cdot; \boldsymbol{\alpha})$ over $\mathcal{W}$. Under Assumption 4.2, the minimizer is unique, so $w_m = w_m^\star(\boldsymbol{\alpha})$ for all $m$ and $\boldsymbol{w} = \boldsymbol{w}^\star(\boldsymbol{\alpha})$. Substituting into the margin constraints yields

$$g_{m,t}(\boldsymbol{w}^\star(\boldsymbol{\alpha})) \geq 0 \qquad \forall t \in \mathcal{T}, \ \forall m \neq D_{\mathrm{val}}(t),$$

so the decision map is realizable under budget $B$.

### D.2 Proof of Theorem D.1 (NP-Hardness of Exact Minimum-Cardinality DPA)

**Theorem D.1** (NP-Hardness of Exact Minimum-Cardinality DPA)**.** *Computing a minimum-cardinality decision-preserving subset is NP-hard, even under the following restrictions:*

1. *binary fixed-quality inclusion decisions $\alpha \in \{0,1\}^N$;*

2. *two models ($M = 2$) with $\mathcal{W} = \mathbb{R}^p$ and strongly convex quadratic training objectives;*

3. *linear validation functionals and $T = p$ scenarios indexed by coordinates, with fixed winner $D_{\mathrm{val}}(t) \equiv 1$ for every $t \in \{1, \ldots, p\}$.*

*Proof.* We give a polynomial-time reduction from MINIMUM SET COVER. An instance consists of a universe $U = \{1, \ldots, p\}$ and subsets $\{S_i\}_{i=1}^N$ whose union is $U$. The objective is to select a minimum-cardinality subcollection whose union covers $U$.

**Constructing the DPA instance.** We construct a binary subset design $\alpha \in \{0,1\}^N$, two models with parameter domain $\mathcal{W} = \mathbb{R}^p$, and $T = p$ scenarios indexed by $t \in \{1, \ldots, p\}$. The observed winner is fixed as

$$D_{\mathrm{val}}(t) \equiv 1 \qquad \text{for all } t.$$

The DPA objective is the retained subset size

$$|\alpha| = \sum_{i=1}^N \alpha_i.$$

We set $n_{\min} = 1$.

**Training objectives.** Fix $\mu > 0$ and define

$$L_1(w; \alpha) = \frac{\mu}{2} \|w\|_2^2, \qquad L_2(w; \alpha) = \frac{\mu}{2} \|w\|_2^2 - \frac{1}{N} \sum_{i=1}^N \alpha_i v_i^\top w,$$

where $v_i \in \{0,1\}^p$ encodes membership in $S_i$:

$$(v_i)_t = \mathbb{1}\{t \in S_i\}.$$

Both objectives are $\mu$-strongly convex quadratics.

**Closed-form minimizers.** The unique minimizers are

$$w_1^\star(\alpha) = 0$$

and

$$w_2^\star(\alpha) = \frac{1}{\mu N} \sum_{i=1}^N \alpha_i v_i.$$

Consequently,

$$(w_2^\star(\alpha))_t = \frac{1}{\mu N} \sum_{i : t \in S_i} \alpha_i.$$

**Validation functionals and margins.** For each scenario $t$, define

$$F_{1,t}(w) = 0, \qquad F_{2,t}(w) = w_t,$$

and set

$$\gamma_t = \frac{1}{\mu N}.$$

Because model 1 is the prescribed winner, decision preservation requires

$$g_{2,t}(\boldsymbol{w}^\star(\alpha)) = F_{2,t}(w_2^\star(\alpha)) - F_{1,t}(w_1^\star(\alpha)) - \gamma_t$$
$$= (w_2^\star(\alpha))_t - \frac{1}{\mu N} \geq 0.$$

This is equivalent to

$$\sum_{i:\, t \in S_i} \alpha_i \geq 1.$$

**Equivalence to set cover.** All decision-margin constraints hold if and only if every element $t \in U$ belongs to at least one selected set $S_i$ with $\alpha_i = 1$. Thus, feasible DPA subsets correspond exactly to set covers.

Moreover,

$$|\alpha| = \sum_{i=1}^{N} \alpha_i$$

is exactly the number of selected sets. Therefore, a minimum-cardinality decision-preserving subset corresponds to a minimum set cover.

**Polynomial size.** The construction uses $N$ membership vectors, two quadratic objectives, and $p$ linear validation functionals, and is computable in time polynomial in the Set Cover instance size.

Hence, computing a minimum-cardinality decision-preserving subset is NP-hard. Equivalently, deciding whether there exists a decision-preserving subset of size at most $L$ is NP-hard. □

# E   Additional details for Algorithm 1

This appendix records options referenced by Algorithm 1 and Section 4. All correctness statements concern accepted designs and follow from the acceptance checks.

## E.1   Violation functionals for decision feasibility

**Definition E.1** (Pipeline violation functional). Let $g_{m,t}(\boldsymbol{w}^\star(\alpha))$ denote the (pipeline) margin for competitor $m$ under scenario $t$, so decision feasibility means $g_{m,t}(\boldsymbol{w}^\star(\alpha)) \geq 0$ for all relevant $(m,t)$. Define the positive-part operator $[x]_+ := \max\{x, 0\}$.

**Fixed-seed violation.** Under the fixed-seed convention, define

$$V_{\mathrm{pipe}}^{\mathrm{fix}}(\alpha) := \sum_{t \in \mathcal{T}} \sum_{m \neq D_{\mathrm{val}}(t)} \left[-g_{m,t}(\boldsymbol{w}^\star(\alpha))\right]_+.$$

**Aggregated violation.** Under the aggregated convention, let $\bar{g}_{m,t}(\alpha)$ denote the aggregated margin (Definition C.2) and define

$$V_{\mathrm{pipe}}^{\mathrm{agg}}(\alpha) := \sum_{t \in \mathcal{T}} \sum_{m \neq D_{\mathrm{val}}(t)} \left[-\bar{g}_{m,t}(\alpha)\right]_+.$$

In either case, we write $V_{\mathrm{pipe}}(\alpha)$ when the convention is clear from context.

## E.2   Proxy margins under the aggregated convention

When the decision convention aggregates over randomness (Definition C.2), the verified constraints depend on $\bar{g}_{m,t}(\alpha)$ and can be expensive to evaluate. For quasi-active set construction and local models, one may use a proxy $\tilde{g}_{m,t}^{(s)}$ that is differentiable and cheaper than $\bar{g}_{m,t}$. Examples include using a smaller seed batch, replacing the aggregator by a smooth approximation, or using an ERM-based surrogate that correlates with pipeline scores. Only the acceptance check uses $\bar{g}_{m,t}$.

### E.3 Optional merit function and sufficient decrease

If one wants a scalar progress test in addition to feasibility, define a pipeline violation $V_{\mathrm{pipe}}(\alpha)$ as in Definition E.1 (fixed-seed or aggregated). Given a penalty weight $\lambda_{\mathrm{pen}} \geq 0$, define

$$\mathcal{M}(\alpha) := |\alpha| + \lambda_{\mathrm{pen}}\, V_{\mathrm{pipe}}(\alpha).$$

A sufficient decrease rule accepts $\alpha^b$ only if it is verified decision-feasible and

$$\mathcal{M}(\alpha^b) \leq \mathcal{M}(\alpha^{(s)}) - \kappa\, r_{\mathrm{tr},s},$$

where $r_{\mathrm{tr},s}$ is the trust-region radius at iteration $s$ (Algorithm 1), and $\kappa \geq 0$ is a chosen constant.

## F  Scenario Construction Details

For each dataset, we define validation scenarios before acquisition. Table 9 summarizes how the validation scenarios are instantiated in each modality. For tabular datasets, we use named slices derived from semantically meaningful covariates in the original data. Adult uses sex, age, and education slices; Credit uses sex, age, credit-limit, and repayment-history slices; and Bank uses housing-loan status, age, and previous-contact slices. These scenarios test whether the selected training evidence preserves the same model-selection conclusion across interpretable subpopulations rather than only on the aggregate validation set. For vision datasets, scenarios are constructed from validation labels because the cached representation contains frozen image features rather than additional metadata. We use classwise scenarios for CIFAR10 and STL10, and fixed class groups for larger-label datasets such as CIFAR100 and Oxford Pets so that each scenario has adequate validation support. For text datasets, scenarios correspond to topic labels: AG News uses its four topic classes, while 20 Newsgroups uses one scenario per newsgroup class in the main experiment. Across all datasets, the scenarios are fixed before acquisition and are never optimized; acquisition changes only the training subset used to retrain the candidate convex ERM models.

**Vision class-group dictionary.**  For vision datasets with $T = 10$ scenarios, `class_group_i` denotes the $i$th block obtained by splitting the sorted class labels into ten groups. The resulting mappings are:

**CIFAR10.** 0: airplane; 1: automobile; 2: bird; 3: cat; 4: deer; 5: dog; 6: frog; 7: horse; 8: ship; 9: truck.

**STL10.** 0: airplane; 1: bird; 2: car; 3: cat; 4: deer; 5: dog; 6: horse; 7: monkey; 8: ship; 9: truck.

**CIFAR100.** 0: apple, aquarium fish, baby, bear, beaver, bed, bee, beetle, bicycle, bottle; 1: bowl, boy, bridge, bus, butterfly, camel, can, castle, caterpillar, cattle; 2: chair, chimpanzee, clock, cloud, cockroach, couch, crab, crocodile, cup, dinosaur; 3: dolphin, elephant, flatfish, forest, fox, girl, hamster, house, kangaroo, keyboard; 4: lamp, lawn mower, leopard, lion, lizard, lobster, man, maple tree, motorcycle, mountain; 5: mouse, mushroom, oak tree, orange, orchid, otter, palm tree, pear, pickup truck, pine tree; 6: plain, plate, poppy, porcupine, possum, rabbit, raccoon, ray, road, rocket; 7: rose, sea, seal, shark, shrew, skunk, skyscraper, snail, snake, spider; 8: squirrel, streetcar, sunflower, sweet pepper, table, tank, telephone, television, tiger, tractor; 9: train, trout, tulip, turtle, wardrobe, whale, willow tree, wolf, woman, worm.

**Oxford Pets.** 0: Abyssinian, Bengal, Birman, Bombay; 1: British Shorthair, Egyptian Mau, Maine Coon, Persian; 2: Ragdoll, Russian Blue, Siamese, Sphynx; 3: American Bulldog, American Pit Bull Terrier, Basset Hound, Beagle; 4: Boxer, Chihuahua, English Cocker Spaniel, English Setter; 5: German Shorthaired, Great Pyrenees, Havanese, Japanese Chin; 6: Keeshond, Leonberger, Miniature Pinscher, Newfoundland; 7: Pomeranian, Pug, Saint Bernard; 8: Samoyed, Scottish Terrier, Shiba Inu; 9: Staffordshire Bull Terrier, Wheaten Terrier, Yorkshire Terrier.

Table 9: Scenario construction for the convex ERM acquisition experiment. Each scenario is a fixed validation subset; acquisition changes only the training data. For the scenario-specification audit, inverse design is run on the search set $S$, and the omitted held-out set $H$ is used only for audit before expansion.

| Task | Frozen representation | Dataset | $T$ | Search scenarios $S$ | Held-out scenarios $H$ |
|---|---|---|---|---|---|
| Tabular | One-hot + standardized numeric | Adult | 6 | `female`, `male`, `education_num_le_9`, `education_num_gt_12` | `age_lt_35`, `age_ge_50` |
| | | Credit | 6 | `sex_female`, `sex_male`, `limit_bal_low`, `recent_repayment_delay` | `age_lt_35`, `age_ge_50` |
| | | Bank | 5 | `housing_yes`, `housing_no`, `previous_gt_0` | `age_lt_35`, `age_ge_55` |
| Vision | ResNet50 | CIFAR10 | 10 | `class_group_2`, `class_group_3`, `class_group_4`, `class_group_5`, `class_group_6`, `class_group_8`, `class_group_9` | `class_group_0`, `class_group_1`, `class_group_7` |
| | | CIFAR100 | 10 | `class_group_1`, `class_group_3`, `class_group_4`, `class_group_5`, `class_group_6`, `class_group_8`, `class_group_9` | `class_group_0`, `class_group_2`, `class_group_7` |
| | | STL10 | 10 | `class_group_1`, `class_group_3`, `class_group_4`, `class_group_5`, `class_group_6`, `class_group_8`, `class_group_9` | `class_group_0`, `class_group_2`, `class_group_7` |
| | | Oxford Pets | 10 | `class_group_0`, `class_group_1`, `class_group_2`, `class_group_3`, `class_group_5`, `class_group_6`, `class_group_9` | `class_group_4`, `class_group_7`, `class_group_8` |
| Text | TF–IDF + SVD | AG News | 4 | `Business`, `Sports`, `Sci/Tech` | `World` |
| | | 20Newsgroups | 6 | `rec`, `sci`, `talk`, `religion`, `forsale` | `comp` |

# G   Verified Search in Direct Training Acquisition

## G.1   Verified local search trajectories

**Additional verified search trajectories.**  Figure 3 reports VERDICT trajectories for six additional datasets. The same pattern observed in the main text persists across modalities: the surrogate-predicted margins are usually positive and often track the verified margins along accepted updates, but rejected proposals can have positive surrogate margins while their retraining-verified margins become negative. This confirms that the local decision-response model is useful for generating candidate deletions, but does not itself certify decision preservation. The trajectories also show that VERDICT's gains are not due to a single search behavior. CIFAR10 and Credit exhibit gradual verified compression with intermittent rejected proposals, indicating that the algorithm can continue reducing the subset after local surrogate failures. CIFAR100 and STL10 show much sharper compression, reaching very small verified fractions. In contrast, Adult and AG News show flatter trajectories after moderate compression, suggesting regimes where additional deletions are frequently rejected or where the decision map is already partly governed by coarse balance. Overall, the appendix trajectories support the same conclusion as the main figure: VERDICT converts an imperfect local response model into a safe acquisition procedure by accepting only retraining-verified proposals.

Figure 3: Verified local search trajectories for VERDICT for the remaining 6 datasets.

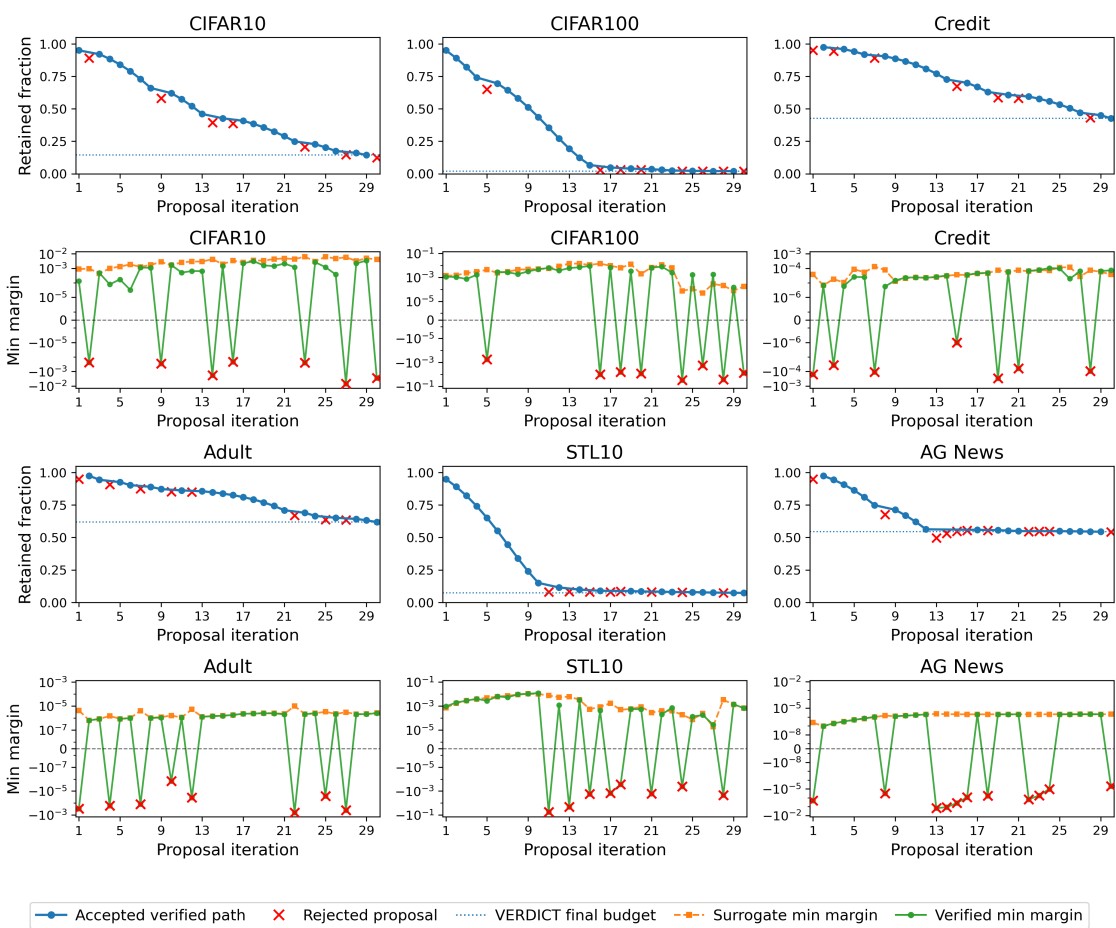

## G.2   Ablation of Margin Sensitivity

We sweep the margin multiplier $\eta$ for VERDICT, varying the constraint from winner preservation only ($\eta = 0$) to preservation of a larger fraction of the full-data winner gap. Figure 4 reports the smallest retraining-verified

retained fraction found under each setting. Several datasets, such as Adult, Bank, and Credit, show the expected increase in retained fraction as $\eta$ grows. Others, such as AG News and STL10, are relatively flat, suggesting that the selected subsets preserve the decision map with slack across multiple margin levels. Some datasets exhibit non-monotone found costs. This non-monotonicity is a property of the finite-budget search: increasing $\eta$ makes the decision-preservation constraints stricter and can only remove feasible subsets. However, VERDICT does not enumerate all feasible subsets; it follows a local proposal trajectory. Changing $\eta$ changes which constraints are active, which deletions are proposed, and which candidates pass verification, so the retained fraction found by the search need not vary monotonically.

Figure 4: Sensitivity of VERDICT to the preservation margin with $\eta \in \{0, 0.25, 0.5, 0.75\}$

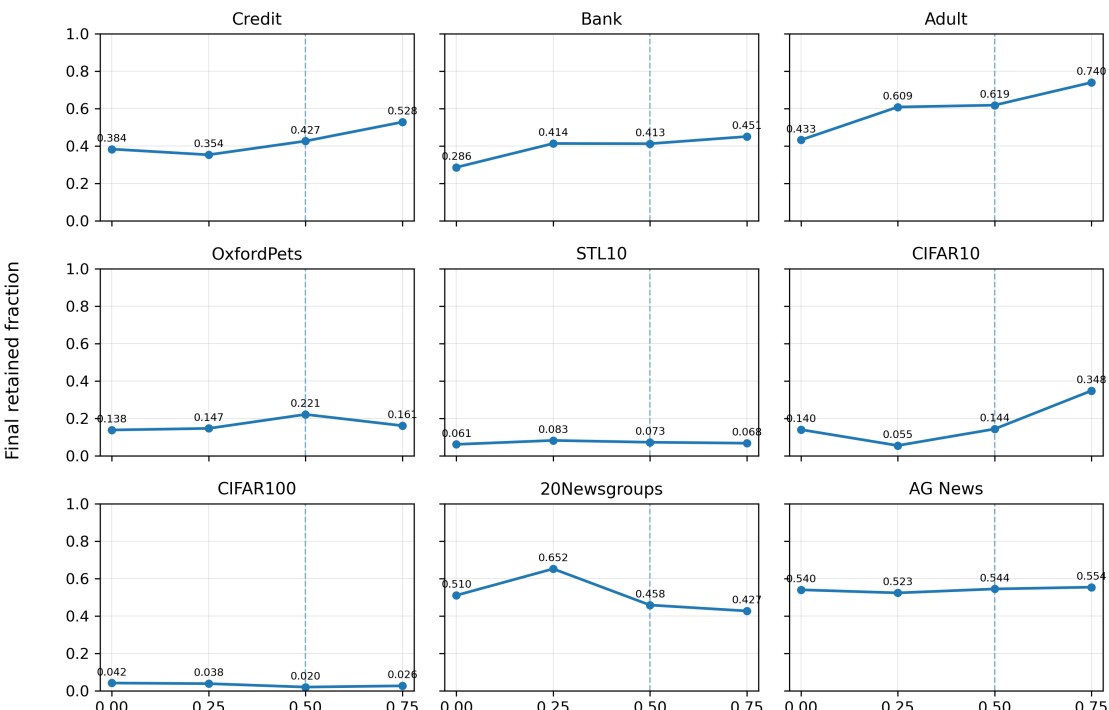

Table 10: Margin-sensitivity diagnostics for VERDICT. A/R reports accepted/rejected proposals; margin is the final verified minimum margin. All returned subsets are retraining-verified; small positive margins indicate that the search often stops near the feasibility boundary.

| Dataset | $\eta = 0$ | | $\eta = 0.25$ | | $\eta = 0.5$ | | $\eta = 0.75$ | |
|---|---|---|---|---|---|---|---|---|
| | A/R | Margin | A/R | Margin | A/R | Margin | A/R | Margin |
| 20Newsgroups | 22/8 | $1.10 \times 10^{-6}$ | 22/8 | $7.44 \times 10^{-7}$ | 22/8 | $6.58 \times 10^{-7}$ | 22/8 | $8.65 \times 10^{-8}$ |
| Adult | 21/9 | $8.11 \times 10^{-6}$ | 22/8 | $4.66 \times 10^{-6}$ | 22/8 | $2.54 \times 10^{-6}$ | 22/8 | $4.61 \times 10^{-6}$ |
| AG News | 20/10 | $2.21 \times 10^{-6}$ | 19/11 | $2.25 \times 10^{-6}$ | 19/11 | $2.09 \times 10^{-6}$ | 17/12 | $2.01 \times 10^{-6}$ |
| Bank | 21/9 | $9.72 \times 10^{-7}$ | 21/9 | $1.99 \times 10^{-7}$ | 21/9 | $2.50 \times 10^{-8}$ | 20/10 | $2.50 \times 10^{-7}$ |
| CIFAR10 | 24/6 | $6.19 \times 10^{-3}$ | 23/7 | $6.07 \times 10^{-4}$ | 23/7 | $3.43 \times 10^{-3}$ | 23/7 | $6.48 \times 10^{-4}$ |
| CIFAR100 | 24/6 | $1.00 \times 10^{-2}$ | 22/8 | $9.41 \times 10^{-5}$ | 22/8 | $1.44 \times 10^{-4}$ | 23/7 | $1.54 \times 10^{-3}$ |
| Credit | 23/7 | $7.83 \times 10^{-5}$ | 23/7 | $4.14 \times 10^{-5}$ | 23/7 | $7.27 \times 10^{-5}$ | 23/7 | $8.44 \times 10^{-6}$ |
| OxfordPets | 21/9 | $1.57 \times 10^{-4}$ | 21/9 | $2.18 \times 10^{-4}$ | 21/9 | $4.96 \times 10^{-4}$ | 16/11 | $1.12 \times 10^{-4}$ |
| STL10 | 20/10 | $3.56 \times 10^{-4}$ | 21/9 | $3.93 \times 10^{-5}$ | 22/8 | $6.70 \times 10^{-4}$ | 18/12 | $3.32 \times 10^{-4}$ |

### G.3 Ablation of VERDICT components

We further ablate the main components of VERDICT in the direct subset-acquisition setting on six datasets, two for each task modality. These variants are designed to isolate the roles of the three main ingredients of the method: verification by the original forward protocol, response-guided proposal construction, and local control of the surrogate approximation. **VERDICT-NoVerify** removes the retrain-and-check acceptance rule and follows the local surrogate path without using verified forward-execution feedback during search. This shows that surrogate feasibility alone is not sufficient for decision preservation. **VERDICT-RandomWS** keeps the same surrogate, trust-region, and verification machinery, but replaces response-guided working-set construction with uniformly random working sets of comparable size; this tests whether the decision-response signal is useful for identifying candidate examples to modify. **VERDICT-FixedTR** keeps the decision-response model and verification step, but disables adaptive trust-region updates, using a fixed radius throughout the search; this isolates the effect of adapting the local step size when managing approximation error. **VERDICT-AllConstraints** replaces the quasi-active constraint set with the full set of scenario-model margin constraints in the local surrogate, while retaining the VERDICT proposal and verification loop.

Table 11: Component ablation for direct decision-preserving subset acquisition. Each variant modifies one part of the VERDICT search procedure while keeping the same target winner map, margin constraints, and retrain-and-check evaluation protocol.

| Task | Dataset | $n$ | VERDICT | VERDICT NoVerify | VERDICT NoDR | VERDICT RandomWS | VERDICT FixedTR | VERDICT-AllConstraints |
|------|---------|-----|---------|------------------|--------------|------------------|-----------------|------------------------|
| Tabular | Credit | 30000 | **0.427** | 0.947 | $0.891 \pm 0.075$ | $0.767 \pm 0.019$ | 0.513 | 0.857 |
|  | Adult | 48842 | 0.619 | 0.950 | $0.736 \pm 0.097$ | $0.431 \pm 0.085$ | 0.664 | 0.777 |
| Vision | STL10 | 13000 | **0.073** | 0.080 | $0.728 \pm 0.152$ | $0.197 \pm 0.130$ | 0.217 | 0.101 |
|  | CIFAR100 | 50000 | **0.020** | 0.650 | $0.389 \pm 0.086$ | $0.310 \pm 0.111$ | 0.215 | 0.033 |
| Text | 20Newsgroups | 18846 | 0.458 | 0.950 | $0.729 \pm 0.028$ | $0.735 \pm 0.010$ | 0.464 | **0.441** |
|  | AG News | 127600 | 0.544 | 0.950 | $0.918 \pm 0.015$ | $0.897 \pm 0.044$ | 0.359 | **0.133** |

Default VERDICT achieves the best retained fraction on three of the six datasets, with especially strong reductions on Credit, STL10, and CIFAR100. The remaining cases are informative rather than contradictory: they show that, once verification is fixed, the particular local proposal model can matter in a problem-dependent way. On Adult, random working sets obtain a smaller verified subset, suggesting that response-guided working-set construction is not always the best exploration strategy. On the two text datasets, AllConstraints is strongest, indicating that the quasi-active constraint set may be too narrow when useful information is distributed across a broader set of scenario-model comparisons. Thus, Table 11 does not suggest a single universally optimal local surrogate; rather, it shows that the DPA objective can be approached by several verified proposal mechanisms, whose effectiveness depends on the structure of the scenario margins.

This interpretation is reinforced by Table 12. Accepted VERDICT outputs typically have positive but small final verified margins, indicating that the search approaches the decision-preserving boundary rather than stopping conservatively. This matches Figures 2 and 3: the surrogate enables aggressive deletion, while retrain-and-check verification prevents infeasible updates. The necessity of this step is clear from NoVerify, whose final verified margins are negative on every dataset, and from rejected VERDICT proposals, which are surrogate-feasible but become infeasible after retraining, sometimes causing margin violations or winner flips. In contrast, the remaining proposal variants exhibit dataset-dependent behavior, showing that verification is essential for reliability, whereas the most effective surrogate configuration depends on the local margin geometry and the informativeness of the response signal.

The ablation study in Table 11 also demonstrates that alternative proposal components, most notably RandomWS and AllConstraints, occasionally achieve smaller verified subsets than the default VERDICT. To understand these behaviors, Table 13 separates final compression from the mechanisms that produce it by reporting the retained fraction, number of surrogate constraints, mean accepted deletion step, surrogate optimism gap on rejected proposals, and score-construction time. Together, these metrics indicate whether a variant succeeds because it proposes larger feasible deletions, better predicts retrained margins, uses broader constraint information, or simply incurs greater computational cost.

Table 12: Verification diagnostics across acquisition methods. All margins are reported after final verification, and the first rejected margin reports the surrogate margin followed by the verified margin after retraining. NoDR diagnostics are means over five seeds; N/A indicates that NoDR does not construct a decision-response surrogate.

| Dataset | Method | Accept / Reject | Final verified margin | First rejected margin surrogate $\rightarrow$ verified | Viol. / Flip |
|---|---|---|---|---|---|
| Credit | **VERDICT** | 23/7 | $7.27\times10^{-5}$ | $3.80\times10^{-5} \rightarrow -1.59\times10^{-4}$ | 3/3 |
| | NoVerify | 2/0 | $-8.92\times10^{-4}$ | N/A | N/A |
| | NoDR | 15.6/9.8 | $3.86\times10^{-6}$ | N/A $\rightarrow -2.27\times10^{-4}$ | 4.6/2.4 |
| | RandomWS | 21.3/8.7 | $9.83\times10^{-7}$ | $5.40\times10^{-6} \rightarrow -1.13\times10^{-5}$ | 2.3/0.7 |
| | FixedTR | 13/17 | $1.18\times10^{-5}$ | $3.80\times10^{-5} \rightarrow -1.59\times10^{-4}$ | 3/3 |
| | AllConstraints | 21/9 | $1.38\times10^{-7}$ | $1.15\times10^{-6} \rightarrow -1.20\times10^{-4}$ | 3/2 |
| Adult | **VERDICT** | 22/8 | $2.54\times10^{-6}$ | $4.30\times10^{-6} \rightarrow -3.02\times10^{-4}$ | 2/2 |
| | NoVerify | 1/0 | $-3.02\times10^{-4}$ | N/A | N/A |
| | NoDR | 18.2/10.0 | $1.33\times10^{-6}$ | N/A $\rightarrow -1.55\times10^{-5}$ | 1.2/0.8 |
| | RandomWS | 22.3/7.7 | $4.74\times10^{-6}$ | $2.83\times10^{-7} \rightarrow -2.78\times10^{-6}$ | 1/0.3 |
| | FixedTR | 8/22 | $4.13\times10^{-7}$ | $4.30\times10^{-6} \rightarrow -3.02\times10^{-4}$ | 2/2 |
| | AllConstraints | 22/8 | $2.88\times10^{-6}$ | $1.62\times10^{-6} \rightarrow -6.56\times10^{-6}$ | 1/0 |
| STL10 | **VERDICT** | 22/8 | $6.70\times10^{-4}$ | $7.60\times10^{-3} \rightarrow -5.57\times10^{-2}$ | 5/1 |
| | NoVerify | 11/0 | $-5.57\times10^{-2}$ | N/A | N/A |
| | NoDR | 18.4/9.4 | $1.43\times10^{-4}$ | N/A $\rightarrow -5.13\times10^{-4}$ | 1.0/0.6 |
| | RandomWS | 21.5/8.5 | $1.13\times10^{-4}$ | $1.11\times10^{-4} \rightarrow -1.98\times10^{-4}$ | 1/0.2 |
| | FixedTR | 30/0 | $1.32\times10^{-2}$ | N/A | N/A |
| | AllConstraints | 20/10 | $1.55\times10^{-4}$ | $3.06\times10^{-5} \rightarrow -1.82\times10^{-4}$ | 1/1 |
| CIFAR100 | **VERDICT** | 22/8 | $1.44\times10^{-4}$ | $4.33\times10^{-3} \rightarrow -5.42\times10^{-4}$ | 1/0 |
| | NoVerify | 5/0 | $-5.42\times10^{-4}$ | N/A | N/A |
| | NoDR | 21.4/8.6 | $1.16\times10^{-4}$ | N/A $\rightarrow -3.02\times10^{-4}$ | 1.0/0.0 |
| | RandomWS | 22/8 | $1.39\times10^{-4}$ | $1.18\times10^{-5} \rightarrow -1.31\times10^{-4}$ | 1/0 |
| | FixedTR | 30/0 | $1.45\times10^{-2}$ | N/A | N/A |
| | AllConstraints | 22/8 | $7.20\times10^{-4}$ | $2.15\times10^{-5} \rightarrow -2.01\times10^{-3}$ | 1/1 |
| 20Newsgroups | **VERDICT** | 22/8 | $6.58\times10^{-7}$ | $1.71\times10^{-7} \rightarrow -5.79\times10^{-4}$ | 1/1 |
| | NoVerify | 1/0 | $-5.79\times10^{-4}$ | N/A | N/A |
| | NoDR | 16.4/10.4 | $2.92\times10^{-9}$ | N/A $\rightarrow -6.42\times10^{-5}$ | 1.4/0.2 |
| | RandomWS | 13/9.3 | $3.26\times10^{-9}$ | $4.82\times10^{-8} \rightarrow -8.25\times10^{-5}$ | 1.3/0.3 |
| | FixedTR | 15/15 | $3.82\times10^{-7}$ | $1.71\times10^{-7} \rightarrow -5.79\times10^{-4}$ | 1/1 |
| | AllConstraints | 19/11 | $1.33\times10^{-9}$ | $1.11\times10^{-7} \rightarrow -2.37\times10^{-6}$ | 1/0 |
| AG News | **VERDICT** | 19/11 | $2.09\times10^{-6}$ | $2.36\times10^{-7} \rightarrow -1.99\times10^{-4}$ | 1/1 |
| | NoVerify | 1/0 | $-1.99\times10^{-4}$ | N/A | N/A |
| | NoDR | 6.8/8.2 | $1.52\times10^{-8}$ | N/A $\rightarrow -8.82\times10^{-5}$ | 1.0/0.0 |
| | RandomWS | 10.7/9 | $1.52\times10^{-8}$ | $1.95\times10^{-8} \rightarrow -9.42\times10^{-5}$ | 1/0 |
| | FixedTR | 20/10 | $8.62\times10^{-6}$ | $2.36\times10^{-7} \rightarrow -1.99\times10^{-4}$ | 1/1 |
| | AllConstraints | 24/6 | $1.19\times10^{-5}$ | $1.77\times10^{-7} \rightarrow -5.74\times10^{-6}$ | 1/0 |

Essentially, RandomWS preserves VERDICT's adaptive trust region and retrain-and-check verification, but replaces response-guided ordering with random working sets. It therefore trades exploitation of the current local response signal for additional exploration and can uncover feasible deletion paths that the deterministic ordering initially underranks. AllConstraints instead retains response-guided construction but builds the surrogate from every non-winner scenario–model margin rather than only the quasi-active set. This provides broader information about the decision map, but increases scoring cost and can also restrict proposals through margins that are currently far from binding.

The diagnostics reveal three regimes. On Credit, STL10, and CIFAR100, default VERDICT achieves the smallest retained fraction using only 4–8 constraints and substantially lower scoring time; although AllConstraints is often better calibrated, its smaller or less productive accepted steps do not improve the final deletion path. On 20Newsgroups and AG News, the broader constraint set produces more effective accepted deletions and yields smaller subsets, indicating that relevant response information extends beyond the quasi-active margins. Adult exhibits a different behavior: RandomWS performs best, suggesting that exploration of alternative deletion directions matters more than broader constraint coverage. Overall, Table

13 shows that final compression depends more on the verified proposal trajectory than on local surrogate calibration alone, while supporting quasi-active filtering as an efficient default rather than a universally optimal choice.

Table 13: Effect of proposal construction and constraint scope on VERDICT. The table shows how quasi-active, random-working-set, and full-constraint proposals trade off verified compression, accepted deletion size, surrogate calibration, and scoring cost. Bold denotes the best result within each dataset.

| Dataset | Method | Retained fraction ↓ | Constraints ↓ | Accepted step (%) ↑ | Optimism gap | Score time / proposal (s) ↓ |
|---|---|---|---|---|---|---|
| Credit | VERDICT | **0.427** | **5** | **3.62** | $1.72 \times 10^{-4}$ | **0.12** |
| | RandomWS | $0.767 \pm 0.019$ | 5 | 1.23 | $\mathbf{1.18 \times 10^{-5}}$ | 0.12 |
| | AllConstraints | 0.857 | 48 | 0.73 | $1.99 \times 10^{-5}$ | 1.01 |
| Adult | VERDICT | 0.619 | **5** | 2.16 | $2.11 \times 10^{-4}$ | **0.16** |
| | RandomWS | $\mathbf{0.431 \pm 0.085}$ | 5 | **3.73** | $2.93 \times 10^{-5}$ | 0.16 |
| | AllConstraints | 0.777 | 48 | 1.13 | $\mathbf{6.89 \times 10^{-6}}$ | 1.43 |
| STL10 | VERDICT | **0.073** | 8 | **10.61** | $1.31 \times 10^{-2}$ | **0.96** |
| | RandomWS | $0.197 \pm 0.130$ | 8 | 4.67 | $\mathbf{1.87 \times 10^{-4}}$ | 2.06 |
| | AllConstraints | 0.101 | 80 | 10.31 | $2.85 \times 10^{-3}$ | 9.41 |
| CIFAR100 | VERDICT | **0.020** | 8 | **15.54** | $1.48 \times 10^{-2}$ | **67.92** |
| | RandomWS | $0.310 \pm 0.111$ | 8 | 5.79 | $\mathbf{6.27 \times 10^{-4}}$ | 130.73 |
| | AllConstraints | 0.033 | 80 | 13.53 | $1.64 \times 10^{-3}$ | 618.48 |
| 20Newsgroups | VERDICT | 0.458 | 5 | 3.48 | $2.55 \times 10^{-4}$ | **4.15** |
| | RandomWS | $0.735 \pm 0.010$ | 5 | 2.31 | $2.37 \times 10^{-5}$ | 4.53 |
| | AllConstraints | **0.441** | 48 | **4.14** | $\mathbf{5.39 \times 10^{-6}}$ | 34.43 |
| AG News | VERDICT | 0.544 | 4 | 3.11 | $3.35 \times 10^{-4}$ | **5.76** |
| | RandomWS | $0.897 \pm 0.044$ | 4 | 1.01 | $\mathbf{3.29 \times 10^{-5}}$ | 7.72 |
| | AllConstraints | **0.133** | 32 | **7.79** | $3.68 \times 10^{-4}$ | 37.36 |

## G.4 Loss-space baseline results

Table 14 reports the smallest verified retained training fraction found on the evaluated budget grid for three static loss-space baselines adapted to direct training acquisition. Each training example is represented by its centered vector of losses under the full-data reference candidates; loss-$k$-center and approximate loss-$k$-medoids select subsets by covering or representing this fixed geometry, while FW-Mean greedily matches its full-pool mean with an equal-weight binary subset. All candidate models are then retrained from scratch on the selected subset and evaluated using the same scenario-wise winner and margin verification as VERDICT. Results are reported as mean $\pm$ standard deviation over five seeds; $> 0.950$ indicates that no seed was feasible within the evaluated grid, while $\geq$ indicates that at least one seed remained infeasible and was conservatively assigned a retained fraction of 1.0 in the reported summary. VERDICT achieves a smaller verified retained fraction than all three static loss-space baselines on every dataset, indicating that preserving the geometry or mean of full-reference losses is generally insufficient to preserve the model-selection decision after retraining.

# H Downstream Applications

## H.1 Decision-critical scenario discovery

**Setup and audit procedure.** VERDICT verifies decision preservation with respect to the declared scenario set. This protocol-relative nature suggests a natural auditing use case: if a practitioner is unsure whether a scenario family should be included in the declared scenario set, we can temporarily omit it, run VERDICT on the remaining scenarios, and test whether the selected subset also preserves the omitted family. Importantly, this audit does not change the candidate models, training routine, validation functionals, or decision convention. It varies only the scenario specification used during subset selection.

For each dataset, we construct the audit by holding out either semantically meaningful scenario families or automatically generated scenario groups, while keeping the candidate models, training routine, validation

Table 14: Smallest verified retained training fraction found on the evaluated budget grid for static loss-space baselines. Lower is better. Results are reported as mean $\pm$ standard deviation where applicable.

| Dataset | $n_{\text{train}}$ | Loss-$k$-Center | Loss-$k$-Medoids | FW-Mean | VERDICT |
|---|---|---|---|---|---|
| Credit | 30,000 | $0.930 \pm 0.057$ | $> 0.950$ | $> 0.950$ | **0.427** |
| Bank | 45,211 | $0.540 \pm 0.055$ | $0.560 \pm 0.108$ | $0.780 \pm 0.091$ | **0.413** |
| Adult | 48,842 | $> 0.950$ | $> 0.950$ | $0.850 \pm 0.187$ | **0.619** |
| OxfordPets | 7,349 | $> 0.950$ | $> 0.950$ | $> 0.950$ | **0.221** |
| STL10 | 13,000 | $> 0.950$ | $> 0.950$ | $0.910 \pm 0.022$ | **0.073** |
| CIFAR10 | 50,000 | $> 0.950$ | $0.900 \pm 0.050$ | $0.830 \pm 0.115$ | **0.144** |
| CIFAR100 | 50,000 | $0.520 \pm 0.091$ | $0.460 \pm 0.089$ | $0.530 \pm 0.104$ | **0.020** |
| 20Newsgroups | 18,846 | $0.630 \pm 0.027$ | $0.620 \pm 0.027$ | $0.800 \pm 0.035$ | **0.458** |
| AG News | 127,600 | $> 0.950$ | $0.950 \pm 0.000$ | $0.950 \pm 0.000$ | **0.544** |

losses, and decision convention fixed. For tabular datasets, the omitted scenario family is specified by interpretable feature slices: age slices for Adult and Bank. For text datasets, we perform leave-one-family audits: AG News holds out one label class at a time, and 20Newsgroups holds out one topic family at a time. For vision datasets, scenarios are automatically constructed as class groups from validation labels. We evaluate five random held-out class-group splits, each using seven class groups as $T_{\text{search}}$ and three as $T_{\text{heldout}}$, then use the strongest failing split for the expanded-scenario experiment.

Let $T_{\text{search}}$ denote the scenarios declared to VERDICT during subset selection, and let $T_{\text{heldout}}$ denote omitted scenarios used only for audit. The audit has three stages. First, VERDICT selects a subset using only $T_{\text{search}}$, and the selected subset is verified to preserve the original winners and margins on those search scenarios. Second, we retrain the candidate models on the selected subset and evaluate the omitted scenarios $T_{\text{heldout}}$. A held-out winner flip or margin violation indicates that the omitted family contains decision-critical constraints not implied by the declared search scenarios. Third, we form an expanded scenario set $T_{\text{expanded}} = T_{\text{search}} \cup T_{\text{heldout}}$, rerun VERDICT on this expanded scenario specification, and measure the additional retained evidence required to preserve the enlarged decision map.

We report metrics corresponding to the three stages of the audit. The held-out audit columns measure whether the subset selected from the search scenario set transfers to omitted scenarios: winner flips count scenarios where the selected winner changes, margin violations count failed winner-versus-competitor constraints, and $\Delta_{\min}^{\text{heldout}}$ records the worst held-out margin. The retained fractions $\rho_{\text{search}}$ and $\rho_{\text{expanded}}$ measure the evidence cost before and after adding the omitted scenarios back, while $\Delta_{\min}^{\text{expanded}}$ verifies whether the expanded scenario set is preserved after rerunning VERDICT. Finally, baseline feasible counts summarize how often random, stratified, KCenter, or DecisionIF subsets at VERDICT's expanded-scenario retained size also satisfy all expanded-scenario constraints. Together, these metrics distinguish three effects: whether an omission is decision-critical, how costly it is to cover, and whether success is specific to VERDICT's selected evidence rather than retained size alone.

**Experimental results.** Table 15 shows that held-out failures can diagnose incomplete scenario specifications. In five out of six datasets, VERDICT preserves the search scenario set, but the selected subset fails on omitted scenarios through winner flips or negative held-out margins. These failures indicate that the omitted scenarios impose decision constraints not implied by the search scenarios alone. Adding the omitted scenarios back makes the diagnostic actionable. VERDICT then finds feasible subsets with nonnegative margins on the expanded scenario set, and the increase from $\rho_{\text{search}}$ to $\rho_{\text{expanded}}$ measures the evidence cost of covering the omitted scenarios. This cost varies substantially: Adult and AG News require large increases, OxfordPets and 20Newsgroups require moderate increases, and STL10 requires only a small increase, suggesting that decision-critical scenarios may differ in how much additional evidence they require.

We compare random, stratified sampling, KCenter, and DecisionIF subsets at the same retained size selected by VERDICT for $T_{\text{expanded}}$. Table 16 shows that these gains are not explained merely by retaining more data. For the five positive discovery cases, random, stratified sampling, K-center, and DecisionIF subsets at

Table 15: Scenario-specification audits across three modalities. $\rho_{\text{search}}$ is the retained fraction selected using only the search scenario set; $\rho_{\text{expanded}}$ is the retained fraction after adding the omitted family back. $\Delta_{\min}^{\text{heldout}}$ is the minimum held-out decision margin, so negative values indicate margin violations.

| Task | Dataset | Omitted family | Held-out audit | | | Expanded protocol | | | |
|------|---------|----------------|-------|------|---------------------------|-------------------|-------------------|---------------------------|------------------------|
| | | | Flips | Viol. | $\Delta_{\min}^{\text{heldout}}$ | $\rho_{\text{search}}$ | $\rho_{\text{expanded}}$ | $\Delta_{\min}^{\text{expanded}}$ | Best baseline feasible |
| Tabular | Adult | age slices | 2 | 5 | $-2.15\times10^{-3}$ | 0.378 | 0.619 | $2.54\times10^{-6}$ | 0/20 |
| | Bank | age slices | 0 | 0 | $5.63\times10^{-5}$ | 0.397 | 0.413 | $2.50\times10^{-8}$ | 6/20 |
| | Credit | age slices | 1 | 1 | $-7.41\times10^{-5}$ | 0.427 | 0.427 | $7.27\times10^{-5}$ | 0/20 |
| Vision | OxfordPets | failure splits | 1 | 2 | $-9.51\times10^{-2}$ | 0.116 | 0.221 | $4.96\times10^{-4}$ | 0/20 |
| | STL10 | failure splits | 3 | 9 | $-5.73\times10^{-2}$ | 0.061 | 0.073 | $6.70\times10^{-4}$ | 0/20 |
| | CIFAR10 | failure splits | 3 | 3 | $-5.76\times10^{-2}$ | 0.246 | 0.144 | $3.43\times10^{-3}$ | 0/20 |
| | CIFAR100 | failure splits | 3 | 12 | $-5.86\times10^{-2}$ | 0.023 | 0.020 | $1.44\times10^{-4}$ | 0/20 |
| Text | AG News | world news | 1 | 5 | $-8.18\times10^{-2}$ | 0.070 | 0.544 | $2.09\times10^{-6}$ | 0/20 |
| | 20Newsgroups | computing | 1 | 1 | $-2.62\times10^{-3}$ | 0.411 | 0.458 | $6.58\times10^{-7}$ | 0/20 |

Table 16: Baseline comparison across datasets. Feasible rate reports the fraction of feasible runs. Avg winner agreement, Avg violation rate, Avg min margin, and Avg winner flips are averaged across runs.

| Dataset | Method | Feasible rate | Avg winner agreement | Avg violation rate | Avg min margin | Avg winner flips |
|---------|--------|---------------|----------------------|--------------------|----------------|------------------|
| | VERDICT | **1/1** | **1.00** | **0.000** | **$2.54\times10^{-6}$** | **0.0** |
| | Random | 0/5 | 0.60 | 0.079 | $-2.19\times10^{-4}$ | 2.4 |
| Adult | Stratified | 0/5 | 0.567 | 0.113 | $-2.82\times10^{-4}$ | 2.6 |
| | KCenter | 0/5 | 0.633 | 0.088 | $-3.71\times10^{-4}$ | 2.2 |
| | DecisionIF | 0/5 | 0.333 | 0.167 | $-2.55\times10^{-3}$ | 4.0 |
| | VERDICT | **1/1** | **1.00** | **0.000** | **$2.50\times10^{-8}$** | **0.0** |
| | Random | 4/5 | 1.00 | 0.005 | $-8.11\times10^{-7}$ | 0.0 |
| Bank | Stratified | 2/5 | 0.88 | 0.050 | $-2.24\times10^{-5}$ | 0.6 |
| | KCenter | 0/5 | 0.48 | 0.235 | $-1.64\times10^{-4}$ | 2.6 |
| | DecisionIF | 0/5 | 0.00 | 0.575 | $-7.12\times10^{-2}$ | 5.0 |
| | VERDICT | **1/1** | **1.00** | **0.000** | **$2.09\times10^{-6}$** | **0.0** |
| | Random | 0/5 | 0.65 | 0.194 | $-2.35\times10^{-3}$ | 1.4 |
| AG News | Stratified | 0/5 | 0.75 | 0.156 | $-2.34\times10^{-3}$ | 1.0 |
| | KCenter | 0/5 | 0.75 | 0.156 | $-2.54\times10^{-3}$ | 1.0 |
| | DecisionIF | 0/5 | 0.75 | 0.031 | $-3.94\times10^{-3}$ | 1.0 |
| | VERDICT | **1/1** | **1.00** | **0.000** | **$6.58\times10^{-7}$** | **0.0** |
| | Random | 0/5 | 0.233 | 0.221 | $-1.93\times10^{-3}$ | 4.6 |
| 20Newsgroups | Stratified | 0/5 | 0.333 | 0.208 | $-1.75\times10^{-3}$ | 4.0 |
| | KCenter | 0/5 | 0.500 | 0.188 | $-1.13\times10^{-3}$ | 3.0 |
| | DecisionIF | 0/5 | 0.167 | 0.229 | $-6.01\times10^{-2}$ | 5.0 |
| | VERDICT | **1/1** | **1.00** | **0.000** | **$4.96\times10^{-4}$** | **0.0** |
| | Random | 0/5 | 0.80 | 0.0375 | $-4.87\times10^{-2}$ | 2.0 |
| OxfordPets | Stratified | 0/5 | 0.80 | 0.0300 | $-4.21\times10^{-2}$ | 2.0 |
| | KCenter | 0/5 | 0.80 | 0.0275 | $-5.27\times10^{-2}$ | 2.0 |
| | DecisionIF | 0/5 | 0.70 | 0.0375 | $-5.34\times10^{-2}$ | 3.0 |
| | VERDICT | **1/1** | **1.00** | **0.000** | **$6.70\times10^{-4}$** | **0.0** |
| | Random | 0/5 | 0.62 | 0.200 | $-7.88\times10^{-2}$ | 3.8 |
| STL10 | Stratified | 0/5 | 0.64 | 0.185 | $-6.42\times10^{-2}$ | 3.6 |
| | KCenter | 0/5 | 0.82 | 0.100 | $-8.54\times10^{-2}$ | 1.8 |
| | DecisionIF | 0/5 | 0.80 | 0.075 | $-2.85\times10^{-2}$ | 2.0 |

VERDICT's expanded-scenario retained size all fail to preserve the expanded scenario set. This indicates that the selected evidence, not only the retained fraction, is important for decision preservation. Bank provides an important contrast. Its omitted age slices do not produce a strong scenario-specification gap, and several same-size baselines as VERDICT are feasible in several seeds. This shows that the audit is not biased toward always declaring omitted scenarios decision-critical. Instead, VERDICT distinguishes cases where omitted scenarios introduce new decision constraints from cases where the declared search scenarios already transfer.

## H.2 Evaluation Subset Selection Protocol for a Fixed Candidate Set

Model selection can be expensive even after candidate models have been trained, because each candidate must be evaluated on a potentially large validation set. In this downstream application, the acquisition variable is not the training set but the evaluation subset used to compare a fixed set of trained candidate models. The goal is to select a small subset of validation examples whose aggregate losses induce the same model-selection conclusion as the full validation set.

This setting instantiates the decision-preserving view with a different forward protocol. The candidate models are fixed, the full validation set defines the reference decision, and a subset is feasible if evaluating the candidates only on that subset preserves the relevant comparison structure. Depending on the intended use case, preservation may mean preserving the top-ranked model, preserving all pairwise orderings above a margin, or preserving the full model ranking. The selected subset can support repeated execution, reproduction, or auditing of the same declared comparison protocol. When the complete validation evidence is costly to process or access repeatedly, its one-time construction cost can be amortized over subsequent evaluations of the fixed candidate set.

We therefore report both ranking metrics and decision metrics. For a validation subset $S$, let $F_m(S)$ denote the average loss of candidate model $m$ on $S$, and let $m^\star$ be the winner under the full validation set. A low-cost evaluation subset should satisfy

$$F_m(S) - F_{m^\star}(S) \geq \gamma_m \quad \text{for all } m \neq m^\star,$$

where $\gamma_m$ is a prescribed margin derived from the full-validation gap. This is the evaluation-subset analogue of the DPA constraint used in the training-acquisition experiments. In addition to ranking correlations, we therefore evaluate whether the selected subset preserves the model-selection decision itself.

We first define the low-cost model-selection protocol. For each dataset and backbone, we train a fixed pool of candidate classifiers and compute their losses on the full validation set. The full validation set determines the reference ranking and reference winner. Each subset-selection method then chooses a small budgeted subset of validation examples. We evaluate the candidate models on this subset and compare the induced model ranking and winner against the full-validation reference.

We report the following metrics:

- **Winner agreement**: whether the subset selects the same top model as the full validation set.

- **Selection regret**: the full-validation loss difference between the model selected by the subset and the true full-validation winner.

- **Pairwise decision agreement**: fraction of candidate pairs whose ordering is preserved by the subset.

- **Margin violation rate**: the fraction of required winner-vs-competitor margins violated on the subset.

- **Minimum decision margin**: the smallest subset margin between the reference winner and any competitor.

- **Kendall $\tau$ and Spearman $\rho$**: rank-correlation metrics measuring preservation of the full model ranking.

- **Budget fraction and runtime**: the evaluation cost of the subset relative to the full validation set.

This metric suite distinguishes two goals. Ranking correlations measure whether the subset approximately preserves the full ordering of all candidate models. Decision metrics measure the downstream model-selection consequence: whether a practitioner using the subset would deploy the same model as they would using the full validation set. These metrics make the downstream application operational. A subset is useful for low-cost model selection if it either recovers the exact full-validation winner or selects a model with low full-validation regret while preserving most candidate comparisons. Thus, the reported metrics should be interpreted as model-selection fidelity metrics, not merely as generic ranking-correlation statistics.

**Backbone Robustness for Evaluation Subsets** We study evaluation subset selection in vision datasets to preserve model comparisons under a fixed protocol, in the spirit of data efficient evaluation and tailored benchmark design Vivek et al. (2024); Saranathan et al. (2025); Yuan et al. (2025). For vision datasets, we repeat the ranking preservation protocol across five frozen vision backbones: ResNet50, ResNet18, ConvNeXt-Tiny, Swin-T, and ViT-B/16. We cache frozen features for the train and validation splits, then train 20 linear classifiers with varying random seeds and hyperparameters on top of these features to generate the validation score matrix. We run subset selection at various budgets and measure the Kendall $\tau$ correlation between the subset ranking and the full dataset ranking.

Experiments cover subsets with sizes of 0.5%, 1%, 2%, and 5%. We also evaluate 10% and 20% when estimating minimum budget thresholds. We compare Random sampling, Stratified sampling, Difficulty-based selection based on hard example mining (Shrivastava et al., 2016), Disagreement-based selection, and k center clustering on cached embeddings following Sener & Savarese (2018) to VERDICT. We also include score-space baselines: loss-space k center and k medoids (as in tailored benchmark selection Yuan et al. (2025)), a Frank–Wolfe mean coreset motivated by Frank–Wolfe coreset constructions Clarkson (2010), and a pairwise hinge baseline inspired by pairwise ranking objectives Joachims (2002).

**Detailed Kendall $\tau$ breakdown across budgets (ResNet50)** Since the main text uses ResNet50 as the default backbone, Table 17 provides the full Kendall $\tau$ correlation breakdown across budgets for ResNet50 (mean$_{\pm std}$ over 5 seeds), complementing the multi-backbone summary metrics reported below. We analyze robustness using three metrics: fidelity at low budgets, stability against ranking inversions, and sample efficiency. For each dataset, the full evaluation set defines the reference ranking of a fixed $M = 20$ candidate-model pool evaluated using frozen ResNet50 representations. Each method selects a subset of evaluation examples at the indicated budget, and Kendall's $\tau$ compares the ranking induced by the subset with the full-set ranking. The entries report mean $\pm$ standard deviation over five subset-selection seeds. Table 17 uses the candidate-score matrices from the main baseline-comparison experiment and is distinct from the independently constructed ResNet50 instance in the backbone-sensitivity study of Table 22. Averages can mask failures on specific datasets. Table 18 counts the datasets where methods produce a ranking inversion (Kendall $\tau < 0$) at the 0.5% budget. A negative correlation implies the subset is misleading and performs worse than random guessing. VERDICT yields zero inversions across all backbones. Difficulty and Disagreement methods exhibit instability in this low budget regime and produce inversions in most datasets with ConvNeXt features. Table 19 summarizes the minimum budget required for VERDICT to reach a correlation of 0.9. The 0.5% budget is sufficient in 32 out of 35 cases. We identify a harder regime: CIFAR100 with ResNet50 (2%). The method resolves the ranking using a small fraction of the data even in these cases.

Detailed per-backbone tables reporting dataset metrics, budget curves, and baseline comparisons are provided in Tables 20 through 24. This experiment constructs an independent $M = 20$ candidate-model pool on frozen ResNet50 representations and evaluates how well selected evaluation subsets preserve its full-set ranking. Randomized methods are reported as mean $\pm$ standard deviation over five subset-selection seeds. [2]

---

[2]We report two distinct ResNet50 evaluation settings. Table 17 uses the ResNet50 candidate pool from the main baseline-comparison experiment. Tables 21–25 report a separate backbone-sensitivity study in which an $M = 20$ candidate pool is constructed independently for each frozen representation backbone. Because these experiments use different candidate-score matrices, they define different full-set rankings and should not be compared entry by entry.

Table 17: Detailed evaluation-subset results for the main ResNet50 benchmark.

| Budget | Method | CIFAR10 | CIFAR100 | Oxford Pets | STL10 |
|---|---|---|---|---|---|
| 0.5% | Difficulty | $0.074_{\pm0.000}$ | $-0.400_{\pm0.000}$ | $-0.032_{\pm0.000}$ | $0.737_{\pm0.000}$ |
| | Disagreement | $-0.032_{\pm0.000}$ | $-0.442_{\pm0.000}$ | $-0.032_{\pm0.000}$ | $0.747_{\pm0.000}$ |
| | K center | $0.535_{\pm0.159}$ | $0.655_{\pm0.144}$ | $0.213_{\pm0.089}$ | $-0.158_{\pm0.653}$ |
| | Random | $0.554_{\pm0.131}$ | $0.741_{\pm0.063}$ | $0.139_{\pm0.159}$ | $-0.221_{\pm0.658}$ |
| | Stratified | $0.579_{\pm0.182}$ | $0.693_{\pm0.138}$ | $0.232_{\pm0.220}$ | $-0.629_{\pm0.020}$ |
| | Loss k-center | $0.204_{\pm0.009}$ | $0.145_{\pm0.038}$ | $0.636_{\pm0.009}$ | $0.937_{\pm0.000}$ |
| | Loss k-medoids | $0.255_{\pm0.005}$ | $0.122_{\pm0.019}$ | $0.674_{\pm0.000}$ | $0.958_{\pm0.000}$ |
| | FW mean | $0.825_{\pm0.042}$ | $0.937_{\pm0.021}$ | $0.105_{\pm0.000}$ | $0.074_{\pm0.000}$ |
| | Pairwise hinge | $-0.118_{\pm0.020}$ | $0.076_{\pm0.033}$ | $-0.074_{\pm0.000}$ | $-0.074_{\pm0.000}$ |
| | **VERDICT** | $\mathbf{0.899}_{\pm0.047}$ | $\mathbf{0.935}_{\pm0.046}$ | $\mathbf{0.556}_{\pm0.005}$ | $\mathbf{0.789}_{\pm0.000}$ |
| 1.0% | Difficulty | $0.084_{\pm0.000}$ | $-0.379_{\pm0.000}$ | $0.021_{\pm0.000}$ | $0.695_{\pm0.000}$ |
| | Disagreement | $0.095_{\pm0.000}$ | $-0.347_{\pm0.000}$ | $-0.021_{\pm0.000}$ | $0.726_{\pm0.000}$ |
| | K center | $0.545_{\pm0.038}$ | $0.655_{\pm0.117}$ | $0.122_{\pm0.060}$ | $0.267_{\pm0.510}$ |
| | Random | $0.659_{\pm0.094}$ | $0.718_{\pm0.065}$ | $0.162_{\pm0.183}$ | $-0.339_{\pm0.475}$ |
| | Stratified | $0.707_{\pm0.111}$ | $0.699_{\pm0.070}$ | $0.392_{\pm0.191}$ | $0.011_{\pm0.658}$ |
| | Loss k-center | $0.219_{\pm0.014}$ | $0.145_{\pm0.017}$ | $0.674_{\pm0.000}$ | $0.941_{\pm0.014}$ |
| | Loss k-medoids | $0.255_{\pm0.005}$ | $0.152_{\pm0.046}$ | $0.794_{\pm0.006}$ | $0.907_{\pm0.005}$ |
| | FW mean | $0.865_{\pm0.014}$ | $0.956_{\pm0.009}$ | $0.232_{\pm0.000}$ | $0.305_{\pm0.000}$ |
| | Pairwise hinge | $-0.145_{\pm0.009}$ | $0.036_{\pm0.012}$ | $-0.074_{\pm0.000}$ | $-0.074_{\pm0.000}$ |
| | **VERDICT** | $\mathbf{0.920}_{\pm0.028}$ | $\mathbf{0.968}_{\pm0.025}$ | $\mathbf{0.895}_{\pm0.062}$ | $\mathbf{0.798}_{\pm0.014}$ |
| 2.0% | Difficulty | $0.074_{\pm0.000}$ | $-0.326_{\pm0.000}$ | $-0.011_{\pm0.000}$ | $0.726_{\pm0.000}$ |
| | Disagreement | $0.179_{\pm0.000}$ | $-0.358_{\pm0.000}$ | $-0.063_{\pm0.000}$ | $0.726_{\pm0.000}$ |
| | K center | $0.663_{\pm0.049}$ | $0.705_{\pm0.106}$ | $0.116_{\pm0.041}$ | $0.337_{\pm0.284}$ |
| | Random | $0.672_{\pm0.144}$ | $0.834_{\pm0.044}$ | $0.244_{\pm0.239}$ | $-0.352_{\pm0.611}$ |
| | Stratified | $0.735_{\pm0.085}$ | $0.897_{\pm0.019}$ | $0.339_{\pm0.335}$ | $-0.253_{\pm0.577}$ |
| | Loss k-center | $0.276_{\pm0.012}$ | $0.175_{\pm0.009}$ | $0.709_{\pm0.027}$ | $0.968_{\pm0.000}$ |
| | Loss k-medoids | $0.248_{\pm0.021}$ | $0.164_{\pm0.025}$ | $0.709_{\pm0.009}$ | $0.989_{\pm0.000}$ |
| | FW mean | $0.888_{\pm0.012}$ | $0.983_{\pm0.009}$ | $0.242_{\pm0.000}$ | $0.516_{\pm0.000}$ |
| | Pairwise hinge | $-0.154_{\pm0.006}$ | $0.585_{\pm0.083}$ | $-0.074_{\pm0.000}$ | $-0.074_{\pm0.000}$ |
| | **VERDICT** | $\mathbf{0.935}_{\pm0.038}$ | $\mathbf{0.987}_{\pm0.012}$ | $\mathbf{0.958}_{\pm0.024}$ | $\mathbf{0.798}_{\pm0.014}$ |
| 5.0% | Difficulty | $0.105_{\pm0.000}$ | $-0.284_{\pm0.000}$ | $0.074_{\pm0.000}$ | $0.758_{\pm0.000}$ |
| | Disagreement | $0.221_{\pm0.000}$ | $-0.095_{\pm0.000}$ | $0.053_{\pm0.000}$ | $0.758_{\pm0.000}$ |
| | K center | $0.682_{\pm0.067}$ | $0.821_{\pm0.116}$ | $0.141_{\pm0.041}$ | $0.661_{\pm0.067}$ |
| | Random | $0.800_{\pm0.105}$ | $0.893_{\pm0.040}$ | $0.371_{\pm0.343}$ | $0.246_{\pm0.413}$ |
| | Stratified | $0.773_{\pm0.123}$ | $0.886_{\pm0.047}$ | $0.371_{\pm0.288}$ | $0.126_{\pm0.500}$ |
| | Loss k-center | $0.307_{\pm0.012}$ | $0.194_{\pm0.016}$ | $0.798_{\pm0.014}$ | $1.000_{\pm0.000}$ |
| | Loss k-medoids | $0.297_{\pm0.005}$ | $0.185_{\pm0.006}$ | $0.794_{\pm0.016}$ | $1.000_{\pm0.000}$ |
| | FW mean | $0.937_{\pm0.020}$ | $0.992_{\pm0.005}$ | $0.579_{\pm0.000}$ | $0.811_{\pm0.000}$ |
| | Pairwise hinge | $-0.158_{\pm0.000}$ | $0.813_{\pm0.033}$ | $-0.368_{\pm0.000}$ | $-0.074_{\pm0.000}$ |
| | **VERDICT** | $\mathbf{0.981}_{\pm0.022}$ | $\mathbf{0.996}_{\pm0.009}$ | $\mathbf{0.958}_{\pm0.024}$ | $\mathbf{0.859}_{\pm0.071}$ |

Table 18: Number of datasets with Ranking Inversion (Kendall $\tau < 0$) at 0.5% budget. Lower is better.

| Backbone | VERDICT | Random | Stratified | Difficulty | Disagreement | k-center |
|---|---|---|---|---|---|---|
| ConvNeXt-T | **0** | 1 | 0 | 6 | 5 | 1 |
| ResNet18 | **0** | 1 | 2 | 0 | 0 | 2 |
| ResNet50 | **0** | 2 | 2 | 2 | 2 | 3 |
| Swin-T | **0** | 3 | 1 | 1 | 1 | 3 |
| ViT-B/16 | **0** | 1 | 2 | 0 | 0 | 3 |

Table 19: Minimum evaluation budget required for VERDICT to reach mean Kendall's $\tau \geq 0.9$ in the backbone-sensitivity study. Each dataset–backbone pair defines an independent model-selection instance with a separately constructed $M = 20$ candidate pool on the corresponding frozen representation. Each cell reports the smallest evaluated budget in $\{0.5\%, 1\%, 2\%, 5\%, 10\%, 20\%\}$ at which the mean Kendall's $\tau$ over five subset-selection seeds reaches 0.9. Values are derived from the per-backbone experiments reported in Tables 19–22; in particular, the ResNet50 column is derived from Table 21.

| Dataset | ConvNeXt Tiny | ResNet18 | ResNet50 | Swin T | ViT B/16 |
|---|---|---|---|---|---|
| CIFAR10 | 0.5% | 0.5% | 0.5% | 0.5% | 0.5% |
| CIFAR100 | 0.5% | 0.5% | 2% | 0.5% | 0.5% |
| OxfordPets | 0.5% | 0.5% | 0.5% | 0.5% | 0.5% |
| STL10 | 0.5% | 0.5% | 0.5% | 0.5% | 0.5% |

Table 20: Results for **ConvNeXt-Tiny**.

**(1) Kendall $\tau$ @ 0.5% per dataset**

| Method | CIFAR10 | CIFAR100 | OxfordPets | STL10 |
|---|---|---|---|---|
| VERDICT | 0.96±0.01 | 1.00 | 1.00±0.00 | 0.96 |
| Random | 0.74±0.06 | 0.70±0.05 | 0.70±0.10 | -0.04±0.43 |
| Stratified | 0.71±0.08 | 0.83±0.06 | 0.56±0.46 | 0.54±0.09 |
| Difficulty | -0.60 | -0.62 | -0.40 | -0.40 |
| Disagreement | -0.64 | -0.54 | -0.56 | -0.40 |
| k-center | 0.76±0.05 | 0.87±0.01 | 0.62±0.08 | 0.42±0.06 |

**(2) Macro Kendall $\tau$ vs Budget**

| Budget | VERDICT | Random | Stratified | Difficulty | Disagreement | k-center |
|---|---|---|---|---|---|---|
| 0.5% | 0.984±0.017 | 0.637±0.278 | 0.736±0.129 | -0.355±0.429 | -0.162±0.555 | 0.484±0.410 |
| 1% | 0.989±0.015 | 0.709±0.163 | 0.639±0.327 | -0.353±0.426 | 0.087±0.577 | 0.513±0.302 |
| 2% | 0.989±0.012 | 0.759±0.135 | 0.700±0.185 | -0.292±0.483 | 0.368±0.496 | 0.555±0.405 |

**(3) Min Budget ($\tau \geq 0.9$)**

| Dataset | VERDICT | Baseline min budget | Baseline at min budget | Base. $\tau$@0.5% | Base. $\tau$@Max |
|---|---|---|---|---|---|
| CIFAR10 | 0.5% | 10% | Disagreement | -0.64 | 0.97 |
| CIFAR100 | 0.5% | 10% | Disagreement | -0.54 | 0.94 |
| OxfordPets | 0.5% | 20% | Stratified | 0.56 | 0.92 |
| STL10 | 0.5% | 20% | Difficulty | -0.40 | 0.93 |

Table 21: Results for **ResNet18**.

**(1) Kendall $\tau$ @ 0.5% per dataset**

| Method | CIFAR10 | CIFAR100 | OxfordPets | STL10 |
|---|---|---|---|---|
| VERDICT | 1.00±0.01 | 0.99±0.01 | 0.92 | 0.98 |
| Random | 0.75±0.10 | 0.81±0.07 | -0.04±0.27 | 0.10±0.35 |
| Stratified | 0.82±0.03 | 0.86±0.03 | -0.09±0.40 | -0.06±0.11 |
| Difficulty | 0.77 | 0.86 | 0.71 | 0.84 |
| Disagreement | 0.80 | 0.53 | 0.73 | 0.79 |
| k-center | 0.67±0.06 | 0.70±0.09 | -0.13±0.10 | -0.13 |

**(2) Macro Kendall $\tau$ vs Budget**

| Budget | VERDICT | Random | Stratified | Difficulty | Disagreement | k-center |
|---|---|---|---|---|---|---|
| 0.5% | 0.962±0.041 | 0.518±0.335 | 0.537±0.405 | 0.687±0.154 | 0.647±0.143 | 0.381±0.355 |
| 1% | 0.975±0.039 | 0.555±0.365 | 0.651±0.297 | 0.753±0.163 | 0.672±0.148 | 0.506±0.326 |
| 2% | 0.978±0.034 | 0.617±0.346 | 0.689±0.285 | 0.767±0.163 | 0.729±0.119 | 0.545±0.336 |

**(3) Min Budget ($\tau \geq 0.9$)**

| Dataset | VERDICT | Baseline min budget | Baseline at min budget | Base. $\tau$@0.5% | Base. $\tau$@Max |
|---|---|---|---|---|---|
| CIFAR10 | 0.5% | 5% | Random | 0.75 | 0.95 |
| CIFAR100 | 0.5% | 2% | Stratified | 0.86 | 0.98 |
| OxfordPets | 0.5% | 1% | Difficulty | 0.71 | 1.00 |
| STL10 | 0.5% | 1% | Difficulty | 0.84 | 1.00 |

Table 22: Results for **ResNet50**.

**(1) Kendall $\tau$ @ 0.5% per dataset**

| Method | CIFAR10 | CIFAR100 | OxfordPets | STL10 |
|---|---|---|---|---|
| VERDICT | 0.90±0.06 | 0.87±0.09 | 0.91 | 1.00 |
| Random | 0.34±0.09 | 0.56±0.12 | -0.49±0.25 | -0.36±0.68 |
| Stratified | 0.29±0.31 | 0.50±0.19 | -0.29±0.38 | -0.83±0.02 |
| Difficulty | 0.22 | 0.03 | 0.71 | 0.94 |
| Disagreement | 0.14 | 0.06 | 0.62 | 0.98 |
| k-center | 0.57±0.14 | 0.46±0.18 | -0.23±0.19 | -0.26±0.61 |

**(2) Macro Kendall $\tau$ vs Budget**

| Budget | VERDICT | Random | Stratified | Difficulty | Disagreement | k-center |
|---|---|---|---|---|---|---|
| 0.5% | 0.952±0.053 | 0.333±0.512 | 0.209±0.547 | 0.189±0.653 | 0.155±0.669 | 0.178±0.570 |
| 1% | 0.958±0.046 | 0.390±0.456 | 0.462±0.308 | 0.202±0.648 | 0.186±0.651 | 0.197±0.599 |
| 2% | 0.964±0.039 | 0.442±0.546 | 0.541±0.428 | 0.226±0.655 | 0.229±0.629 | 0.215±0.603 |

**(3) Min Budget ($\tau \geq 0.9$)**

| Dataset | VERDICT | Baseline min budget | Baseline at min budget | Base. $\tau$@0.5% | Base. $\tau$@Max |
|---|---|---|---|---|---|
| CIFAR10 | 0.5% | > 20% | - | - | - |
| CIFAR100 | 2% | 20% | Stratified | 0.50 | 0.95 |
| OxfordPets | 0.5% | > 20% | - | - | - |
| STL10 | 0.5% | 0.5% | Difficulty | 0.94 | 1.00 |

Table 23: Results for **Swin-T**.

**(1) Kendall $\tau$ @ 0.5% per dataset**

| Method | CIFAR10 | CIFAR100 | OxfordPets | STL10 |
|---|---|---|---|---|
| VERDICT | 0.96±0.04 | 0.98±0.02 | 0.91 | 0.99 |
| Random | 0.34±0.29 | 0.71±0.10 | -0.54±0.10 | -0.38±0.56 |
| Stratified | 0.23±0.16 | 0.64±0.15 | 0.20±0.48 | -0.71±0.18 |
| Difficulty | 0.73 | 0.71 | 0.70 | 0.91 |
| Disagreement | 0.74 | 0.81 | 0.77 | 0.91 |
| k-center | 0.60±0.06 | 0.69±0.05 | -0.26±0.26 | 0.67 |

**(2) Macro Kendall $\tau$ vs Budget**

| Budget | VERDICT | Random | Stratified | Difficulty | Disagreement | k-center |
|---|---|---|---|---|---|---|
| 0.5% | 0.976±0.032 | 0.211±0.562 | 0.343±0.492 | 0.568±0.494 | 0.540±0.582 | 0.189±0.606 |
| 1% | 0.982±0.020 | 0.309±0.545 | 0.356±0.492 | 0.588±0.572 | 0.535±0.596 | 0.376±0.527 |
| 2% | 0.983±0.020 | 0.439±0.422 | 0.522±0.460 | 0.612±0.548 | 0.592±0.540 | 0.421±0.507 |

**(3) Min Budget ($\tau \geq 0.9$)**

| Dataset | VERDICT | Baseline min budget | Baseline at min budget | Base. $\tau$@0.5% | Base. $\tau$@Max |
|---|---|---|---|---|---|
| CIFAR10 | 0.5% | 20% | Difficulty | 0.73 | 0.94 |
| CIFAR100 | 0.5% | > 20% | - | - | - |
| OxfordPets | 0.5% | 10% | Difficulty | 0.70 | 0.98 |
| STL10 | 0.5% | 0.5% | Difficulty | 0.91 | 1.00 |

Table 24: Results for **ViT-B/16**.

**(1) Kendall $\tau$ @ 0.5% per dataset**

| Method | CIFAR10 | CIFAR100 | OxfordPets | STL10 |
|---|---|---|---|---|
| VERDICT | 0.99±0.00 | 0.98±0.01 | 0.93±0.01 | 0.99 |
| Random | 0.42±0.36 | 0.65±0.18 | -0.35±0.57 | 0.03±0.40 |
| Stratified | 0.41±0.39 | 0.66±0.09 | -0.03±0.40 | -0.46±0.13 |
| Difficulty | 0.94 | 0.90 | 0.85 | 0.83 |
| Disagreement | 0.86 | 0.86 | 0.85 | 0.87 |
| k-center | 0.62±0.38 | 0.80±0.04 | -0.17±0.68 | -0.70±0.07 |

**(2) Macro Kendall $\tau$ vs Budget**

| Budget | VERDICT | Random | Stratified | Difficulty | Disagreement | k-center |
|---|---|---|---|---|---|---|
| 0.5% | 0.971±0.036 | 0.255±0.402 | 0.345±0.434 | 0.818±0.112 | 0.752±0.238 | 0.171±0.522 |
| 1% | 0.977±0.026 | 0.300±0.501 | 0.403±0.521 | 0.845±0.121 | 0.788±0.229 | 0.326±0.533 |
| 2% | 0.983±0.022 | 0.437±0.403 | 0.494±0.509 | 0.868±0.103 | 0.857±0.102 | 0.528±0.332 |

**(3) Min Budget ($\tau \geq 0.9$)**

| Dataset | VERDICT | Baseline min budget | Baseline at min budget | Base. $\tau$@0.5% | Base. $\tau$@Max |
|---|---|---|---|---|---|
| CIFAR10 | 0.5% | 0.5% | Difficulty | 0.94 | 1.00 |
| CIFAR100 | 0.5% | 1% | Difficulty | 0.90 | 0.93 |
| OxfordPets | 0.5% | 2% | Difficulty | 0.85 | 1.00 |
| STL10 | 0.5% | 1% | Difficulty | 0.83 | 1.00 |

# I Runtime and scalability

## I.1 Runtime components and measurement protocol

For each dataset, we record the following runtime components for the direct train-and-verify search:

- **End-to-end search time** (`select_time_sec`): total wall-clock time for the full VERDICT search path under the fixed proposal budget.

- **Proposal-construction time** (`score_time_sec`): time spent computing the decision-response scores, forming the working-set proposal, and constructing the proposed subset at each search iteration.

- **Incumbent retrain-and-verify time** (`current_train_verify_time_sec`): time spent retraining and checking the currently accepted design at each iteration.

- **Proposed-design retrain-and-verify time** (`candidate_retrain_verify_time_sec`): time spent retraining on the newly proposed design and verifying the decision-margin constraints under the fixed protocol.

- **Per-iteration averages**: proposal-construction and retrain-and-verify totals divided by the number of search iterations. These averages summarize the cost of one VERDICT proposal-and-check step.

All timings are single-run wall-clock seconds. In the direct-acquisition experiments, VERDICT uses a fixed 30-iteration search budget, so the per-iteration quantities are obtained by dividing the corresponding totals by 30 iterations.

## I.2 Direct training acquisition runtime breakdown

The runtime breakdown shows that the direct train-and-verify search is generally dominated by proposal construction rather than by retraining-based verification. On most datasets, the decision-response scoring and subset-proposal step accounts for the largest share of wall-clock time, especially on CIFAR10 and CIFAR100. By contrast, retraining and verification of the incumbent and proposed designs remains comparatively modest. VERDICT is fast on tabular datasets, while text and vision datasets require more time due to larger feature spaces, more scenarios, or larger candidate/margin computations. The main outlier is CIFAR100, where almost all of the cost comes from proposal construction rather than verification. These results indicate that the verification step, although essential for correctness, is not the primary computational bottleneck in most settings; scaling VERDICT further would mainly require accelerating the decision-response scoring and working-set proposal computation.

Table 25: Runtime breakdown across datasets on direct training acquisition over 30 proposal iterations.

| Dataset | End-to-end search | Proposal construction | Incumbent retrain-verify | Proposed-design retrain+verify | Proposal construction per iter. | Retrain-verify per iter. |
|---|---|---|---|---|---|---|
| Credit | 4.22 | 3.01 | 0.62 | 0.58 | 0.10 | 0.04 |
| Bank | 2.38 | 1.65 | 0.36 | 0.35 | 0.06 | 0.02 |
| Adult | 8.44 | 6.23 | 1.11 | 1.08 | 0.21 | 0.07 |
| OxfordPets | 138.94 | 86.88 | 26.23 | 25.51 | 2.90 | 1.72 |
| STL10 | 90.14 | 29.96 | 29.80 | 30.01 | 1.00 | 1.99 |
| CIFAR10 | 467.31 | 386.65 | 41.71 | 38.29 | 12.89 | 2.67 |
| CIFAR100 | 2127.96 | 2046.56 | 41.54 | 39.47 | 68.22 | 2.70 |
| 20Newsgroups | 129.64 | 104.07 | 13.10 | 12.43 | 3.47 | 0.85 |
| AG News | 285.11 | 173.42 | 57.19 | 54.40 | 5.78 | 3.72 |

### I.3 Runtime tradeoff

To distinguish evidence compression from computational savings, we compare the one-time cost of constructing a verified subset with the runtime of subsequent executions of the full and reduced protocols. Table 26 reports VERDICT's end-to-end search cost, the retained fraction, and the measured runtime of retraining and evaluating the candidate set under both designs. This comparison indicates whether a smaller decision-preserving subset also reduces per-run computation, while making explicit the upfront overhead required to identify and verify that subset.

Table 26: Runtime comparison between the full and subset protocols. The retained fraction is the proportion of evidence retained by the selected subset.

| Dataset | Retained fraction | Search cost (s) | Full protocol (s) | Subset protocol (s) | Per-run reduction fraction | Speedup |
|---|---|---|---|---|---|---|
| Credit | 0.427 | 4.22 | 0.0249 | 0.0112 | 0.549 | 2.22 |
| Bank | 0.413 | 2.38 | 0.0190 | 0.0091 | 0.520 | 2.08 |
| Adult | 0.619 | 8.44 | 0.0457 | 0.0338 | 0.261 | 1.35 |
| OxfordPets | 0.222 | 138.94 | 0.8909 | 0.8227 | 0.077 | 1.08 |
| STL10 | 0.073 | 90.14 | 0.9240 | 0.9563 | -0.035 | 0.97 |
| CIFAR10 | 0.144 | 467.31 | 1.5926 | 1.0633 | 0.332 | 1.50 |
| CIFAR100 | 0.020 | 2127.96 | 1.6944 | 1.1567 | 0.317 | 1.46 |
| 20Newsgroups | 0.458 | 129.64 | 0.5707 | 0.3483 | 0.390 | 1.64 |
| AG News | 0.544 | 285.11 | 2.2802 | 2.3176 | -0.016 | 0.98 |

The selected subsets reduce subsequent protocol runtime on seven of the nine datasets, with substantial relative reductions on the tabular datasets, CIFAR10, CIFAR100, and 20Newsgroups. OxfordPets shows a smaller reduction, whereas STL10 and AG News show no reduction under the recorded timings. These latter cases indicate that, when the underlying protocol is already inexpensive, fixed execution overhead and solver-level variation can dominate the effect of reducing the training set.

Table 26 also shows that the one-time search cost exceeds the runtime of a single full or reduced protocol execution, particularly for the image datasets. This overhead is driven primarily by response scoring and proposal construction rather than retraining-based verification. Because our direct-acquisition experiments use fixed representations and inexpensive convex ERM, reductions in retained evidence yield only modest absolute runtime savings. Thus, the results demonstrate evidence compression and lower subsequent execution cost on most datasets, but not an end-to-end wall-clock advantage for one-off use. Such an advantage is more plausible when the forward protocol is substantially more expensive or when the verified subset is reused across repeated model, checkpoint, or hyperparameter comparisons.

### I.4 Runtime comparison with other baselines

The runtime comparison separates inexpensive one-shot selection from iterative verified search. Random, stratified, loss-, gradient-, and coverage-based methods construct one candidate subset before verification, whereas VERDICT uses up to 30 adaptive proposal-and-check steps. Its larger runtime is therefore the computational cost of exploring the decision-feasible region rather than of producing a single ranking of examples. Despite this additional search, VERDICT finishes within eight minutes on all datasets except CIFAR100. The CIFAR100 result is consistent with the component breakdown, where response scoring and proposal construction dominate the runtime, identifying these operations, rather than direct verification alone, as the principal target for further scaling. Combined with the verified retained fractions, the table shows that inexpensive selection and strong decision-preserving compression are distinct objectives.

### I.5 Evaluation runtime breakdown on vision datasets

For each dataset, budget, method, and seed, we record:

Table 27: Runtime comparison in seconds. Random through DecisionIF report mean $\pm$ standard deviation of total runtime, where total runtime is selection time plus retraining-verification time. VERDICT reports end-to-end search runtime.

| Dataset | Random | Stratified | HighLoss | GradNorm | KCenter | DecisionIF | VERDICT |
|---|---|---|---|---|---|---|---|
| Credit | $0.049 \pm 0.004$ | $0.050 \pm 0.001$ | $0.103 \pm 0.001$ | $0.129 \pm 0.002$ | $8.460 \pm 0.099$ | $0.125 \pm 0.002$ | 4.220 |
| Bank | $0.015 \pm 0.003$ | $0.025 \pm 0.001$ | $0.098 \pm 0.006$ | $0.137 \pm 0.002$ | $5.736 \pm 0.115$ | $0.106 \pm 0.001$ | 2.380 |
| Adult | $0.068 \pm 0.006$ | $0.065 \pm 0.001$ | $0.182 \pm 0.014$ | $0.189 \pm 0.025$ | $10.139 \pm 0.266$ | $0.246 \pm 0.001$ | 8.440 |
| OxfordPets | $1.663 \pm 0.035$ | $1.593 \pm 0.217$ | $2.350 \pm 0.031$ | $2.395 \pm 0.145$ | $1.468 \pm 0.163$ | $8.157 \pm 0.308$ | 138.940 |
| STL10 | $1.126 \pm 0.073$ | $1.151 \pm 0.061$ | $1.822 \pm 0.230$ | $2.434 \pm 0.119$ | $1.767 \pm 0.321$ | $4.184 \pm 0.288$ | 90.140 |
| CIFAR10 | $1.328 \pm 0.058$ | $1.187 \pm 0.028$ | $3.683 \pm 0.045$ | $4.420 \pm 0.119$ | $5.106 \pm 0.353$ | $24.603 \pm 0.525$ | 467.310 |
| CIFAR100 | $1.533 \pm 0.128$ | $1.413 \pm 0.058$ | $4.599 \pm 0.388$ | $5.276 \pm 0.087$ | $2.522 \pm 0.251$ | $241.928 \pm 1.722$ | 2127.960 |
| 20Newsgroups | $0.540 \pm 0.013$ | $0.520 \pm 0.074$ | $1.001 \pm 0.192$ | $1.048 \pm 0.097$ | $1.523 \pm 0.058$ | $4.756 \pm 0.310$ | 129.640 |
| AG News | $2.358 \pm 0.007$ | $2.318 \pm 0.009$ | $4.790 \pm 0.158$ | $6.663 \pm 0.011$ | $9.688 \pm 0.495$ | $10.681 \pm 0.419$ | 285.110 |

- **Verification and metrics time** (`verify_metrics_time_sec`): time to compute model means on the selected subset and compute Kendall $\tau$, Spearman $\rho$, winner agreement, pairwise agreement, margin violation rate, and the subset minimum margin magnitude.

- **End-to-end time within the script** (`end2end_time_sec`): selection time plus verification and metrics time.

Table 28 reports a breakdown for CIFAR10, CIFAR100, STL10, and Oxford Pets at budgets near 5% with three seeds. Across all datasets and methods, the verification and metrics stage takes about 1 to 3 milliseconds, while selection dominates `end2end_time_sec`. The table also reports ranking and decision preservation metrics under the same cached score setting, enabling direct comparison of methods at fixed budgets.

Table 28: Runtime breakdown ($\text{mean}_{\pm\ std}$ over seeds).

| Dataset | Budget | Method | Select Time (s) | Verify Time (s) | End-to-End Time (s) |
|---|---|---|---|---|---|
| CIFAR10 | 0.05 | VERDICT | $14.489_{\pm 0.052}$ | $0.001_{\pm 0.000}$ | $14.490_{\pm 0.052}$ |
| CIFAR10 | 0.05 | kcenter | $37.454_{\pm 0.757}$ | $0.002_{\pm 0.000}$ | $37.456_{\pm 0.757}$ |
| CIFAR10 | 0.05 | random | $0.000_{\pm 0.000}$ | $0.001_{\pm 0.000}$ | $0.001_{\pm 0.000}$ |
| CIFAR100 | 0.05 | VERDICT | $14.750_{\pm 0.356}$ | $0.001_{\pm 0.000}$ | $14.751_{\pm 0.356}$ |
| CIFAR100 | 0.05 | kcenter | $37.981_{\pm 0.957}$ | $0.003_{\pm 0.001}$ | $37.983_{\pm 0.956}$ |
| CIFAR100 | 0.05 | random | $0.000_{\pm 0.000}$ | $0.001_{\pm 0.000}$ | $0.001_{\pm 0.000}$ |
| OxfordPets | 0.05 | VERDICT | $0.405_{\pm 0.038}$ | $0.001_{\pm 0.000}$ | $0.406_{\pm 0.038}$ |
| OxfordPets | 0.05 | kcenter | $0.262_{\pm 0.007}$ | $0.002_{\pm 0.000}$ | $0.264_{\pm 0.007}$ |
| OxfordPets | 0.05 | random | $0.000_{\pm 0.000}$ | $0.001_{\pm 0.000}$ | $0.001_{\pm 0.000}$ |
| STL10 | 0.05 | VERDICT | $0.786_{\pm 0.034}$ | $0.001_{\pm 0.000}$ | $0.787_{\pm 0.034}$ |
| STL10 | 0.05 | kcenter | $0.534_{\pm 0.005}$ | $0.002_{\pm 0.000}$ | $0.535_{\pm 0.006}$ |
| STL10 | 0.05 | random | $0.000_{\pm 0.000}$ | $0.001_{\pm 0.000}$ | $0.001_{\pm 0.000}$ |

## I.6 Scaling with number of examples $N$

Table 29 evaluates scaling on CIFAR100 by subsampling the evaluation set to fractions $\{0.10, 0.25, 0.50, 1.00\}$ while keeping the cached model set fixed. We report budgets 1% and 5% and three seeds. For VERDICT, selection time increases with the number of examples, while verification and metrics time remains near 0.001 seconds across fractions.

Table 29: Scaling in $N$ runtime summary (mean$_{\pm std}$ over 3 seeds and 3 runs).

| Dataset | Frac. | Budget | Method | End-to-End Time (s) | Select Time (s) | Verify Time (s) |
|---------|-------|--------|--------|---------------------|-----------------|-----------------|
| CIFAR100 | 0.10 | 0.01 | VERDICT | $0.23_{\pm 0.07}$ | $0.23_{\pm 0.07}$ | $0.00_{\pm 0.00}$ |
| CIFAR100 | 0.10 | 0.01 | kcenter | $0.13_{\pm 0.00}$ | $0.13_{\pm 0.00}$ | $0.00_{\pm 0.00}$ |
| CIFAR100 | 0.10 | 0.01 | random | $0.00_{\pm 0.00}$ | $0.00_{\pm 0.00}$ | $0.00_{\pm 0.00}$ |
| CIFAR100 | 0.10 | 0.05 | VERDICT | $0.73_{\pm 0.04}$ | $0.73_{\pm 0.04}$ | $0.00_{\pm 0.00}$ |
| CIFAR100 | 0.10 | 0.05 | kcenter | $0.42_{\pm 0.00}$ | $0.42_{\pm 0.00}$ | $0.00_{\pm 0.00}$ |
| CIFAR100 | 0.10 | 0.05 | random | $0.00_{\pm 0.00}$ | $0.00_{\pm 0.00}$ | $0.00_{\pm 0.00}$ |
| CIFAR100 | 0.25 | 0.01 | VERDICT | $0.98_{\pm 0.04}$ | $0.98_{\pm 0.04}$ | $0.00_{\pm 0.00}$ |
| CIFAR100 | 0.25 | 0.01 | kcenter | $0.58_{\pm 0.11}$ | $0.58_{\pm 0.10}$ | $0.00_{\pm 0.00}$ |
| CIFAR100 | 0.25 | 0.01 | random | $0.00_{\pm 0.00}$ | $0.00_{\pm 0.00}$ | $0.00_{\pm 0.00}$ |
| CIFAR100 | 0.25 | 0.05 | VERDICT | $4.76_{\pm 0.03}$ | $4.76_{\pm 0.03}$ | $0.00_{\pm 0.00}$ |
| CIFAR100 | 0.25 | 0.05 | kcenter | $2.32_{\pm 0.00}$ | $2.32_{\pm 0.00}$ | $0.00_{\pm 0.00}$ |
| CIFAR100 | 0.25 | 0.05 | random | $0.00_{\pm 0.00}$ | $0.00_{\pm 0.00}$ | $0.00_{\pm 0.00}$ |
| CIFAR100 | 0.50 | 0.01 | VERDICT | $1.49_{\pm 0.05}$ | $1.49_{\pm 0.05}$ | $0.00_{\pm 0.00}$ |
| CIFAR100 | 0.50 | 0.01 | kcenter | $1.74_{\pm 0.19}$ | $1.74_{\pm 0.19}$ | $0.00_{\pm 0.00}$ |
| CIFAR100 | 0.50 | 0.01 | random | $0.00_{\pm 0.00}$ | $0.00_{\pm 0.00}$ | $0.00_{\pm 0.00}$ |
| CIFAR100 | 0.50 | 0.05 | VERDICT | $7.34_{\pm 0.08}$ | $7.34_{\pm 0.08}$ | $0.00_{\pm 0.00}$ |
| CIFAR100 | 0.50 | 0.05 | kcenter | $8.23_{\pm 0.25}$ | $8.23_{\pm 0.25}$ | $0.00_{\pm 0.00}$ |
| CIFAR100 | 0.50 | 0.05 | random | $0.00_{\pm 0.00}$ | $0.00_{\pm 0.00}$ | $0.00_{\pm 0.00}$ |
| CIFAR100 | 1.00 | 0.01 | VERDICT | $3.00_{\pm 0.04}$ | $3.00_{\pm 0.04}$ | $0.00_{\pm 0.00}$ |
| CIFAR100 | 1.00 | 0.01 | kcenter | $6.74_{\pm 0.15}$ | $6.74_{\pm 0.15}$ | $0.00_{\pm 0.00}$ |
| CIFAR100 | 1.00 | 0.01 | random | $0.00_{\pm 0.00}$ | $0.00_{\pm 0.00}$ | $0.00_{\pm 0.00}$ |
| CIFAR100 | 1.00 | 0.05 | VERDICT | $14.89_{\pm 0.24}$ | $14.89_{\pm 0.25}$ | $0.00_{\pm 0.00}$ |
| CIFAR100 | 1.00 | 0.05 | kcenter | $37.67_{\pm 0.93}$ | $37.67_{\pm 0.93}$ | $0.00_{\pm 0.00}$ |
| CIFAR100 | 1.00 | 0.05 | random | $0.00_{\pm 0.00}$ | $0.00_{\pm 0.00}$ | $0.00_{\pm 0.00}$ |

## I.7 Scaling with number of models $M$

Table 30 evaluates scaling on CIFAR100 by truncating the cached score matrix to $M \in \{5, 10, 20\}$ models while keeping the evaluation set fixed. We report budgets 1% and 5% and three seeds. For VERDICT, selection time increases with $M$ in this range, while verification and metrics time remains near 0.001 seconds across values of $M$.

The results in Tables 28–30 measure the cost of selection and verification after scores are cached. They do not include the one time cost of computing cached scores and embeddings. When reporting deployment costs, this precomputation cost should be accounted for separately, since it depends on the model architecture, hardware, and batching strategy used for evaluation.

Table 30: Scaling in $M$ runtime summary (mean$_{\pm\,std}$ over 3 seeds and 3 runs).

| Dataset | Max Models | Budget | Method | End-to-End Time (s) | Select Time (s) | Verify Time (s) |
|---------|-----------|--------|--------|---------------------|-----------------|-----------------|
| CIFAR100 | 5 | 0.01 | VERDICT | $2.28_{\pm 0.06}$ | $2.28_{\pm 0.06}$ | $0.00_{\pm 0.00}$ |
| CIFAR100 | 5 | 0.01 | kcenter | $7.29_{\pm 0.13}$ | $7.29_{\pm 0.13}$ | $0.00_{\pm 0.00}$ |
| CIFAR100 | 5 | 0.01 | random | $0.00_{\pm 0.00}$ | $0.00_{\pm 0.00}$ | $0.00_{\pm 0.00}$ |
| CIFAR100 | 5 | 0.05 | VERDICT | $11.75_{\pm 0.54}$ | $11.75_{\pm 0.54}$ | $0.00_{\pm 0.00}$ |
| CIFAR100 | 5 | 0.05 | kcenter | $36.82_{\pm 0.95}$ | $36.81_{\pm 0.95}$ | $0.00_{\pm 0.00}$ |
| CIFAR100 | 5 | 0.05 | random | $0.00_{\pm 0.00}$ | $0.00_{\pm 0.00}$ | $0.00_{\pm 0.00}$ |
| CIFAR100 | 10 | 0.01 | VERDICT | $2.39_{\pm 0.03}$ | $2.39_{\pm 0.03}$ | $0.00_{\pm 0.00}$ |
| CIFAR100 | 10 | 0.01 | kcenter | $7.19_{\pm 0.11}$ | $7.19_{\pm 0.11}$ | $0.00_{\pm 0.00}$ |
| CIFAR100 | 10 | 0.01 | random | $0.00_{\pm 0.00}$ | $0.00_{\pm 0.00}$ | $0.00_{\pm 0.00}$ |
| CIFAR100 | 10 | 0.05 | VERDICT | $11.86_{\pm 0.13}$ | $11.86_{\pm 0.13}$ | $0.00_{\pm 0.00}$ |
| CIFAR100 | 10 | 0.05 | kcenter | $37.17_{\pm 0.55}$ | $37.17_{\pm 0.55}$ | $0.00_{\pm 0.00}$ |
| CIFAR100 | 10 | 0.05 | random | $0.00_{\pm 0.00}$ | $0.00_{\pm 0.00}$ | $0.00_{\pm 0.00}$ |
| CIFAR100 | 20 | 0.01 | VERDICT | $2.97_{\pm 0.05}$ | $2.97_{\pm 0.05}$ | $0.00_{\pm 0.00}$ |
| CIFAR100 | 20 | 0.01 | kcenter | $7.26_{\pm 0.09}$ | $7.26_{\pm 0.09}$ | $0.00_{\pm 0.00}$ |
| CIFAR100 | 20 | 0.01 | random | $0.00_{\pm 0.00}$ | $0.00_{\pm 0.00}$ | $0.00_{\pm 0.00}$ |
| CIFAR100 | 20 | 0.05 | VERDICT | $14.83_{\pm 0.47}$ | $14.83_{\pm 0.47}$ | $0.00_{\pm 0.00}$ |
| CIFAR100 | 20 | 0.05 | kcenter | $36.02_{\pm 0.83}$ | $36.02_{\pm 0.83}$ | $0.00_{\pm 0.00}$ |
| CIFAR100 | 20 | 0.05 | random | $0.00_{\pm 0.00}$ | $0.00_{\pm 0.00}$ | $0.00_{\pm 0.00}$ |

