# OpenReview forum: "Which Examples Preserve Decisions? Verified Inverse Design for Model Selection"
_TMLR — Under review for TMLR_

### Review · Reviewer_hVv3 · 2026-07-04

**Summary Of Contributions:**

The paper introduces decision-preserving acquisition (DPA): given a fixed model-selection protocol and an observed scenario-wise winner map, find low-cost training (or evaluation) evidence that reproduces the same winners with prescribed margins after retraining. Unlike coreset or data-valuation objectives which target predictive performance or full-data approximation, DPA targets the stability of the discrete selection decision itself and the guarantee is kept explicitly protocol-relative.

The authors formalize the inverse-design problem, show that in the convex ERM regime realizability is equivalent to feasibility of a budgeted inverse KKT system (Proposition 4.4), and establish that the associated cost-minimization problem is NP-hard by reduction from Minimum-Cost Set Cover (Theorem D.1). The proposed method, VERDICT, derives a local decision-response operator via implicit differentiation, solves a trust-region convex surrogate over a quasi-active constraint set to propose lower-cost designs and accepts a candidate only after retraining under the fixed protocol verifies the margins. Experiments span nine fixed-feature classification datasets across tabular, vision and text modalities, plus two downstream applications: decision-critical scenario discovery and low-cost evaluation-subset selection.

Key strengths: The problem reframing is useful and is made operational rather than rhetorical. The propose-and-verify architecture is the right design for a setting where the surrogate is provably unreliable since accepted designs do not inherit surrogate approximation error (Proposition 4.11); the paper demonstrates the necessity of verification convincingly, as surrogate-feasible proposals become infeasible after retraining on every dataset in Table 2. The multi-backbone evaluation-subset study (Tables 16 through 23) is the most compelling empirical result, with VERDICT holding rank correlation near 0.95 to 0.98 at a 0.5% budget and producing zero ranking inversions across five architectures.

Key weaknesses: The NP-hardness result motivates abandoning exact optimization but does no work in the method which never targets minimum cost and carries no approximation guarantee. The verification certificate weakens substantially under stochastic training and is close to tautological as evaluated, since designs are certified against the same protocol and seed used to define preservation. The strongest coreset and data-selection baselines appear only in the downstream study, not in the main training-acquisition experiment that carries the primary claim.

**Additional Comments:**

Best of Luck! Looking forward to the final paper.

**Audience:**

Yes

**Audience Explanation:**

The paper sits at the intersection of several active TMLR-relevant areas and multiple segments of the audience would find its findings of interest.

Researchers working on coresets, data selection, data valuation and data-efficient evaluation are the most direct audience. The reframing from performance-preservation or full-data approximation to preserving a discrete model-selection decision is a different objective and even readers who do not adopt VERDICT would find the formulation and the accompanying diagnostics (particularly the surrogate-versus-verified margin gap in Table 2 and the Random@final control) worth knowing. The evaluation-subset results are of independent interest to anyone building cheap validation sets for repeated model or checkpoint comparison and the multi-backbone Kendall-tau study is a useful empirical contribution on its own.

The work also connects to inverse optimization, bilevel and decision-focused learning and influence-function-based attribution and the KKT feasibility characterization gives that audience a clean reference point for the inverse view. The scenario-discovery application further speaks to practitioners concerned with auditing whether a declared evaluation protocol omits decision-critical slices, which is relevant to the robustness and evaluation-methodology community.

The interest is real regardless of the evidence concerns raised in the previous box: those bear on whether specific claims are currently supported not on whether the questions the paper asks and the findings it reports would engage TMLR readers.

**Broader Impact Concerns:**

No unaddressed ethical concerns. The paper includes a Broader Impact Statement (Appendix B) that is appropriately scoped and concrete.

No Broader Impact Statement needs to be added or substantially revised.

**Claims And Evidence:**

No

**Claims Explanation:**

Most claims in the paper are well-supported. The propose-and-verify guarantee (Proposition 4.11), the necessity of the verification step (Table 2, where surrogate-feasible proposals become infeasible after retraining on every dataset), and the evaluation-subset results (Tables 16 through 23) are all backed by clear and convincing evidence. The honest treatment of the Adult and Bank exceptions further supports the paper's central empirical narrative.

Three claims, however, currently outrun their evidence and each is a central problem:

First, the paper's headline framing of a "verified" certificate is stronger than what the experiments establish. Proposition 4.11 guarantees decision-feasibility only with respect to the randomness convention used at verification. Under the fixed-seed convention the certificate holds for a single seed and the paper provides no evidence that a design verified at seed omega_0 preserves the winner map at any other seed. All main-text experiments certify a design against the same protocol and seed used to define preservation, which makes the certificate close to tautological as evaluated: a subset is declared to preserve the decision under exactly the evaluation that defined preservation. The claim that VERDICT produces meaningful decision-preservation certificates is therefore not convincingly supported without a cross-seed stability experiment and the scope of the certificate is stated only in Appendix C.3.

Second, the primary claim that VERDICT finds compact verified subsets where baselines fail (Table 1) rests on a baseline pool that omits the strongest available competitors. The direct training-acquisition experiment compares against Random, Stratified, HighLoss, GradNorm, KCenter, and DecisionIF, while the stronger loss-space and coreset baselines (loss k-center, k-medoids, Frank-Wolfe mean coreset) appear only in the downstream evaluation-subset study. Because the strongest methods are absent from the experiment carrying the main claim, the evidence does not let the reader separate the method's contribution from the weakness of the comparison set.

Third, the claim is undercut by the paper's ablations. DecisionIF uses essentially the same decision-response signal as VERDICT without the verified iteration, yet fails within budget on six of nine datasets (Table 1), and VERDICT-RandomWS remains competitive and even beats default VERDICT on Adult (Table 11). Together these suggest that the verified trust-region loop may be the active ingredient. The claim that the local decision-response model is what enables compact subsets is thus not clearly supported by the presented evidence.

The NP-hardness result (Theorem D.1) appears correct, but it supports a weaker claim than the narrative implies: it establishes hardness of the exact minimum-cost design, whereas VERDICT never targets minimum cost and carries no approximation guarantee, so the theorem does not bear on the quality of the method's output.

None of these gaps concern novelty or significance and all are addressable within a revision: a cross-seed verification experiment, the missing baselines run in the direct-acquisition setting, and a diagnostic isolating the decision-response signal from the verified loop would move this to a clear Yes.

**Requested Changes:**

Each item below is marked as either **[Critical]** or **[Strengthen]**

### Critical

**1. [Critical] State and test the scope of the verification certificate:** The paper's central claim rests on the word "verified," but Proposition 4.11 guarantees decision-feasibility only under the randomness convention used at verification and all main-text experiments certify a design against the same protocol and seed that defined the winner map. Please (a) state the scope of the certificate (single-seed vs. aggregated, and its dependence on the verification seed set) in the abstract and introduction rather than only in Appendix C.3, and (b) add an experiment that verifies a VERDICT-selected subset at held-out seeds distinct from the verification seed, reporting how often the winner map and margins survive. If cross-seed stability is weak, this should be stated as a limitation; if strong, it materially strengthens the paper.

**2. [Critical] Run the strongest baselines in the direct training-acquisition setting:** The main claim in Table 1 relies on a baseline pool (Random, Stratified, HighLoss, GradNorm, KCenter, DecisionIF) that excludes the loss-space and coreset methods (loss k-center, k-medoids, Frank-Wolfe mean coreset) used in the downstream study. Please run these stronger baselines in the Table 1 setting, or justify explicitly why they do not transfer to training-subset acquisition. As written, the reader cannot separate the method's contribution from the weakness of the comparison set.

**3. [Critical] Clarify the logical role of Theorem D.1:** The NP-hardness result is invoked to motivate local search, but VERDICT never targets minimum cost and carries no approximation guarantee or optimality-gap analysis, so the theorem does not bear on the quality of the method's output. Please either connect the hardness result to a structural property the method exploits, or state plainly that it is context rather than motivation and adjust the surrounding narrative accordingly.

### Strengthen

**4. [Strengthen] Isolate whether the decision-response signal or the verified loop drives performance:** DecisionIF uses essentially the same signal without verified iteration and fails on six of nine datasets (Table 1), while VERDICT-RandomWS is competitive and beats default VERDICT on Adult (Table 11). A diagnostic that separates the contribution of the decision-response operator from the verified trust-region wrapper would clarify what the paper is actually contributing, and would strengthen or appropriately temper the emphasis on the local operator.

**5. [Strengthen] Exercise the multi-quality cost model or scope it down:** Section 3 develops quality levels, per-point overheads, and a general cost function, but every experiment sets K=1 with unit cost, so cost equals subset size throughout. Please include at least one experiment with K>1 that demonstrates the quality-level formulation buys something, or defer the general cost model and frame the evaluation honestly around subset-size preservation.

**6. [Strengthen] Report search cost against the cost of the evidence saved:** The method is motivated by the expense of acquiring and retaining data, yet VERDICT's own runtime is dominated by proposal construction and reaches 2128 seconds on CIFAR100 (Table 24). A direct comparison of VERDICT's wall-clock search cost against the retraining or acquisition cost of the data it removes would let a practitioner assess the efficiency claim on net.

**7. [Strengthen] Reconcile the two scenario-discovery tables:** Table 3 (main text) and Table 13 (appendix) report overlapping scenario-discovery results for the same datasets with differing presentation and some differing values. Please make the relationship explicit or consolidate them, so the reader is not left comparing two versions of the same experiment.

### Minor (presentation)

- Section 6: "the downstream action is a often comparison result" (typo).
- Section 6: "fail within the evaluated grid'" (stray apostrophe for a period).
- Definition 4.9 and surrounding text alternate between $\bar g_{m,t}$ and $g_{m,t}$ for the aggregated margin without consistently signaling the convention in force.
- Figure 2 and Figure 3 captions describe the bottom row as showing the surrogate-predicted margin, while the legends distinguish surrogate from verified margins; captions should match the legends.
- Table 3 and Table 13 use "failure splits" and "age slices" as omitted-family labels without a pointer to where these are defined (Appendix H.1); a forward reference at first use would help.

---

> ### Author Response · Authors · 2026-07-19
> **Response to Reviewer hVv3 (1/6)**
>
> **1.1. The paper's central claim rests on the word "verified," but Proposition 4.11 guarantees decision-feasibility only under the randomness convention used at verification and all main-text experiments certify a design against the same protocol and seed that defined the winner map.**
>
> Our use of “verified” is operational and protocol-relative: **VERDICT accepts a proposed design only after rerunning the declared training-and-evaluation protocol and checking the prescribed decision-margin constraints under the stated decision convention**. The resulting certificate therefore applies only to the candidate pool, scenarios, margins, and randomness convention included in that verification check; it does not imply seed-uniform, distributional, or globally optimal preservation beyond the declared protocol.
>
> This scope is already part of the paper’s formal setup. In particular, Section 3.1 states:
>
> > “Fix a baseline design $\alpha^{(0)}$ and a decision convention, either a fixed-seed convention or an aggregated convention over a finite seed set, with deterministic tie-breaking.”
>
> Appendix C.3 then makes the dependence on that convention explicit:
>
> > “When training is stochastic, preserving a selection requires fixing how randomness is handled. We formalize training with an explicit randomness input and verify candidates under the same convention used to define the winners.”
>
> and:
>
> > “All verification uses the same convention (including aggregation and tie-breaking) that defines $D_{\mathrm{val}}$.”
>
> **For a stochastic pipeline, under the fixed-seed convention the certificate is relative to the declared seed $\omega_0$, while under the aggregated convention it is relative to the declared finite seed set $\Omega$ and aggregation rule.** Proposition 4.11 should be read under exactly this protocol specification. The main experiments instantiate the deterministic special case discussed in Q1.2, where trained solutions and verified margins do not depend on a training seed.
>
> We agree, however, that the manuscript should make the distinction between the deterministic protocol used in the main experiments and the more general stochastic formulation clear. We have revised the abstract and introduction accordingly.
>
> >> **Abstract**: “The resulting certificate is relative to the declared protocol and decision convention.”
>
> >> **Introduction**: “Throughout the paper, ‘verified’ refers to the acceptance procedure used by VERDICT: a proposed design is accepted only after the fixed training-and-evaluation protocol is rerun and the prescribed decision-margin constraints are checked under the stated decision convention. Under a fixed-seed convention, the certificate applies to the declared seed; under an aggregated convention, it applies to the declared finite seed set and aggregation rule. It does not imply preservation uniformly over unseen seeds or outside the declared protocol.”
>
> **1.2. Please (a) state the scope of the certificate in the abstract and introduction rather than only in Appendix C.3, and (b) add an experiment that verifies a VERDICT-selected subset at held-out seeds distinct from the verification seed.**
>
> For the **main direct-acquisition** experiments, the reviewer’s characterization of the results as single-seed stochastic-training certificates does not match the evaluated setting. The main experiments use a **fixed-feature convex ERM** forward protocol:
>
> > “All direct-acquisition experiments use fixed features. Each example $z_i$ is mapped once to $x_i = \phi(z_i)$ before acquisition ... Acquisition therefore changes only the weighted convex ERM objective, not the representation.”
>
> The experimental setup also explicitly reports:
>
> > “VERDICT, HighLoss, GradNorm and DecisionIF are deterministic and have no seed variance.”
>
> The limitations section states the intended consequence directly:
>
> > “Our fixed-feature convex ERM setting isolates decision preservation from representation learning and stochastic training effects.”
>
> The **five seeds** reported in the experimental section are used to summarize **randomized subset-selection baselines**, not repeated retraining of VERDICT-selected subsets. Therefore, in the main direct-acquisition protocol there is no separate verification-training seed and held-out-training seed whose variation could invalidate the certificate. Given a subset, the declared fixed-feature convex ERM protocol defines the retrained candidates and their decision margins. A held-out-seed retraining experiment would consequently not test an additional source of variation in the reported main setting. For genuinely stochastic training pipelines, we agree that **held-out-seed stability is a separate empirical generalization question**; however, it is not part of Proposition 4.11’s protocol-relative guarantee and is not claimed by the current paper.

---

> ### Author Response · Authors · 2026-07-19
> **Response to Reviewer hVv3 (2/6)**
>
> **2. The main claim in Table 1 relies on a baseline pool (Random, Stratified, HighLoss, GradNorm, KCenter, DecisionIF) that excludes the loss-space and coreset methods (loss k-center, k-medoids, Frank-Wolfe mean coreset) used in the downstream study. Please run these stronger baselines in the Table 1 setting, or justify explicitly why they do not transfer to training-subset acquisition.**
>
> Thank you for raising this point. **We initially separated the two baseline sets because direct training acquisition and evaluation-subset selection have different forward maps.** In evaluation-subset selection, the candidate models are fixed, so each example has a fixed vector of candidate losses and subset selection operates on a fixed loss-space geometry or aggregate. **In direct training acquisition, selecting examples changes each candidate's ERM objective; retraining therefore changes the model parameters, the per-example losses, and ultimately the scenario-wise winner margins.** Consequently, covering the geometry or matching the mean of full-data reference-loss vectors does not directly account for the acquisition-induced response of the retrained candidates.
>
> We nevertheless agree that static adaptations of these methods provide informative and stronger comparisons. **We have therefore added loss-space k-center, approximate loss-space k-medoids, and the same FW-inspired equal-weight mean coreset used in our downstream experiment to the direct training-acquisition comparison.** We first train all candidate models on the full training pool and represent each training example by its vector of losses under these full-data reference models. As in the downstream experiment, we center each vector across candidates before applying the selection rules. Loss-space k-center and k-medoids select subsets that cover or represent this fixed loss geometry, while FW-Mean greedily constructs a binary subset whose unweighted mean reference-loss vector approximates the full-pool mean. We then retrain every candidate from scratch on the selected training subset and apply exactly the same scenario-wise winner and margin verification used for VERDICT and all other baselines.
>
> The table below reports the smallest verified retained training fraction found within the evaluated budget grid; lower is better.
>
> | Dataset      | $n_{\mathrm{train}}$ |        Loss-$k$-Center |       Loss-$k$-Medoids |                FW-Mean |   VERDICT |
> | ------------ | -------------------: | ---------------------: | ---------------------: | ---------------------: | --------: |
> | Credit       |               30,000 | $0.930 \pm 0.057$ | $> 0.950$ |              $> 0.950$ | **0.427** |
> | Bank         |               45,211 |      $0.540 \pm 0.055$ |      $0.560 \pm 0.108$ |      $0.780 \pm 0.091$ | **0.413** |
> | Adult        |               48,842 | $> 0.950$ | $> 0.950$ | $0.850 \pm 0.187$ | **0.619** |
> | OxfordPets   |                7,349 |              $> 0.950$ |              $> 0.950$ |              $> 0.950$ | **0.221** |
> | STL10        |                13,000 | $> 0.950$ | $> 0.950$ |      $0.910 \pm 0.022$ | **0.073** |
> | CIFAR10      |               50,000 | $> 0.950$ |      $0.900 \pm 0.050$ | $0.830 \pm 0.115$ | **0.144** |
> | CIFAR100     |               50,000 |      $0.520 \pm 0.091$ |      $0.460 \pm 0.089$ |      $0.530 \pm 0.104$ | **0.020** |
> | 20Newsgroups |               18,846 |      $0.630 \pm 0.027$ |      $0.620 \pm 0.027$ |      $0.800 \pm 0.035$ | **0.458** |
> | AG News      |              127,600 | $> 0.950$ |      $0.950 \pm 0.000$ |      $0.950 \pm 0.000$ | **0.544** |
>
> Randomized methods are reported as mean ± standard deviation over five seeds. A value >0.950 indicates that no seed became feasible within the evaluated grid. The symbol > indicates that at least one seed did not become feasible by the budget of 95% the original data.
>
> **These additional comparisons do not change the main conclusion.** VERDICT achieves a smaller verified retained fraction than each of the three added static loss-space baselines on all nine datasets, **indicating that preserving the geometry or mean of losses under the full-data reference models is generally insufficient to preserve the model-selection decision after retraining**. We have added the complete per-method results to Appendix G.4, and added the best-performing static loss-space result for each dataset to the revised main Table 1.

---

> ### Author Response · Authors · 2026-07-19
> **Response to Reviewer hVv3 (3/6)**
>
> **3. Clarify the logical role of Theorem D.1: The NP-hardness result is invoked to motivate local search, but VERDICT never targets minimum cost and carries no approximation guarantee or optimality-gap analysis, so the theorem does not bear on the quality of the method's output. Please either connect the hardness result to a structural property the method exploits, or state plainly that it is context rather than motivation and adjust the surrounding narrative accordingly.**
>
> We agree that Theorem D.1 does not establish the quality of VERDICT’s output, and we have revised the manuscript to make its role precise. We would first like to clarify that VERDICT does target acquisition cost, but only through a local heuristic rather than by solving or approximating the global minimum-cost problem.
>
> The original formulation in Section 3.2 explicitly defines DPA as a constrained cost-minimization problem:
>
> > “We minimize acquisition cost subject to preserving the selection map with margins and operational constraints.”
>
> The local surrogate used by VERDICT retains this cost objective:
>
> > “A first-order convex surrogate is
> >
> > $$\min_{\alpha,\xi} C(\alpha) + \lambda_{\mathrm{lin}}\lVert \xi \rVert_1$$
> >
> > subject to the linearized decision-margin and trust-region constraints.”
>
> **Thus, VERDICT searches for successively lower-cost designs, but only within a local approximation around the current verified design.** It does not solve the global minimum-cost objective and does not provide an approximation ratio or optimality-gap certificate.
>
> The original manuscript also explicitly separated the feasibility characterization from global cost optimality:
>
> > “**Proposition 4.4** gives a feasibility characterization in the convex ERM case, but it does not imply that the lowest-cost decision-preserving design can be computed efficiently.”
>
> Likewise, the limitations section already characterized VERDICT as a fallible local-search procedure rather than an optimal algorithm:
>
> > “VERDICT trades acquisition or evaluation savings for search cost, and as a local search method can fail when the response model mispredicts margins, rounding breaks feasibility, or the winner map depends on broad coverage rather than near-active comparisons.”
>
> We agree, however, that our use of the word “motivate” blurred two logically distinct roles. The KKT and sensitivity analysis provides the structural basis for VERDICT’s local decision-response operator. The NP-hardness result instead characterizes the computational difficulty of the ideal global cost-minimization problem. It explains why we do not design the method around a general efficient exact solver, but the NP-hardness result itself neither determines VERDICT’s particular local-search construction nor establishes the quality of the designs returned by VERDICT. We have revised the manuscript accordingly.
>
> > **Abstract:** “In an idealized convex ERM regime, DPA admits a Karush–Kuhn–Tucker feasibility view, while computing an exact minimum-cost design is NP-hard even in a simplified setting. These results characterize the structure and computational difficulty of the idealized inverse problem.”
>
> > **Contribution summary:** “**Structural context.** In an idealized convex ERM regime, decision preservation can be expressed through a KKT feasibility system, and computing an exact minimum-cost decision-preserving design is NP-hard. The KKT characterization provides the basis for the local response analysis, while the hardness result provides computational context for the ideal global objective. Neither result implies that VERDICT is globally or approximately optimal.”
>
> > **Discussion around Theorem D.1:** “**Role of the hardness result.** Theorem D.1 concerns the ideal global DPA objective: finding the minimum-cost design satisfying all decision-preservation constraints. It shows that this problem is NP-hard even in a simplified convex ERM setting. VERDICT does not solve this global problem and carries no approximation ratio or optimality-gap guarantee. Instead, it uses a local cost-bearing surrogate to propose lower-cost candidate designs, while direct retrain-and-check verification establishes only the feasibility of accepted designs.”
>
> We now state the logical relationship explicitly: the ideal DPA problem minimizes cost subject to decision preservation; VERDICT heuristically pursues local reductions in that same cost; and Theorem D.1 provides context for why globally exact optimization is difficult, not a guarantee on VERDICT’s returned solution.

---

> ### Author Response · Authors · 2026-07-19
> **Response to Reviewer hVv3 (4/6)**
>
> **4. Isolate whether the decision-response signal or the verified loop drives performance.**
>
> We agree that the original ablations did not completely isolate the contribution of the decision-response operator from that of the verified iterative wrapper.
>
> VERDICT uses the local decision-response model to generate candidate designs, while feasibility is determined only by rerunning the forward protocol. Section 4.4 states:
>
> > “In the general training setting, the convex surrogate acts as a proposal mechanism. We certify candidates by rerunning the pipeline and verifying margin constraints under the chosen convention, ensuring accepted designs satisfy the requirements.”
>
> Proposition 4.11 formalizes the certification role:
>
> > “Any algorithm accepting a candidate design only after retraining and verifying margin constraints yields decision-feasible designs.”
>
> Thus, the decision-response operator is meant to guide the search toward promising deletions, while direct execution of the forward protocol determines whether a proposal is accepted. The original Table 2 already demonstrates why verification cannot be replaced by the surrogate:
>
> > “The rejected-proposal columns show why retrain-and-check verification is necessary. In every dataset, the surrogate eventually proposes a candidate with positive predicted margin that becomes infeasible after retraining, sometimes with winner flips.”
>
> The reviewer correctly notes, however, that **DecisionIF** and **RandomWS** do not fully isolate these two roles. DecisionIF is a one-shot baseline computed at the full-data solution. Unlike VERDICT, it does not relinearize around accepted subsets, use rejection feedback, or adapt its proposal radius. RandomWS tests a narrower component. As stated in Appendix G.3:
>
> > “VERDICT-RandomWS keeps the same surrogate, trust-region, and verification machinery, but replaces response-guided working-set construction with uniformly random working sets of comparable size.”
>
> RandomWS's advantage on Adult shows that the deterministic working-set ordering is not uniformly optimal. This dataset dependence was already acknowledged in the original manuscript:
>
> > “Thus, Table 11 does not suggest a single universally optimal local surrogate; rather, it shows that the DPA objective can be approached by several verified proposal mechanisms, whose effectiveness depends on the structure of the scenario margins.”
>
> To isolate the decision-response mechanism, we added a matched Verified-NoDR control. It keeps VERDICT’s initialization, proposal budget, retraining, acceptance rule, adaptive radius schedule, and stopping criterion, but removes all decision-response information. Proposals are accepted only after passing the same retrain-and-check verification.
>
> | Dataset      | Full VERDICT retained | Verified-NoDR retained | NoDR Accept / Reject | NoDR final verified margin | NoDR first rejected margin, surrogate → verified | NoDR Viol. / Flip |
> | ------------ | --------------------: | ---------------------: | -------------------: | -------------------------: | -----------------------------------------------: | ----------------: |
> | Credit       |             **0.427** |          0.891 ± 0.075 |           15.6 / 9.8 |                3.86 × 10⁻⁶ |                               N/A → −2.27 × 10⁻⁴ |         4.6 / 2.4 |
> | Adult        |             **0.619** |          0.736 ± 0.097 |          18.2 / 10.0 |                1.33 × 10⁻⁶ |                               N/A → −1.55 × 10⁻⁵ |         1.2 / 0.8 |
> | STL10        |             **0.073** |          0.728 ± 0.152 |           18.4 / 9.4 |                1.43 × 10⁻⁴ |                               N/A → −5.13 × 10⁻⁴ |         1.0 / 0.6 |
> | CIFAR100     |             **0.020** |          0.389 ± 0.086 |           21.4 / 8.6 |                1.16 × 10⁻⁴ |                               N/A → −3.02 × 10⁻⁴ |         1.0 / 0.0 |
> | 20Newsgroups |             **0.458** |          0.729 ± 0.028 |          16.4 / 10.4 |                2.92 × 10⁻⁹ |                               N/A → −6.42 × 10⁻⁵ |         1.4 / 0.2 |
> | AG News      |             **0.544** |          0.918 ± 0.015 |            6.8 / 8.2 |                1.52 × 10⁻⁸ |                               N/A → −8.82 × 10⁻⁵ |         1.0 / 0.0 |
>
> Notes. Lower is better; the best result in each row is bolded. Verified-NoDR results are means over five seeds, with sample standard deviations for retained fraction. N/A indicates that NoDR uses no decision-response surrogate.
>
> All accepted NoDR designs pass the same verification check, showing that verification alone preserves feasibility. However, Full VERDICT still achieves a lower retained fraction on all six datasets under the same search budget. This isolates the two roles: **verification establishes feasibility, while the decision-response operator improves search and compression efficiency**. We added the full trajectory diagnostics to Appendix G.3.

---

> ### Author Response · Authors · 2026-07-19
> **Response to Reviewer hVv3 (5/6)**
>
> **5. Exercise the multi-quality cost model or scope it down: Section 3 develops quality levels, per-point overheads, and a general cost function, but every experiment sets K=1 with unit cost, so cost equals subset size throughout.**
>
> Thank you for identifying this mismatch. We have taken the scope-down option and revised the paper so that its formal, algorithmic, and empirical scope is fixed-quality subset-size preservation. The revised paper no longer develops a multi-quality cost model as part of the present contribution.
>
> In Section 3, we now define the design directly as a binary inclusion vector $\alpha \in \{0,1\}^N,$ with retained cardinality $|\alpha| = \sum_i \alpha_i,$ and use cardinality-based feasible sets. We removed the quality-level set, quality-indexed variables, one-quality-per-example constraints, quality-dependent costs, and point-specific overheads from the main formulation. The revised scope is stated explicitly:
>
> > “The fixed-quality inclusion model is the setting studied throughout this paper. A natural extension would allow an evidence unit to be acquired at one of several annotation, measurement, preprocessing, or fidelity levels with different costs; jointly selecting evidence units and their quality levels is left for future work.”
>
> We also reformulated the inverse objective as minimum-cardinality decision preservation:
>
> > “The ideal fixed-quality DPA problem minimizes retained subset cardinality subject to preserving the baseline selection map with margins and operational constraints.”
>
> This scope change is propagated through the technical development. The convex ERM objective now uses one inclusion weight $\alpha_i$ per example; the response and decision-response operators act on perturbations in $\mathbb{R}^N$; and the local surrogate minimizes retained cardinality rather than a general acquisition-cost function. We similarly revised Algorithm 1, the verification checks, Figure 1, the notation tables, and the relaxation semantics to use subset-size constraints.
>
> The complexity result has also been aligned with the scoped problem. The revised theorem establishes NP-hardness of computing an exact minimum-cardinality decision-preserving subset through a reduction from Minimum Set Cover, without relying on heterogeneous costs or multiple quality levels.
>
> Finally, the experimental section no longer describes the evaluation as a specialization of a broader multi-quality formulation. It now states directly:
>
> > “All experiments study fixed-quality subset selection with unit cost per retained example; thus, the objective is retained subset size.”
>
> Multi-quality or quality-aware acquisition is retained only as a future extension, not as a developed or empirically supported contribution of the present paper. These revisions align the problem statement, theory, algorithm, complexity result, and empirical claims with the subset-size preservation setting that is actually evaluated.
>
> **6. Reconcile the two scenario-discovery tables: Table 3 (main text) and Table 13 (appendix) report overlapping scenario-discovery results for the same datasets with differing presentation and some differing values. Please make the relationship explicit or consolidate them, so the reader is not left comparing two versions of the same experiment.**
>
> We have verified that the overlapping entries in Table 3 and Table 13 are numerically identical; Table 3 is meant to be a selected subset of the full results reported in Table 13 due to space constraint. The only difference arises from presentation, including row ordering and the explicit display of positive signs. We agree that the current formatting can make the tables appear to report different versions of the experiment. In the revision, we have reconciled the notation, styling, row order, and captions for these two tables.

---

> ### Author Response · Authors · 2026-07-19
> **Response to Reviewer hVv3  (6/6)**
>
> **7. Report search cost against the cost of the evidence saved: The method is motivated by the expense of acquiring and retaining data, yet VERDICT's own runtime is dominated by proposal construction and reaches 2128 seconds on CIFAR100 (Table 24).**
>
> Thank you for requesting this comparison. The submitted manuscript already notes in **Section 6, Scope and limitations** that
> >"VERDICT trades acquisition or evaluation savings for search cost”
>
> and identified proposal construction as the dominant computational bottleneck. It also motivated compact subsets in settings where evidence is repeatedly processed, including repeated comparisons of models, hyperparameters, and checkpoints. The original analysis, however, did not quantify this tradeoff by comparing VERDICT’s one-time search cost with the runtime of the full and selected-subset protocols. We have added that direct comparison below.
>
> | Dataset      | Retained fraction | Search cost | Full protocol | Subset protocol | Runtime saved/run | Per-run reduction | Speedup |
> | ------------ | ----------------: | ----------: | ------------: | --------------: | ----------------: | ----------------: | ------: |
> | Credit       |             0.427 |      4.22 s |      0.0249 s |        0.0112 s |          0.0137 s |             54.9% |   2.22× |
> | Bank         |             0.413 |      2.38 s |      0.0190 s |        0.0091 s |          0.0099 s |             52.0% |   2.08× |
> | Adult        |             0.619 |      8.44 s |      0.0457 s |        0.0338 s |          0.0119 s |             26.1% |   1.35× |
> | OxfordPets   |             0.222 |    138.94 s |      0.8909 s |        0.8227 s |          0.0682 s |              7.7% |   1.08× |
> | STL10        |             0.073 |     90.14 s |      0.9240 s |        0.9563 s |         −0.0323 s |             −3.5% |   0.97× |
> | CIFAR10      |             0.144 |    467.31 s |      1.5926 s |        1.0633 s |          0.5293 s |             33.2% |   1.50× |
> | CIFAR100     |             0.020 |  2,127.96 s |      1.6944 s |        1.1567 s |          0.5377 s |             31.7% |   1.46× |
> | 20Newsgroups |             0.458 |    129.64 s |      0.5707 s |        0.3483 s |          0.2224 s |             39.0% |   1.64× |
> | AG News      |             0.544 |    285.11 s |      2.2802 s |        2.3176 s |         −0.0373 s |             −1.6% |   0.98× |
>
> The selected subsets reduce subsequent protocol runtime on seven of the nine datasets, with substantial relative reductions on the tabular datasets, CIFAR10, CIFAR100, and 20Newsgroups. OxfordPets shows a smaller reduction, while STL10 and AG News show no reduction under the recorded timings. These latter cases indicate that, when the underlying protocol is already inexpensive, fixed execution overhead and solver-level variation can dominate the effect of reducing the training set.
>
> At the same time, the table shows that the one-time search cost is considerably larger than a single execution of either the full or reduced protocol, particularly for the image datasets. The runtime decomposition further identifies response scoring and proposal construction, rather than retraining-based verification, as the dominant source of this overhead.
>
> This analysis clarifies an important feature of our experimental setting. The direct-acquisition experiments use fixed representations and inexpensive convex ERM protocols so that many candidate designs can be retrained and verified during the search. As a result, even a large reduction in retained evidence can correspond to only a modest absolute reduction in execution time.
>
> The comparison therefore separates three notions of efficiency. VERDICT substantially reduces the evidence required to reproduce the declared decision; the resulting subsets often make subsequent protocol executions cheaper; but the present experiments do not establish an end-to-end wall-clock advantage for one-off use. Such an advantage is more likely when the forward protocol is more computationally expensive or when the selected evidence is reused across repeated model, checkpoint, or hyperparameter comparisons.
>
> We revised the manuscript accordingly to distinguish evidence compression and per-execution runtime reduction from overall search-time efficiency, and to identify proposal construction as the primary computational bottleneck of the current implementation.

---

### Review · Reviewer_aKob · 2026-07-07

**Summary Of Contributions:**

This paper investigates Decision-Preserving Acquisition (DPA), the problem of selecting compact training or evaluation evidence that preserves the outcome of a fixed model-selection protocol. The goal is to find the minimal fraction of the data that can be kept while keeping the top model for each validation scenario(task) the same within a certain margin. This work formalizes this task mathematically, and provides a structural analysis under an idealized strongly convex ERM setting. Under these, and some additional, assumptions, the authors show that DPA can be solved as the feasibilyty of a Karush-Kuhn-Tucker system, while the associated problem remains NP-hard. To this end, it proposes `VERDICT`, a method that uses an implicit-differentiation response model to propose lower-cost data subsets (designs) that satisfy the margin constraints.

The paper primarily contributes with the problem framing and to some extent, the theoretical framework. Narrowing down model selection to preserving a discrete winner map is interesting, and simpler, and can be useful for future work. The connection to convex optimization and the theoretical restrictions of the problem (NP-hardness) are also important for understanding the difficulty of the investifated problem. The verification-based design is also sensible, as the local surrogate is used only to propose candidates, while feasibility is certified by direct retrain-and-check evaluation.

On the other hand, the paper has some important weaknesses. First of all, the DPA problem is not sufficiently motivated or grounded in practice. Experimentally, the paper is not yet fully convincing. Several important details are underspecified, including the candidate model pools, exact scenario definitions, hyperparameters, solver tolerances, and numerical verification thresholds. The baseline comparison are relatively simple, and do not sufficiently cover prior work.

Overall, I find the problem interesting and potentially relevant, but believe the current submission requires some clarification and experimental revision that can be resolved during the discussion period.

**Audience:**

Yes

**Audience Explanation:**

The paper should be of interest to at least some parts of the TMLR audience, especially researchers working on optimization, model selection, data subset selection and pruning, with some connections to applications of influence functions.

The main reason is that the paper proposes a distinct target for data selection. Most data subset-selection methods try to preserve training performance, validation performance, gradients, losses, representations, or data coverage. This work instead targets preservation of a discrete model-selection decision. In many practical workflows, the downstream action is indeed a decision where a single model wins, be it in a hyperparameter search, checkpoint selection, or architecture decision task.

The mathematical connection to KKT systems, MPECs, and implicit differentiation is not shocking, but it is a reasonable way to motivate the algorithm.

**Broader Impact Concerns:**

None.

**Claims And Evidence:**

No

**Claims Explanation:**

If given the option, I would have put the above answer as "Partially". In particular I list some clear aspects, and some methodological concerns on the empirical evaluation and problem framing I have below:

## Valid aspects

1. Mathematically, the theoretical contribution is, to the best of my assessment, sound. The derivations are rigorous under the stated assumptions, including the implicit differentiation and the NP-hardness reduction proof.

2. The evaluation of `VERDICT` is performed on multiple modalities, including tabular, vision, and text datasets.

3. The baseline comparisons show that `VERDICT` significantly outperforms standard baselines, showing its usefulness over uninformed approaches.

4. The diagnostic results in Table 2 are also useful. Random subsets at `VERDICT`'s final retained fraction are often infeasible, and rejected proposals frequently have positive predicted margins but negative verified margins after retraining, which supports the paper's claim that local response models are useful but insufficient without verification.

## Methodological concerns

1. The paper does not, in sufficient detail discuss why investigating DPA is beneficial. The most simple and standard argument would include that it can provide computational alleviation. However, as `VERDICT` performs at least 1 retraining attempt for validation, it is unlikely that the computational benefit is significant. Furthermore, the authors themselves concede that the results are valid only under the given setting, with models and tasks fixed. Therefore, the authors must include the following aspects:
    -  Appendix I reports runtime breakdowns, but the paper still lacks a comparison to full validation/training and to the corresponding baselines.
    - If the gain is negative/insignificant, what are the settings in which `VERDICT` can be most useful.
    - If the authors can show that the selected subset can reasonably transfer to other settings, it could also be a benefit.

2. The baseline set that `VERDICT` is evaluated against is relatively limited. While I do not argue against the potential usefulness of the proposed method, there exists prior works, such as [1] that already develop methods for data subset selection even if under different settings. It would be useful to see how they perform compared to `VERDICT` even if they are mostly targeting a different objective compared to the winner map preservation.

3. The experimental setting seems underspecified. The winner map depends critically on the candidate model pool, hyperparameters, scenario definitions, and margins. Yet the paper does not provide enough detail about the exact candidate models used in the direct-acquisition experiments, the precise scenario thresholds for tabular datasets, the exact class groupings for vision datasets, or the numerical tolerances used to declare margin feasibility. Some margins are small, i,e around $10^{−8}$, so precision becomes important.

4. The results shown in Table 11 show that the full `VERDICT` is not necessarily the best performing variant in different regimes. While the full `VERDICT` is consistently decent across all settings, the results somewhat pose into question the robustness of the algorithm if removing components increases performance in half the cases.

5. Tables 15,16,18,21 seem to contain some inconsistencies between themselves:
    - Table 15 reports at budget $0.5\%$ results on ResNet50 representations. All of the reported results for `VERDICT` are lower than the reported macro average in Table 16.
    - Table 18 states CIFAR100/ResNet50 needs 2% to reach $\tau \geq 90\%$, but Table 15 already reports 0.935 at 0.5%.
    - Results on `VERDICT` between Tables 15 and 21 seem to be different.

6. (Minor) At a couple of points, i.e. the Table 1 caption, the paper states that it measures "Smallest retraining-verified retained fraction needed", while all methods actually measure approximations to this end, and not a global minimum.

Overall, my concerns with the paper lie primarily with the empirical evaluation of `VERDICT` and the positioning of DPA. I do believe most of these issues can be addressed, and am happy to discuss if I have misunderstood/missed some of these aspects.

### References

[1] Killamsetty, Krishnateja, et al. "Automata: Gradient based data subset selection for compute-efficient hyper-parameter tuning." Advances in Neural Information Processing Systems 35 (2022): 28721-28733.

**Requested Changes:**

I list several recommendations that I believe can improve the paper.

- Please include wall-clock-time comparisons between `VERDICT` and proposed alternatives. A discussion on asymptotic time complexity might also be beneficial.

- It will be useful to preface the assumptions for the problem framing and `VERDICT` somewhere, so that it is clearer which properties rely on which assumptions.

- Please add a stronger baseline experiment derived from stronger prior work.

- Please resolve the inconsistencies between tables 15, 16, 18, 21.

- To strengthen the empirical contribution, it could be useful to check whether the properties of this subset transfer over to different models, i.e. if using a similar, but smaller model to determine the data subset can produce decent validation performance on larger models. I understand a lot of this paper's contributions lie in theoretical framing, so this is only an added bonus that can increase the potential impact, and I am not strongly requesting the authors perform this experiment.

- Please discuss (at least briefly) any remaining concerns from the review.

---

> ### Author Response · Authors · 2026-07-19
> **Response to Reviewer aKob (1/4)**
>
> **Q1.1. The paper does not, in sufficient detail discuss why investigating DPA is beneficial. The most simple and standard argument would include that it can provide computational alleviation. However, as VERDICT performs at least 1 retraining attempt for validation, it is unlikely that the computational benefit is significant. Furthermore, the authors themselves concede that the results are valid only under the given setting, with models and tasks fixed.**
>
> The intended motivation for DPA is broader than reducing the cost of a single training run. **The original manuscript already describes DPA primarily as an auditing and evidence-identification framework**, while recognizing that its computational value depends on the application.
>
> The Impact Statement states explicitly:
>
> > “This work develops an auditing-oriented tool: it can identify low-cost training evidence sufficient to reproduce an observed model-selection map under explicit margins and a specified decision convention.”
>
> and:
>
> > “This can support transparency in data curation and quantify how much evidence is needed to support a deployed selection decision.”
>
> **In this role, train-and-verify DPA is useful even when its search is not cheaper than a single full-data training run.** It identifies which acquired examples constitute sufficient evidence for a declared model-selection outcome, which can support data auditing, retention decisions, and analysis of which comparisons constrain the selected model.
>
> **The paper also studies decision-critical scenario discovery.** Here, the purpose is to detect scenarios omitted from the declared protocol on which the selected subset no longer preserves the original decision. Such failures reveal additional scenarios that should be included in the model-selection specification.
>
> **The most direct computational benefit arises in the evaluation-subset application.** The original manuscript motivates this setting as follows:
>
> > “Model selection can be expensive even after candidate models have been trained, because each candidate must be evaluated on a potentially large validation set.”
>
> The objective is therefore to:
>
> > “select a small subset of validation examples whose aggregate losses induce the same model-selection conclusion as the full validation set.”
>
> In this setting, the candidate models are fixed, and the selected evaluation subset can be reused across repeated comparisons of models, hyperparameters, or checkpoints. Its construction cost can therefore be amortized, particularly when evaluation is repeated or itself expensive.
>
> The original manuscript also states the relevant tradeoff directly:
>
> > “VERDICT trades acquisition or evaluation savings for search cost.”
>
> Accordingly, **we do not claim that computational alleviation is automatic in the direct train-and-verify setting.** VERDICT incurs additional cost through its iterative proposal search. The retraining check, however, is the operational test required to determine whether any proposed training subset actually preserves the original model-selection decision; all methods in our direct-acquisition experiments are therefore assessed using the same retrain-and-check feasibility criterion. VERDICT’s additional cost is its iterative search, while its empirical benefit is that it often finds substantially smaller decision-feasible subsets than one-shot sampling, coverage, difficulty, or influence-based methods.
>
> We have added a dedicated motivation paragraph to the Introduction describing three uses:
>
> - identifying compact evidence that supports an observed model-selection decision, for auditing and retention;
> - detecting omitted decision-critical scenarios; and
> - reducing repeated evaluation cost through compact evaluation subsets for a declared candidate family.
>
> **We also clarify that computational alleviation is application-dependent.** Train-and-verify DPA trades search cost for acquisition, retention, and auditing benefits, whereas evaluation-subset DPA can provide direct savings when its construction cost is amortized over repeated or expensive comparisons.
>
> Finally, protocol relativity is intrinsic to this objective rather than an incidental limitation. As the original manuscript states:
>
> > “DPA is protocol-relative: its guarantees apply only to the specified candidate pool, scenarios, margins, costs, and decision convention.”
>
> DPA is not intended to identify a subset sufficient for every possible future task or model family. It identifies and verifies the evidence supporting a specified model-selection decision under a documented protocol.

---

> ### Author Response · Authors · 2026-07-19
> **Response to Reviewer aKob (2/4)**
>
> **Q1.2. Appendix I reports runtime breakdowns, but the paper still lacks a comparison to full validation/training and to the corresponding baselines.**
>
> DPA reduces the evidence needed to reproduce a declared model-selection outcome but incurs an upfront search and verification cost. We therefore added a method-level comparison of total end-to-end runtime for VERDICT and every baseline. For each baseline, total runtime includes subset selection and retrain-and-check verification; for VERDICT, it includes the complete verified search procedure.
>
> **Runtime comparison in seconds.** Random and Stratified report mean ± standard deviation of total runtime, where total runtime is `select_time_sec + retrain_verify_time_sec`. VERDICT reports end-to-end search runtime.
>
> | Dataset      |        Random |    Stratified | HighLoss | GradNorm | KCenter | DecisionIF |  VERDICT |
> | :----------- | ------------: | ------------: | -------: | -------: | ------: | ---------: | -------: |
> | Credit       | 0.049 ± 0.004 | 0.050 ± 0.001 |    0.103 |    0.129 |   8.460 |      0.125 |    4.220 |
> | Bank         | 0.015 ± 0.003 | 0.025 ± 0.001 |    0.098 |    0.137 |   5.736 |      0.106 |    2.380 |
> | Adult        | 0.068 ± 0.006 | 0.065 ± 0.001 |    0.182 |    0.189 |  10.139 |      0.246 |    8.440 |
> | OxfordPets   | 1.663 ± 0.035 | 1.593 ± 0.217 |    2.350 |    2.395 |   1.468 |      8.157 |  138.940 |
> | STL10        | 1.126 ± 0.073 | 1.151 ± 0.061 |    1.822 |    2.434 |   1.767 |      4.184 |   90.140 |
> | CIFAR10      | 1.328 ± 0.058 | 1.187 ± 0.028 |    3.683 |    4.420 |   5.106 |     24.603 |  467.310 |
> | CIFAR100     | 1.533 ± 0.128 | 1.413 ± 0.058 |    4.599 |    5.276 |   2.522 |    241.928 | 2127.960 |
> | 20Newsgroups | 0.540 ± 0.013 | 0.520 ± 0.074 |    1.001 |    1.048 |   1.523 |      4.756 |  129.640 |
> | AG News      | 2.358 ± 0.007 | 2.318 ± 0.009 |    4.790 |    6.663 |   9.688 |     10.681 |  285.110 |
>
> The measured runtimes show that VERDICT is not intended as a minimum-latency selector, but its complete search remains practical on most datasets. Moreover, on eight of the nine datasets, proposal construction and decision-response scoring take more time than the combined incumbent and proposed-design retrain-and-verification steps. Thus, the verification check is generally not the primary computational bottleneck. We therefore present the relevant empirical comparison as a runtime–verified-compression tradeoff rather than claiming that VERDICT is always the fastest subset constructor.
>
> **Q2. The baseline set that VERDICT is evaluated against is relatively limited. While I do not argue against the potential usefulness of the proposed method, there exists prior works, such as [1] that already develop methods for data subset selection even if under different settings.**
>
> AUTOMATA [1] reduces hyperparameter-optimization cost through gradient-based subset selection, but its published protocol uses configuration-specific weighted subsets that are updated during training. **Running it unchanged would therefore allow different candidate configurations to receive different examples and weights, rather than evaluating one common acquired subset**. Conversely, forcing a single fixed binary subset would remove AUTOMATA’s configuration-specific weighting and adaptive reselection, yielding a new adaptation rather than the published method.
>
> Our direct-acquisition setting **fixes the candidate configurations and the training/evaluation protocol in advance.** It **requires a single common acquisition design that is used to retrain every candidate, and evaluates whether this shared design preserves the complete scenario-wise winner map with prescribed margins**. The acquisition cost is therefore defined for one common evidence set. Allowing configuration-specific, adaptive weighted subsets, as in AUTOMATA, **would change the feasible design space, the training protocol, and the meaning of acquisition cost**.
>
> We have therefore not included AUTOMATA as a direct entry in the main comparison table, and **instead expanded the related-work discussion to explain the connection and distinction explicitly**. We also strengthened the direct-acquisition comparison with loss-space k-center, approximate loss-space k-medoids, and FW-inspired mean matching, all of which admit well-defined binary shared-subset adaptations without changing the basic identity of their selection objectives. These additions directly address whether stronger static subset-selection criteria explain VERDICT’s performance while preserving the common-design requirement of DPA.
>
> [1] Killamsetty, Krishnateja, et al. "Automata: Gradient based data subset selection for compute-efficient hyper-parameter tuning." Advances in Neural Information Processing Systems 35 (2022): 28721-28733.

---

> ### Author Response · Authors · 2026-07-19
> **Response to Reviewer aKob (3/4)**
>
> **Q3. The results shown in Table 11 show that the full VERDICT is not necessarily the best performing variant in different regimes. While the full VERDICT is consistently decent across all settings, the results somewhat pose into question the robustness of the algorithm if removing components increases performance in half the cases.**
>
> We agree that Table 11 shows that full VERDICT is not uniformly the lowest-retained-fraction variant. This reflects the **non-monotonic nature** of the verified deletion search: **changing one proposal component can move the method onto a different feasible trajectory and occasionally produce a smaller subset**. Importantly, this affects compression efficiency rather than correctness, since all verified variants retain the same retrain-and-check acceptance rule.
>
> The clearest component-level finding is that verification is essential. NoVerify follows the surrogate path without retraining feedback, but its post-hoc verified subsets remain poor, showing that local response estimates are useful for proposing deletions but do not certify preservation of the retrained decision. The other ablations instead modify how verified proposals are constructed. RandomWS replaces response-guided ordering with random exploration and can therefore discover feasible paths that the deterministic score initially underranks. AllConstraints replaces the quasi-active margin set with every non-winner comparison, providing broader information but increasing cost and potentially restricting proposals through constraints that are not currently close to binding.
>
> To distinguish these mechanisms, Table 13 reports the final retained fraction, the number of surrogate constraints, the mean fraction removed by accepted proposals, the optimism gap between surrogate-predicted and retraining-verified minimum margins on rejected proposals, and the score-construction time per proposal. These quantities separate final compression from the proposal size, local surrogate fidelity, constraint scope, and computational cost that produce it.
>
> The results reveal three regimes. On Credit, STL10, and CIFAR100, quasi-active VERDICT reaches the smallest verified subset while using only 4--8 constraints and substantially less scoring time; the full-constraint surrogate is often better calibrated but does not translate that calibration into better compression. On 20Newsgroups and AG News, broader constraint information produces more effective accepted deletions and improves the final subset, although at considerably higher cost. Adult is different: RandomWS performs best, indicating that proposal ordering, rather than constraint coverage, determines the favorable trajectory. Overall, the table supports quasi-active filtering as an efficient default while showing that neither broader constraint coverage nor more accurate local margin prediction guarantees a better deletion path. We will add this diagnostic analysis to the revised manuscript as Table 13, since it more directly explains when and why the ablated proposal rules outperform the default VERDICT trajectory.
>
> **Table 13: Comparison of default VERDICT, RandomWS, and AllConstraints.** Accepted step is the mean percentage of the current subset removed by accepted proposals. Optimism gap is the mean difference between the surrogate-predicted and retraining-verified minimum margins over rejected proposals. Results are averaged across seeds $\in$ {0,1,2,3,4}.
> |Dataset|Method|Retained↓|Constraints↓|Accepted%↑|Gap|Time(s)↓|
> |---|---|---:|---:|---:|---:|---:|
> |Credit|VERDICT|**0.427**|**5**|**3.62**|1.72e-4|**.12**|
> ||RandomWS|0.767|**5**|1.23|**1.18e-5**|.12|
> ||AllConstraints|0.857|48|.73|1.99e-5|1.01|
> |Adult|VERDICT|0.619|**5**|2.16|2.11e-4|**.16**|
> ||RandomWS|**0.431**|**5**|**3.73**|2.93e-5|.16|
> ||AllConstraints|0.777|48|1.13|**6.89e-6**|1.43|
> |STL10|VERDICT|**0.073**|**8**|**10.61**|1.31e-2|**.96**|
> ||RandomWS|.347|**8**|4.67|**1.87e-4**|2.06|
> ||AllConstraints|0.101|80|10.31|2.85e-3|9.41|
> |CIFAR100|VERDICT|**0.020**|**8**|**15.54**|1.48e-2|**67.92**|
> ||RandomWS|0.301|**8**|5.79|**6.27e-4**|130.73|
> ||AllConstraints|0.033|80|13.53|1.64e-3|618.48|
> |20Newsgroups|VERDICT|0.458|**5**|3.48|2.55e-4|**4.15**|
> ||RandomWS|0.735|**5**|2.31|2.37e-5|4.53|
> ||AllConstraints|**0.441**|48|**4.14**|**5.39e-6**|34.43|
> |AG News|VERDICT|0.544|**4**|3.11|3.35e-4|**5.76**|
> ||RandomWS|0.897|**4**|1.01|**3.29e-5**|7.72|
> ||AllConstraints|**0.133**|32|**7.79**|3.68e-4|37.36|

---

> ### Author Response · Authors · 2026-07-19
> **Response to Reviewer aKob (4/4)**
>
> **Q4. Tables 15,16,18,21 seem to contain some inconsistencies between themselves.**
>
> Thank you for identifying this. **The apparent inconsistencies come from two distinct evaluation settings that were labeled too similarly, plus one redundant aggregate. Table 15 reports at budget 0.5 results on ResNet50 representations. All of the reported results for VERDICT are lower than the reported macro average in Table 16. Table 18 states CIFAR100/ResNet50 needs 2% to reach $\tau \geq 90$, but Table 15 already reports 0.935 at 0.5%. Results on VERDICT between Tables 15 and 21 seem to be different.**
>
> **The apparent discrepancy between Tables 15 and 21 reflects two distinct evaluation settings**. Table 15 reports the main evaluation-subset benchmark using a fixed ResNet50 representation and its associated $N \times M$ candidate-score matrix. Tables 19–22 report a separate backbone-sensitivity study, where each backbone defines an independently constructed $M=20$ model-selection instance; Table 21 is the ResNet50 instance from that study. Because the full reference ranking is determined by the candidate-score matrix, the two settings can yield different Kendall-$\tau$ curves even when they use the same budgets and metric. The submitted version did not distinguish these candidate pools and experimental roles clearly enough, which made the two ResNet50 results appear directly comparable. We have revised the captions and surrounding text to make this distinction explicit.
>
> **Table 18 was computed from the auxiliary per-backbone results in Tables 19–22**. Consequently, its CIFAR100/ResNet50 entry of $2\%$ refers to the ResNet50 model-selection instance reported in Table 21, not to the main ResNet50 experiment in Table 15. It therefore does not conflict with the $0.935$ Kendall-$\tau$ reported at $0.5\%$ in Table 15.
>
> Table 16 had a separate presentation issue. **It aggregated a broader collection of evaluation results than the datasets displayed in Table 15, but that broader scope was not stated clearly**. Because this macro average was redundant with the more informative per-dataset results and supported no unique conclusion, we have removed Table 16 and the associated macro-level claim.
>
> We have revised the manuscript as follows.
>
> **First, we revised the caption of Table 15 to identify its experimental role and distinguish it from the backbone-sensitivity study:**
>
> >> **Table 15:** Detailed evaluation-subset results for the main ResNet50 benchmark. For each dataset, the full evaluation set defines the reference ranking of a fixed $M=20$ candidate-model pool evaluated using frozen ResNet50 representations. Each method selects a subset of evaluation examples at the indicated budget, and Kendall’s $\tau$ compares the ranking induced by the subset with the full-set ranking. Entries report mean $\pm$ standard deviation over five subset-selection seeds.
>
> **Second, we revised Table 18 to state explicitly that it summarizes the separate backbone-sensitivity experiment:**
>
> >> **Table 18:** Minimum evaluation budget required for VERDICT to reach mean Kendall’s $\tau \geq 0.9$ in the backbone-sensitivity study. Each dataset–backbone pair defines an independent model-selection instance with a separately constructed $M=20$ candidate pool on the corresponding frozen representation. Each cell reports the smallest evaluated budget in {0.5%, 1%, 2%, 5%, 10%, 20%} at which the mean Kendall’s $\tau$ over five subset-selection seeds reaches $0.9$. All values are compared and derived from the per-backbone experiments reported in Tables 19–22.
>
> **Third, we revised the caption of Table 21 to clarify that it is one component of the auxiliary backbone-sensitivity study:**
>
> >> **Table 21:** ResNet50 results within the backbone-sensitivity study. This experiment constructs an independent $M=20$ candidate-model pool on frozen ResNet50 representations and evaluates how well selected evaluation subsets preserve its full-set ranking. The results belong to the same per-backbone study as Tables 19, 20, and 22. Randomized methods are reported as mean $\pm$ standard deviation over five subset-selection seeds.
>
> We also added the following clarification before the backbone-sensitivity results:
>
> >> We report two distinct ResNet50 evaluation settings. Table 15 uses the ResNet50 candidate pool from the main baseline-comparison experiment. Tables 18–22 report a separate backbone-sensitivity study in which an $M=20$ candidate pool is constructed independently for each frozen representation backbone.
>
> The underlying runs are unchanged. The revision corrects the experimental labeling in Tables 15, 18, 21, and removes the redundant aggregate in Table 16 that made distinct model-selection instances appear directly comparable.

---

> ### Comment · Reviewer_aKob · 2026-07-20
>
> Thank you for the detailed rebuttal, the clarification about the tables in Appendix and the further reports. I now believe I have a better understanding of the paper, and list below what I consider resolved, and what remains partially or insufficiently addressed.
>
> ## Fully Addressed
>
> Q3. The new table produces a valuable account on why RandomWS or AllConstraints occasionally beat VERDICT, and the arguments seem reasonable.
>
> Q4. The explanation for the inconsistencies make sense, thank you for the clarification! I will expect that the captions appear in the next revision of the paper.
>
> ## Partially Addressed
>
> Q1.2: The end-to-end runtime table is genuinelly helpful, but it only compares the search-based method to each other. It still omits the cost of simply of doing the full-data train-verify loop (no search). Without that row, I cannot tell whether the costs presented are cheap or expensive.
>
> ## Further Concerns
>
> Re-reading the problem setup in light of the rebuttal, I want to request further discussion from the authors with respect to the core motivation of the paper.
>
> 1. The formal problem requires the winner map and the margins to already be computed from a full-data run before DPA can even be posed. This somewhat contrasts with some of the paper's motivation, i.e.the rebuttal's clarification of "Model selection can be expensive even after candidate models have been trained, because each candidate must be evaluated on a potentially large validation set.", or Section 6's " which candidate should be selected before a final full-data training or deployment step.".  Both read as prospective, decide-before-you-commit motivations, while the formalism is retrospective by construction. I'd ask the authors to clarify which framing is intended, since the direct-acquisition experiments only support the latter.
>
> 2. If I'm correct, the same information assymmetry is also used by VERDICT and the DecisionIF baseline, but not from the remaining ones. I think the authors should be explicit about this asymmetry wherever Table 1 is discussed, since part of VERDICT's advantage over most baselines is an advantage in information access, not only search quality.
>
> 3. I was initially trying to reason about the motivation beyond the auditing aspect from a low-cost angle in the sense of end-to-end latency. However, the rebuttal's answer to "why pay VERDICT's extra cost" is amortization over repeated future use, i.e., the tool is worth it because the compact subset will still be valid later. However, the lacking comparison of full train-validate vs VERDICT efficiency means that it remains unclear how many "audits" are required for the overhead to be compensated. The other side of the economy, storage and evaluationg savings, are rarely the critical constraints in modern ML pipeline, as compute is significantly more critical, making this point even more important.
>
>
> 4. As the work itself concedes, the evaluation and problem are only defined on the same model selection setting. This means the work provides no guarantees or empirical evidence for when a practicioner wants to extend the selection to more models, or settings, and indeed the scenario discovery experiment shows that the retained subset is unlikely to generalize in other settings, making its value limtied, as the moment you add a checkpoint, extend a scenario set, or change a hyperparameter grid, the retained subset's certificate no longer applies.
>
> 5. The remaining motivation seems quite narrow. As scenario discovery is the one application that doesn't depend on any of the above, and it's the one place the paper actually tests generalization outside the searched-over set, it remains a fairly narrow contribution relative to the framing in the abstract and Section 1, which sells DPA as a general answer to "what compact evidence supports a model-selection decision" across deployment slices, checkpoints, and hyperparameter search.
>
> Given the above, I would ask the authors to comment on these issues and add the full-data reference row to the runtime comparison.

---

### Review · Reviewer_kok6 · 2026-07-11

**Summary Of Contributions:**

The paper introduces Decision-Preserving Acquisition (DPA): find the smallest training subset that reproduces a fixed model-selection winner map after retraining. The authors prove realizability is equivalent to KKT feasibility in convex ERM, show cost minimization is NP-hard, and propose VERDICT, a propose-and-verify local search using implicit differentiation and trust-region surrogates. The formulation is novel and the theoretical development is careful, but the core guarantee is too weak to support the claimed applications.

**Audience:**

No

**Audience Explanation:**

The covariate shift problem identified above is structural, not a consequence of experimental design choices. No revision to the experiments can resolve it: as long as subset selection produces a non-representative training or validation distribution, in-protocol consistency carries no guarantee about the full-data conclusion. The problem formulation is therefore not a valid operationalization of the stated goal. A reader who accepts the paper's framing would be misled into believing that decision-preserving subsets are meaningful objects, when in fact the paper only demonstrates that it is possible to find subsets that reproduce a winner map under a shifted distribution—a much weaker and less useful property. The theoretical contributions (KKT equivalence, NP-hardness) are correct in isolation but are motivated by and interpreted through the same flawed framing, limiting their standalone value.

**Broader Impact Concerns:**

The paper's Broader Impact Statement identifies cherry-picking risk. An additional concern: practitioners who rely on VERDICT's in-protocol certificate without understanding its scope may make deployment decisions based on winner maps that do not reflect genuine model superiority on the full population. This risk applies to both Section 5.1 and Section 5.2 and should be addressed explicitly.

**Claims And Evidence:**

No

**Claims Explanation:**

**The fundamental problem: in-protocol consistency is not meaningful consistency.**

Every application the paper proposes requires that the winner map found on the subset generalizes beyond the specific models used to construct it. The paper's guarantee is purely in-protocol: it certifies only that the same winner map is reproduced when the *identical* candidate set is retrained or re-evaluated on the *same* subset. This guarantee is vacuous for any broader use.

**Direct training acquisition (Section 5.1).** All experiments are restricted to fixed-feature convex ERM with frozen pretrained representations, which already excludes end-to-end training, architecture search, and stochastic optimization. Even within this restricted setting, the guarantee is not meaningful: VERDICT selects a non-representative subset $S$, inducing a shifted training distribution $\hat{p}_S(x)$. In convex ERM, the optimal parameters $w_m^*(S)$ are determined by the geometry of $\hat{p}_S(x)$—specifically its first and second moments—rather than those of the full-data distribution $\hat{p}(x)$. The winner map therefore reflects which model best fits $\hat{p}_S(x)$, not which model is genuinely superior. Under model misspecification—inevitable in the fixed-feature linear setting—the decision boundary rotates in directions orthogonal to the quasi-active constraints that VERDICT monitors, and these rotations are uncontrolled. The paper provides no analysis of when in-protocol consistency implies anything about the full-data conclusion.

**Low-cost evaluation subsets (Section 5.2).** The evaluation subset changes the validation distribution from $\hat{p}(x_\text{val})$ to $\hat{p}\_S(x_\text{val})$. The subset is constructed to preserve rankings among a *specific* candidate set $\mathcal{M}$; for any different set of models $\mathcal{M}'$, the loss landscape on $S$ may differ arbitrarily from that on the full validation set. If the subset is used only once for the exact models that constructed it, the guarantee is trivially correct but practically worthless. If the subset is reused for new candidates (the only scenario with practical value), it faces the same distributional shift problem as Section 5.1.

**Requested Changes:**

Given that the issues identified above are structural—no experimental revision can establish that in-protocol consistency implies full-data consistency—there are no changes that would secure a recommendation for acceptance. The authors may wish to consider reframing the contribution around a more limited and defensible claim, such as characterizing the structure of the in-protocol feasibility problem without asserting practical utility, or identifying theoretical conditions under which the covariate shift concern is provably benign.